# MentalChat16K: A Benchmark Dataset for Conversational Mental Health Assistance

## Abstract

We introduce MentalChat16K, an English benchmark dataset combining a synthetic mental health counseling dataset and a dataset of anonymized transcripts from interventions between Behavioral Health Coaches and Caregivers of patients in palliative or hospice care. Covering a diverse range of conditions like depression, anxiety, and grief, this curated dataset is designed to facilitate the development and evaluation of large language models for conversational mental health assistance. By providing a high-quality resource tailored to this critical domain, MentalChat16K aims to advance research on empathetic, personalized AI solutions to improve access to mental health support services. The dataset prioritizes patient privacy, ethical considerations, and responsible data usage. MentalChat16K presents a valuable opportunity for the research community to innovate AI technologies that can positively impact mental well-being.

## 1 Introduction

The proliferation of Large Language Models (LLMs) has transformed artificial intelligence's capability to understand and generate human-like text, unlocking new opportunities for applications in various domains, including mental health support (Hua et al., 2024; van Heerden et al., 2023). This advancement is particularly timely, given the rising global prevalence of mental health disorders such as depression and anxiety (Arias et al., 2022; Organization et al., 2022), highlighting an urgent need for innovative and accessible support solutions (Torous et al., 2021; Lattie et al., 2022). Notably, most recent studies have demonstrated that LLMs, even at the 7B scale, are capable of generating empathic responses that can be more empathic than human-written responses, thus enhancing human peer support in contexts where empathy is crucial (Zhan et al., 2024; Lee et al., 2024).

Recent years have witnessed the emergence of several AI models aimed at addressing mental health challenges, including Psy-LLM (Lai et al., 2023), Mental-LLM (Xu et al., 2023), ChatPsychiatrist (Liu et al., 2023a), and MentalBERT (Ji et al., 2021). Despite the advancements in the field, there is a notable paucity of LLMs that concentrate on mental health counseling. Among the aforementioned work, ChatPsychiatrist is the only open-source English LLM focusing on question-answering in psychological consultation settings. The majority of existing work has primarily emphasized mental health detection, diagnosis, and prediction (Xu et al., 2023; Ji et al., 2021). This disparity can be attributed to several obstacles, including language limitations, a scarcity of domain-specific training data, and privacy concerns surrounding the use of such data.

To overcome these challenges and advance research in this critical domain, we introduce *MentalChat16K*, an English benchmark dataset consisting of 16K question-answer pairs of synthetic mental health counseling conversations and anonymized interview conversations from interventions between Behavioral Health Coaches and Caregivers of patients in palliative or hospice care. This curated dataset covers a diverse range of conditions, including depression, anxiety, and grief, enabling the development and evaluation of LLMs tailored for conversational mental health assistance. Notably, MentalChat16K is twice the size of the training data for ChatPsychiatrist (Liu et al., 2023a), providing a broader and deeper coverage of real-life mental health issues to enhance the capabilities of AI models in offering comprehensive and empathetic support.

MentalChat16K presents a valuable opportunity for the research community to innovate AI technologies that can positively impact mental well-being. By leveraging this dataset, researchers can fine-tune and evaluate LLMs, enabling the development of empathetic and personalized AI solutions

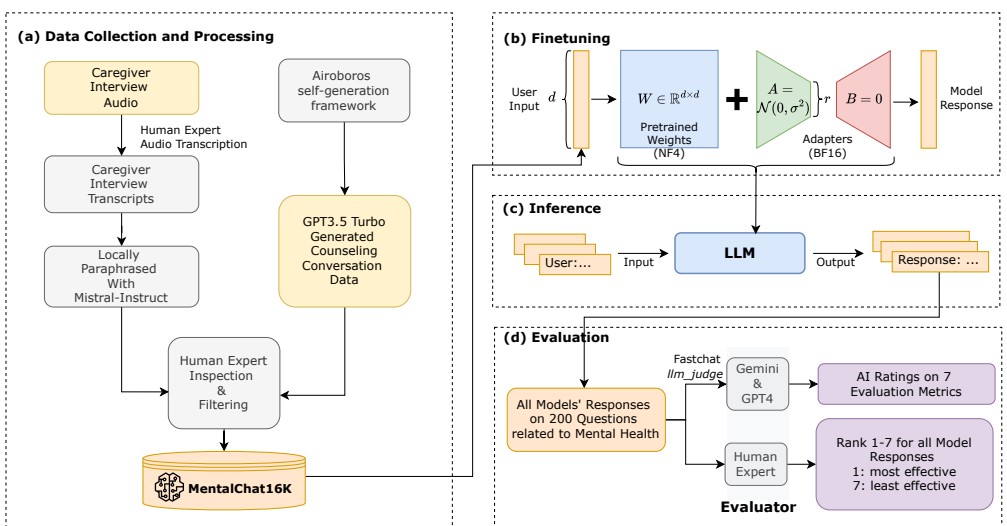

Figure 1: Overall architecture of our approach. (a) Data Collection and Processing: We collect two datasets where one is a synthetic dataset generated by GPT-3.5 Turbo using Airoboros self-generation framework and the other is a real interview transcript dataset paraphrased by a local LLM, Mistral-Instruct. (b) Fine-tuning: We use QLoRA to fine-tune four state-of-the-art light-weight (7B) LLMs on either synthetic dataset, real dataset or their combination. (c) Inference: We curated 200 questions related to mental health to let all the fine-tuned and base models respond respectively. (d) Evaluation: We proposed seven metrics that are widely adopted in the area of mental health and utilize Gemini Pro 1.0, GPT-4 Turbo and human experts as the judges to score those responses.

that can engage in warm and nuanced interactions, simulating the communication typically expected in human counseling sessions. The dataset prioritizes patient privacy, ethical considerations, and responsible data usage, ensuring that the development of AI technologies in this sensitive area is conducted with the utmost care and responsibility.

To demonstrate the effectiveness of the MentalChat16K dataset in tailoring LLMs for mental health counseling, We fine-tuned seven state-of-the-art LLMs. We employed the Quantized Low-Rank Adapter (QLoRA) technique (Dettmers et al., 2024) for efficient fine-tuning of state-of-the-art LLMs, thereby reducing computational demands without sacrificing model performance. We mainly focus on 7B local open-source models because we want to demonstrate a fine-tuned pipeline with limited resources (such as one single A40 or A100 GPU). There are mainly two families of local LLMs in the market, one is the LLaMA family including LLaMA, LLaMA2, Alpaca, Vicuna etc. and the other is the Mistral family including Mistral, Mixtral, Zephyr, etc. Mistral 7B [1] has been validated as one of the most powerful local open-sourced models.

To evaluate the ability of LLMs fine-tuned on our MentalChat16K data in counseling, we curated a specialized counseling evaluation benchmark consisting of 200 questions and developed 7 metrics to rigorously assess the performance of LLMs in the context of mental health counseling. The evaluation is automated by leveraging strong LLMs like GPT-4 Turbo Preview (OpenAI, 2024a) and Gemini Pro 1.0 (Team et al., 2023) as impartial judges. We also incorporated real human evaluations for a more comprehensive and convincing comparison. Our four evaluators include one senior Postdoc, and three Master students , providing interdisciplinary expertise in both computer science and medical sciences, making them well-suited to assess the technical and healthcare aspects of the models' responses. The human evaluation results are consistent with the evaluation results of GPT-4 and Gemini Pro. The complete architecture is summarized in Figure 1.

In summary, our contributions are three-fold:

- We introduce MentalChat16K, a benchmark dataset that contains anonymized transcripts from interventions between Behavioral Health Coaches and Caregivers of patients in pal-

---

[1]https://mistral.ai/news/announcing-mistral-7b/

liative or hospice care. This dataset can be used for fine-tuning pre-trained large language models to provide empathetic, personalized AI solutions to improve access to mental health support services.

• We also generated a synthetic counseling conversation dataset covering a broad range of topics in mental health such as depression, and anxiety. This synthetic data works as a complimentary of the real dataset and composes the complete MentalChat16K benchmark together with the real dataset.

• We provide a pipeline for data collection, data filtering, LLMs fine-tuning, and evaluation. Our extensive experiments demonstrate that the fine-tuned LLMs on the MentalChat16K dataset outperform existing models in providing mental health support, validating the effectiveness of MentalChat16K. This pipeline also serves as a valuable demo for institutions lacking computing resources, enabling them to fine-tune their own large language models.

## 2 RELATED WORK

**Mental Health** Mental health disorders like depression and anxiety have a profound impact, leading to substantial challenges and socio-economic consequences. The global economy faces an estimated annual productivity loss of $1 trillion due to these disorders (National Alliance on Mental Illness, 2023). Depression prevalence among older adults ranges from 7.2% to 49% (Djernes, 2006), even higher than dementia (Allan et al., 2014). AI integration in healthcare, especially through LLMs like Alpaca, GPT, LLaMA, and BERT, offers promising prospects for innovative mental health solutions (Xu et al., 2023; Zhang et al., 2022; Greco et al., 2023).

**LLMs in Mental Health Care** Depression is the leading cause of disability globally (World Health Organization, 2021). LLMs, including GPT3.5, GPT4, LLaMA1, and LLaMA2, have transformed mental health care with their ability to grasp natural language context and produce human-like outputs (Demszky et al., 2023). Researchers have integrated open-source LLMs into mental health chatbots like ChatPsychiatrist (Liu et al., 2023a), MentalBERT (Ji et al., 2021), Mental-LLM (Xu et al., 2023), and Psy-LLM (Lai et al., 2023). LLMs have been employed in various mental health tasks, such as suicide risk detection (Bantilan et al., 2021), psychotherapy homework assignment (Peretz et al., 2023), and emotion recognition (Zhang et al., 2023). They have also aided non-professional counselors (Fu et al., 2023) and supported depression diagnosis and treatment (Wang et al., 2023).

**Benchmark Datasets** Benchmark datasets are crucial for advancing NLP research in mental health. Althoff et al. (2016) presented a large-scale, quantitative study on the SNAP dataset (Althoff et al., 2016), a text-message-based counseling conversation dataset containing over 13 million messages. The PsyQA dataset contains Chinese counseling conversations, and Na et al. used CBT prompts with GPT-3.5-turbo-16k to generate CBT-informed responses (Sun et al., 2021; Na, 2024). CounselChat (Bertagnolli, 2020) includes 3.6k questions and answers from online counseling platforms. The HOPE dataset (Malhotra et al., 2022) includes 12.9k annotated utterances from counseling session videos for dialog-act classification, and the MEMO dataset annotates these for mental health counseling summarization (Srivastava et al., 2022). ChatPsychiatrist's Psych8K dataset (Liu et al., 2023a) comprises data from 260 real counseling recordings. Our MentalChat16K dataset includes face-to-face or video conference conversations, encompassing verbal and non-verbal interactions. These datasets support advancing NLP applications for mental health (Liu et al., 2023a). For a more comprehensive related work, please refer to Appendix A.2.

## 3 APPROACH

This section outlines the methodologies utilized for curating the MentalChat16K datasets, as well as the approaches used for fine-tuning and evaluating LLMs using MentalChat16K. Figure 1 illustrates our pipeline from data collection to model evaluation.

### 3.1 DATA COLLECTION AND PROCESSING

MentalChat16K consists of two datasets. One is the real anonymized interview transcripts between behavioral health coaches and caregivers, and the other is a synthetic mental health counseling

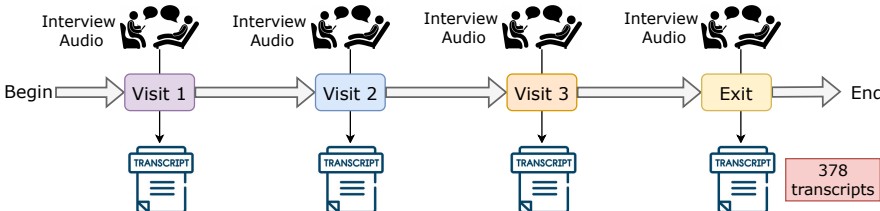

Figure 2: Illustration of behavioral intervention interview data. A caregiver has three formal visits and an exiting visit. Each visit will generate an audio file that will be transcribed into transcripts.

conversation dataset generated by GPT-3.5 Turbo (OpenAI, 2024b). We have provided detailed statistics of both datasets to facilitate understanding and utilization. Table 1 summarizes the key statistics of the MentalChat16K dataset. To illustrate the nature and structure of the datasets, we have included representative examples from both the synthetic and interview data in Appendix A.11.

Table 1: Summary of MentalChat16K Dataset Statistics

| Category | Interview Data | Synthetic Data |
|---|---|---|
| Dataset Size (Rows) | 9,775 | 6,338 |
| Columns | instruction, input, output | instruction, input, output |
| Average Input Word Count | 69.94 | 111.24 |
| Average Output Word Count | 235.85 | 363.94 |
| Number of Sessions | 378 | - |
| Average #QA Pairs per Session | 16.8 | - |
| Number of Topics | - | 33 |

### 3.1.1 INTERVIEW DATA

We collected 378 interview transcripts from an ongoing clinical trial transcribed by human experts based on audio recordings of behavioral intervention sessions between behavior health coaches and caregivers of individuals in palliative or hospice care. The anonymous clinical trial, aims to test a problem-solving therapy intervention to support the emotional needs of hospice caregivers. Upon consent to participate in the study, 514 caregivers were enrolled to randomly receive intervention either face-to-face with a behavioral health coach or via video conference. Figure 2 shows that each caregiver has three formal and one exit visit with a behavioral health coach, generating interview audio files transcribed into text by human experts, ranging from brief greetings to dialogues with filler words. As a token of appreciation for their participation, caregivers receive a $50 reloadable gift card upon completion of the intervention, and $25 upon completion of the 40-day follow-up assessment.

Demographic information was collected from 421 caregivers who completed the information survey, providing insights into their backgrounds. Specifically, the majority of caregivers are female, with 415 out of 421 total participants of the survey. Among female caregivers, White Caucasians constitute the largest group, making up approximately 88% (366 out of 415), with a small proportion identifying as Hispanic (less than 1%). Male caregivers are a small minority, totaling 6, with White Caucasians again being the predominant group. Other racial categories include Asian American, Black/African American, Multi-racial, and others, each comprising a small proportion of the caregiver population. For more details, please refer to Table 4 in the Appendix. The demographic distribution also reflects a real situation in the real world (Chi et al., 2016). Additionally, the skew observed in our dataset aligns with broader trends (Chi et al., 2020) in hospice care and research participation.

To improve data quality by making transcripts more precise, paraphrasing is necessary. Ideally, an LLM like ChatGPT could assist, but privacy concerns prevent uploading patient data to commercial platforms. Therefore, we employed the local Mistral-7B-Instruct-v0.2 (Jiang et al., 2023) model, which is a state-of-the-art lightweight LLM to paraphrase and summarize interview transcripts documents. We fed each page of the 378 transcripts into the model and provided instructions (see Table 5 in the Appendix) to summarize the page into a single round of conversation between the caregiver and the behavioral health coach. Subsequently, we filtered out any conversations with less than 40 words in the question and answer, resulting in a total of 6,338 question-answer pairs. To

ensure privacy and confidentiality, we conducted a manual inspection of the paraphrased transcript to remove any sensitive and identifiable information such as name, address, financial information, and etc. The consent to release the paraphrased transcript data is obtained from the anonymous research group upon removal of all sensitive and identifiable information.

### 3.1.2 SYNTHETIC DATA

To enrich our dataset with diverse therapeutic dialogues, we used the OpenAI GPT-3.5 Turbo (OpenAI, 2024b) model to generate 9,775 question-answer pairs with a customized adaptation of the Airoboros self-generation framework [2]. Under the Airoboros framework, we customized a new prompt (see Table 6) to provide clear instructions to generate patient queries using GPT-3.5 Turbo. These queries were then fed back into GPT-3.5 Turbo to generate corresponding responses. These synthetic conversations covered 33 mental health topics, including Relationships, Anxiety, Depression, Intimacy, Family Conflict, etc. The proportion of each topic (Appendix A.3.2) that typically arises in a counseling session according to the CounselChat (Bertagnolli, 2023) platform was specified in the prompt. This method ensures the synthetic conversations authentically mimic the complexity and diversity of therapist-client interactions, thereby exposes our models to a wide spectrum of psychological conditions and therapeutic strategies.

### 3.2 FINE-TUNING AND INFERENCE

To perform efficient fine-tuning by using only one A40 or A100 GPU that is more affordable, we adopt Quantized Low Rank Adaptation (QLoRA) (Dettmers et al., 2024). For further details, please refer to Appendix A.1.

The inference stage involved using both the fine-tuned and base models, alongside baseline models (Samantha v1.11 and v1.2 (Cognitive Computations Group, 2023), ChatPsychiatrist (Liu et al., 2023a)), to generate responses to 200 sampled questions. These questions were collected from Reddit (InFamousCoder, 2022) and the Mental Health Forum (Forum, 2023), representing a wide range of real-world inquiries in a therapeutic setting. In addition to the questions, the models were given explicit instructions as follows.

> "You are a helpful and empathetic mental health counseling assistant, please answer the mental health questions based on the user's description. The assistant gives helpful, comprehensive, and appropriate answers to the user's questions".

### 3.3 EVALUATION

We employed GPT-4 Turbo (OpenAI, 2024a) and Gemini Pro 1.0 (Team et al., 2023) as robust and scalable judges for automated LLM evaluation. We utilized the LLM Judge framework (Zheng et al., 2024) to generate judgments and ratings that assess the quality of the models' responses to the benchmark questions we collected. We instructed GPT-4 Turbo and Gemini Pro 1.0 to be objective and assess the response based on 7 devised mental health metrics (see Table 2). Our proposed seven metrics for evaluating therapeutic dialogue systems are grounded in both established therapeutic practices and recent advancements in AI-based mental health support evaluation. These metrics synthesize multiple validated frameworks, primarily building upon ChatCounselor's (Liu et al., 2023a) evaluation methodology while incorporating insights from contemporary research. The Active Listening metric, derived from Miller and Moyers' (Miller & Moyers, 2006) foundational work and validated in AI contexts through PsyQA (Sun et al., 2021), assesses the system's comprehension and reflection capabilities. Empathy & Validation draws from EPITOME's (Sharma et al., 2020) empirically validated empathy metrics and Rogers' (Rogers, 1957) therapeutic principles. Safety & Trustworthiness incorporates criteria from Dialogue Safety (Qiu et al., 2023) and PsyEval (Li et al., 2023b), addressing critical aspects of therapeutic interaction safety. The remaining metrics—Open-mindedness & Non-judgment, Clarity & Encouragement, Boundaries & Ethical, and Holistic Approach—integrate established therapeutic principles (Kabat-Zinn, 2013; Rogers, 1957) with recent frameworks from PsyQA (Sun et al., 2021). This comprehensive framework ensures thorough evaluation of AI systems' therapeutic capabilities while maintaining alignment with profes-

---

[2]https://github.com/jondurbin/airoboros

Table 2: LLMs evaluation metrics on mental health.

| Strategy | Description | References |
|---|---|---|
| Active Listening | Responses demonstrate careful consideration of user concerns, reflecting understanding and capturing the essence of the issue. Avoid assumptions or jumping to conclusions. | Miller & Moyers (2006), Sun et al. (2021), Liu et al. (2023a) |
| Empathy & Validation | Convey deep understanding and compassion, validating feelings and emotions without being dismissive or minimizing experiences. | Sharma et al. (2020), Rogers (1957), Liu et al. (2023a) |
| Safety & Trustworthiness | Prioritize safety, refrain from harmful or insensitive language. Ensure the information provided is consistent and trustworthy. | Qiu et al. (2023), Lambert & Barley (2001), Li et al. (2023b) |
| Open-mindedness & Non-judgment | Approach without bias or judgment. Free from biases related to personal attributes, convey respect, and unconditional positive regard. | Rogers (1957), Sun et al. (2021), Kabat-Zinn (2013) |
| Clarity & Encouragement | Provide clear, concise, and understandable answers. Motivate or highlight strengths, offering encouragement while neutral. | Liu et al. (2023a), Li et al. (2023b) |
| Boundaries & Ethical | Clarify the response's role, emphasizing its informational nature. In complex scenarios, guide users to seek professional assistance. | Qiu et al. (2023), Liu et al. (2023a) |
| Holistic Approach | Be comprehensive, addressing concerns from various angles, be it emotional, cognitive, or situational. Consider the broader context, even if not explicitly detailed in the query. | Sun et al. (2021), Liu et al. (2023a) |

sional standards (American Psychological Association, 2017) and empirically validated assessment approaches.

The judge models were tasked to rate each response for each metric on a scale ranging from 1 to 10. In addition, we asked the judge models to justify their ratings and make comments on the model responses. Please refer to Table 7, 8 in Appendix A.4 for a detailed prompt and scoring rubrics used in the evaluation. However, we acknowledge the potential limitations and biases inherent in LLM-based evaluations, especially in specialized fields like mental health counseling. To address this, we incorporated human evaluation to complement the automated assessments and ensure a more comprehensive evaluation of the models' performance. To demonstrate the significance of our results, we conducted statistical analyses by randomly selecting 50 questions from the original 200 test questions and running five rounds of inference on both fine-tuned and base models. We compared the average scores across all five rounds on each of the seven metrics between the fine-tuned and base models through a two-sample t-test with a 0.95 confidence interval. The null hypothesis is that the scores for the fine-tuned and the base models have identical average (expected) values across the specified metrics.

To incorporate human evaluation of the model performance, we invited a senior Postdoc and three Master's students who possessed interdisciplinary expertise in both computer science and medical sciences to compare the responses generated by the base models, fine-tuned models, and baseline models. For each input question, the participants ranked the responses from the following models: (1) the base model, (2) the base model fine-tuned on synthetic data, (3) the base model fine-tuned on interview data, (4) the base model fine-tuned on both synthetic and interview data, and three baseline models: (5) Samantha-1.1, (6) Samantha-1.2, and (7) ChatPsychiatrist. The responses were ranked from 1 to 7, with 1 being the most effective response and 7 being the least effective. For each model, the final ranking results were calculated as the average ranking across 50 input questions randomly sampled from the 200-question evaluation dataset described in Section 3.2.

To make the human evaluation more reliable, we calculate the Cohen's Kappa score (Landis & Koch, 1977) to assess the consistency among the evaluators. Specifically, we randomly selected 30 questions from the 200 evaluation questions and had responses generated by 7 models (3 global baselines including Samantha-1.11, Samantha-1.2 and ChatPsychiatrist, 1 randomly selected base model before fine-tuning such as Zephyr-Alpha, Vicuna-7B-V1.5, LLaMA2-7B, Mistral-7B-V0.1, Mistral-7B-Instruct-V0.2, Mixtral-8x7B-V0.1, Mixtral-8x7B-Instruct-V0.1, and its corresponding 3 fine-tuned models fine-tuned on synthetic, interview and both data respectively). Four human evaluators ranked these responses from 1 (best) to 7 (worst). By treating each rank as the prediction target, we make the task become a 7-class classification problem and each human evaluator will generate a list of predictions for the 30 questions. We calculate Cohen's Kappa score among all the human evaluators' lists of predictions. The resulting agreement is 0.441, which is larger than the acceptable threshold 0.4 as indicated by Landis & Koch (1977).

## 4 EXPERIMENT

Our study aims to investigate the effectiveness of fine-tuning LLMs using MentalChat16K for mental health counseling. By fine-tuning LLMs with this specialized dataset, we aim to enhance the models' capacity to generate empathetic, relevant, and contextually appropriate responses in mental health counseling scenarios. This section details the methodology, implementation, and evaluation metrics employed to assess the performance improvements of the fine-tuned models.

### 4.1 BASELINE MODELS

We selected three baseline models, chosen for their relevance and pioneering contributions to AI-assisted mental health support, setting a benchmark for our fine-tuned models' comparative analysis.

**ChatPsychiatrist** (Liu et al., 2023a) is an instruction-tuned LLM fine-tuned on LLaMA-7B (Touvron et al., 2023a) using the Psych8k dataset, composed of authentic dialogues between clients and psychologists. This model outperformed other open-source solutions such as Alpaca-7B (Taori et al., 2023), LLaMA-7B(Touvron et al., 2023b), and ChatGLMv2-6B (Du et al., 2022) on the counseling Bench the authors devised.

**Samantha-v1.11/v1.2** (Cognitive Computations Group, 2023) are open-source models hosted on Hugging Face, fine-tuned on the LLama-2-7B (Touvron et al., 2023b) and Mistral-7B (Jiang et al., 2023) respectively. Unique for their training in philosophy, psychology, and personal relationships, Samantha models are designed as sentient companions.

### 4.2 BASE MODELS FOR FINE-TUNING

To improve LLM's mental health support capabilities, we've chosen a variety of base models for fine-tuning, each with unique strengths.

**LLaMA-2-7B** (Touvron et al., 2023b) is a well-known pre-trained model developed by Meta, recognized for its scalability and efficiency, and is included for its adaptability and deep language understanding.

**The Mistral Series** comprises four models. *Mistral-7B-v0.1* (Jiang et al., 2023) is a pre-trained LLM engineered for superior performance and efficiency. It outperforms LLaMA2-13B across all tested benchmarks. *Mixtral-8x7B-v0.1* (Jiang et al., 2024) is an advanced generative Sparse Mixture of Experts model. It outperforms LLaMA2-70B on most benchmarks tested. *Mistral-7B-Instruct-v0.2* (Jiang et al., 2023) and *Mixtral-8x7B-Instruct-v0.1* (Jiang et al., 2024) are instruction fine-tuned versions of *Mistral-7B-v0.1* and *Mixtral-8x7B-v0.1*, trained on a variety of publicly available conversation datasets.

**Vicuna-7B-v1.5** (LMsys, 2023) is a chat assistant developed by fine-tuning LLama 2 on user-shared conversations gathered from ShareGPT. It can provide nuanced empathy and understanding, which is essential for effective mental health support.

**Zephyr-7B-Alpha** (Tunstall et al., 2023) is the first in the series of assistant-oriented language models, and is a fine-tuned version of Mistral-7B-v0.1 from Mistral AI. It is trained on a combination of publicly available and synthetic datasets using DPO (Rafailov et al., 2024).

### 4.3 METRICS

In the current landscape of LLM evaluation, several metrics dominate the literature. Common performance measures include perplexity, accuracy (Hendrycks et al., 2021; Clark et al., 2018), semantic similarity (Risch et al., 2021; Bulian et al., 2022), and human evaluation metrics such as fluency, coherence, and relevance (Chiang & yi Lee, 2023). While these metrics offer valuable insights into general-purpose LLM performance across various tasks such as Question-Answering (QA) and multiple-choice, they often fall short when it comes to evaluating LLMs tailored for mental health counseling. Mental health LLMs require nuanced assessments that go beyond traditional language generation tasks, focusing on empathy, sensitivity to emotional nuances, and adherence to ethical guidelines (Li et al., 2024). Current metrics lack the specificity and sensitivity required to gauge these aspects accurately. To address this gap, we devised seven metrics (shown in Table 2)

for evaluating mental health LLMs. These novel metrics aim to provide a comprehensive evaluation framework that better aligns with the unique requirements of mental health counseling applications.

## 4.4 SETUP

Our models were trained using two types of data from MentalChat16K: a real interview dataset and a synthetic dataset. We hypothesized that models fine-tuned on MentalChat16K would exhibit enhanced performance on various mental health counseling evaluation metrics, indicative of a more nuanced understanding of patient interactions.

Each base model underwent fine-tuning under three distinct configurations:

- Fine-tuning with Synthetic Data: Models were fine-tuned exclusively on the synthetic dataset to assess the impact of scenario-based learning.
- Fine-tuning with Interview Data: Models were fine-tuned using real-world interview data, aiming to enhance their understanding of natural conversational dynamics.
- Hybrid Fine-tuning: Models were fine-tuned using the entire MentalChat16K dataset, testing the hypothesis that a diverse training input could yield superior performance.

We fine-tuned each model over five epochs, using a batch size of 64 and a maximum output sequence length of 1024. The pre-trained weights of models were initially loaded with 4-bit precision and subsequently dequantized to 16 precision for computations. Additionally, we enabled double quantization during fine-tuning to enhance model efficiency. We set the LoRA hyperparameters as follows: $r = 64$, $\alpha = 16$, and dropout $= 0.1$, where $\alpha$ determines the magnitude of the impact of updates on the original weights of the pre-trained model, while $r$ defines the rank of the low-rank matrices that approximate these updates. Through these settings, we managed to reduce the number of trainable parameters to approximately 2.14% of the total model parameters. The training process was conducted on a single NVIDIA A100 GPU (80 GB). For the complete set of hyperparameters used during fine-tuning, see Appendix A.9.

## 4.5 RESULTS

**Main Results** In Table 3, the evaluation scores reveal distinct patterns in model performance when assessed by GPT 4 Turbo, Gemini Pro, and human experts. The results show clear patterns in model performance for both evaluation methods. GPT 4, Gemini Pro and human experts evaluation results indicate that fine-tuning models on synthetic data, interview data, or both generally leads to improved performance across all metrics compared to their base models, validating the effectiveness of our MentalChat16K. Refer to Table 9 in Appendix A.5 for the complete results and Appendix A.7 for a complete visualization of the results.

**Discussion on GPT-4 Evaluation** GPT-4's evaluations reveal a consistent pattern favoring models fine-tuned on synthetic data (indicated by *). For example, in "Active Listening", for all the seven base models, the fine-tuned version on synthetic data generated by GPT 3.5 Turbo outperforms the remaining three models including the base model, the model fine-tuned on the interview data and the model fine-tuned on both datasets. The winning times are 6, 7, 7, 7, 7, 7 out of 7 for the other six metrics respectively. This bias towards models fine-tuned on synthetic data may be attributed to GPT-4's alignment with data generated by GPT-3.5 Turbo, possibly introducing an intrinsic preference for similar language patterns and styles. This bias indicates that it may not be fair to merely use GPT-4 as the evaluator. Therefore, we incorporate Gemini Pro from Google as another LLM evaluator.

**Discussion on Gemini Evaluation** In contrast, Gemini's evaluations, while also acknowledging the improvements brought by synthetic data, seem to place more value on the depth and realism provided by interview data, particularly in metrics related to Safety & Trustworthiness and Boundaries & Ethical. The winning times of the version fine-tuned on the interview data compared to the other three models are 7, 7, 7, 6, 4, 5, and 6 out of seven cases in terms of the seven metrics separately. Gemini Pro's evaluations suggest that the interview data contributes significantly to the models' performance in aspects that require real-world contextual understanding and nuanced human interactions. The performance of the model fine-tuned on the combination data also has a chance to outperform the other three models under the evaluation of Gemini Pro. Gemini Pro's evaluations suggest that while synthetic data can contribute to conversational diversity, the integration of real-world dialogues is crucial for achieving the depth of engagement and empathy required in mental health support.

Table 3: Comparison of evaluation scores rated by GPT-4 Turbo and Gemini Pro across all models on 7 mental health metrics, as well as human rankings of model responses on a scale of 1 to 7. In each two-column cell, the best score evaluated by GPT-4 Turbo is highlighted in red, and the best score evaluated by Gemini Pro is highlighted in blue. The score with a significant P-value ($< 0.05$) is marked with ●. The last column contains the average ranking of each model evaluated by humans. The bold numbers represent the highest average rank among the four models in one block and the three baseline models. The ranking procedure is described in Section 3.3. This result showed that fine-tuning on MentalChat16K significantly improved the performance of base LLMs. The model fine-tuned on synthetic data alone outperforms the other three cases most of the time according to GPT-4 and human evaluations. The model fine-tuned on real interview data alone outperforms the other three cases most of the time when using Gemini Pro for evaluation. Models fine-tuned on the entire MentalChat16K also significantly outperform their base model most of the time.

| Model (7B) | Active Listening ↑ | | Empathy & Validation ↑ | | Safety & Trustworthiness ↑ | | Open-mindedness & Non-judgment ↑ | | Clarity & Encouragement ↑ | | Boundaries & Ethical ↑ | | Holistic Approach ↑ | | Human Rank ↓ |
|---|---|---|---|---|---|---|---|---|---|---|---|---|---|---|---|
| | GPT | Gemini | GPT | Gemini | GPT | Gemini | GPT | Gemini | GPT | Gemini | GPT | Gemini | GPT | Gemini | |
| LLaMA2 | 2.32 | 5.61 | 2.47 | 5.60 | 2.49 | 5.76 | 2.93 | 5.96 | 2.38 | 5.32 | 2.46 | 5.56 | 2.11 | 5.29 | 4.45 |
| LLaMA2 † | 7.23● | 8.06● | 8.10● | 8.39● | 6.97● | 7.78● | 8.30● | 8.38● | 7.10● | 7.69● | 6.66● | 7.67● | 6.86● | 8.00● | 3.79 |
| LLaMA2 * | 7.63● | 8.01● | 8.46● | 8.22● | 7.53● | 7.63● | 8.70● | 8.26● | 7.69● | 7.66● | 7.34 ● | 7.63 ● | 7.46● | 7.95● | 3.65 |
| LLaMA2 *† | 7.58● | 8.06● | 8.47● | 8.35● | 7.40● | 7.68● | 8.60● | 8.32● | 7.58● | 7.69● | 7.06● | 7.68● | 7.21● | 7.97● | **3.55** |
| Mistral-Instruct-V0.2 | 7.77 | 8.08 | 8.67 | 8.42 | 7.84 | 7.86 | 8.74 | 8.34 | 7.76 | 7.76 | 7.48 | 7.78 | 7.34 | 8.01 | 3.20 |
| Mistral-Instruct-V0.2 † | 7.33● | 8.13● | 8.21● | 8.51● | 7.05● | 7.90● | 8.46● | 8.47● | 7.15● | 7.79● | 6.73● | 7.83● | 7.01● | 8.12● | 2.55 |
| Mistral-Instruct-V0.2 * | 7.87 | 8.04 | 8.78 | 8.30 | 7.87 | 7.75 | 8.86 | 8.31 | 7.90 | 7.73 | 7.66 | 7.71 | 7.76 | 7.98 | **2.35** |
| Mistral-Instruct-V0.2 *† | 7.60 | 8.13● | 8.45 | 8.38● | 7.38 | 7.89 | 8.65 | 8.36● | 7.54 | 7.81● | 7.08● | 7.83● | 7.26 | 8.12● | 3.10 |
| Mistral-V0.1 | 5.15 | 7.20 | 5.69 | 7.19 | 5.63 | 7.05 | 7.04 | 7.31 | 5.70 | 6.68 | 5.80 | 6.90 | 4.77 | 6.35 | 5.15 |
| Mistral-V0.1 † | 7.25● | 8.23● | 8.16● | 8.57● | 7.06● | 7.98● | 8.36● | 8.52● | 7.15● | 7.82● | 6.69● | 7.92● | 6.98● | 8.24● | 3.30 |
| Mistral-V0.1 * | 7.68● | 8.05 | 8.52● | 8.33 | 7.64● | 7.69 | 8.74● | 8.35 | 7.71● | 7.70 | 7.27● | 7.67 | 7.46● | 8.03 | **1.90** |
| Mistral-V0.1 *† | 7.56● | 8.11● | 8.44● | 8.41● | 7.39● | 7.79● | 8.60● | 8.36● | 7.55● | 7.77● | 7.13● | 7.75● | 7.22● | 8.09● | 2.60 |
| Mixtral-8x7B-Instruct-V0.1 | 4.90 | 4.81 | 5.36 | 4.58 | 6.48 | 5.83 | 7.25 | 5.98 | 5.24 | 4.69 | 7.40 | 6.56 | 4.26 | 4.32 | 6.25 |
| Mixtral-8x7B-Instruct-V0.1 † | 7.53● | 8.11● | 8.43● | 8.39● | 7.22● | 7.77● | 8.56● | 8.34● | 7.31● | 7.68● | 6.81● | 7.72● | 7.13● | 8.06● | 3.10 |
| Mixtral-8x7B-Instruct-V0.1 * | 7.89● | 8.06● | 8.78● | 8.32● | 7.78● | 7.75● | 8.88● | 8.31● | 7.86● | 7.79● | 7.53 | 7.72● | 7.79● | 8.04● | **1.55** |
| Mixtral-8x7B-Instruct-V0.1 *† | 7.69● | 8.03● | 8.49● | 8.35● | 7.36● | 7.71● | 8.67● | 8.40● | 7.61● | 7.76● | 7.12 | 7.74● | 7.27● | 8.07● | 2.70 |
| Mixtral-8x7B-V0.1 | 6.07 | 7.22 | 6.68 | 7.27 | 6.68 | 7.19 | 7.76 | 7.34 | 6.29 | 6.61 | 6.54 | 6.92 | 5.45 | 6.36 | 5.60 |
| Mixtral-8x7B-V0.1 † | 7.47● | 8.10● | 8.30● | 8.44● | 7.15● | 7.78● | 8.39● | 8.42● | 7.25● | 7.70● | 6.82● | 7.73● | 7.09● | 8.11● | 2.55 |
| Mixtral-8x7B -V0.1 * | 7.88● | 8.07● | 8.77● | 8.28● | 7.82● | 7.70● | 8.85● | 8.33● | 7.93● | 7.72● | 7.62● | 7.72● | 7.76● | 8.02● | **1.90** |
| Mixtral-8x7B-V0.1 *† | 7.63● | 8.08● | 8.44● | 8.32● | 7.30● | 7.71● | 8.63● | 8.34● | 7.56● | 7.71● | 6.94● | 7.69● | 7.21● | 8.05● | 3.05 |
| Vicuna-V1.5 | 6.74 | 7.73 | 7.45 | 7.81 | 6.74 | 7.33 | 8.17 | 7.82 | 6.88 | 7.12 | 6.82 | 7.23 | 6.12 | 6.88 | 3.75 |
| Vicuna-V1.5 † | 7.46● | 8.11● | 8.32● | 8.39● | 7.20● | 7.83● | 8.54● | 8.34● | 7.39● | 7.73● | 6.91 | 7.77● | 7.12● | 8.08● | 3.85 |
| Vicuna-V1.5 * | 7.66● | 8.03● | 8.54● | 8.25● | 7.59● | 7.62● | 8.70● | 8.27● | 7.70● | 7.58● | 7.12● | 7.58● | 7.37● | 7.91● | 4.00 |
| Vicuna-V1.5 *† | 7.52● | 8.01● | 8.36● | 8.36● | 7.30● | 7.69● | 8.53● | 8.34● | 7.54● | 7.67● | 6.97● | 7.65● | 7.08● | 7.94● | **3.37** |
| Zephyr-Alpha | 7.28 | 7.97 | 7.95 | 8.02 | 7.18 | 7.64 | 8.50 | 8.08 | 7.36 | 7.63 | 7.15 | 7.59 | 6.81 | 7.61 | 4.20 |
| Zephyr-Alpha † | 7.51● | 8.11● | 8.37● | 8.47● | 7.05 | 7.86● | 8.51 | 8.39● | 7.39 | 7.81 | 6.71 | 7.83● | 7.09● | 8.08● | 2.90 |
| Zephyr-Alpha * | 7.67● | 8.05● | 8.55● | 8.30 | 7.60● | 7.61 | 8.71● | 8.33● | 7.73● | 7.66 | 7.27● | 7.58 | 7.38● | 7.99● | **2.05** |
| Zephyr-Alpha *† | 7.66● | 8.09● | 8.53● | 8.35● | 7.54● | 7.73● | 8.64● | 8.37● | 7.65● | 7.71● | 7.16● | 7.68● | 7.35● | 8.07● | 2.85 |
| ChatPsychiatrist § | 6.46 | 7.54 | 6.74 | 7.48 | 6.45 | 7.28 | 7.98 | 7.68 | 6.49 | 6.88 | 6.68 | 7.19 | 5.54 | 6.40 | 5.74 |
| Samantha-V1.11 § | 6.81 | 7.90 | 7.40 | 8.12 | 6.77 | 7.59 | 8.20 | 8.16 | 6.98 | 7.57 | 6.66 | 7.51 | 6.43 | 7.58 | 4.61 |
| Samantha-V1.2 § | 6.89 | 7.96 | 7.64 | 8.02 | 6.77 | 7.56 | 8.35 | 8.10 | 7.15 | 7.59 | 6.75 | 7.53 | 6.54 | 7.55 | 4.22 |

Notes. No label: Base Model, †: Model fine-tuned on Interview Data (6K), *: Model fine-tuned on Synthetic Data (10K), *†: Model fine-tuned on both Synthetic and Interview Data (16K), §: Baseline Model.

However, the models fine-tuned on the combined data did not consistently outperform those fine-tuned on individual datasets, indicating that simply combining synthetic and interview data may not be the most effective approach. These discrepancies between GPT-4 and Gemini Pro evaluations highlight the potential biases and limitations of relying solely on LLM-based evaluations, especially when different LLMs may have inherent preferences based on their training data and architectures.

**Discussion on Human Evaluation** The human ranking results, shown in the last column of Table 3, clearly indicate that fine-tuned models significantly outperform their base models and the baseline model in the context of mental health counseling. Notably, human evaluators often preferred models fine-tuned on synthetic data, but in several cases, models fine-tuned on interview data or both datasets also performed well. This aligns with GPT-4's and Gemini Pro's evaluations. This trend underscores the effectiveness of our MentalChat16K dataset and fine-tuning pipeline in enhancing the LLMs' capabilities for mental health applications.

**Significance Analysis** Additionally, we conducted statistical analyses to demonstrate the significance of our results. We randomly selected 50 questions from the original 200 test questions and ran five rounds of inference on both fine-tuned and base models. Using GPT-4 Turbo and Gemini Pro for evaluation, we compared the average scores across all five rounds on each of the seven mental health metrics between the fine-tuned models and their base model by running a two-sample t-test with a 0.95 confidence interval. For Gemini Pro evaluation, 18 of 21 fine-tuned models showed significant differences from their base model in at least 6 of 7 metrics. For GPT-4 Turbo, 17 of 21 fine-tuned models showed significant differences from their base model in at least 6 of 7 metrics. We use ● to mark the scores with significant P-values. The more detailed results have been put in Appendix A.5.

## 5 ETHICAL CONSIDERATIONS

Participants in the study signed an informed consent document outlining the risks and benefits of the study. All anonymous study sessions were conducted in a private environment to ensure

confidentiality and privacy. These sessions were recorded using team-provided devices and stored on secure institutionally backed cloud servers. Study data was captured and stored on institutionally backed data management software. Audio files and study data were labeled with unique identifiers and without the inclusion of personal identifying information. Participants' personal identifying information was stored on a separate database linked to the study data through the unique identifier. Only members of the research team had access to this data. Our study aims to maintain high ethical standards, focusing on safety and privacy. While our evaluations did not reveal errors or hallucinations, we acknowledge such risks with pre-trained LLMs in mental health tasks and advise against their current practical application.

## 6 Conclusions

In this paper, we propose MentalChat16K, a benchmark dataset that significantly advances the field of AI-driven mental health support. By incorporating both synthetic counseling conversations and anonymized real-life intervention interview data, MentalChat16K addresses the critical need for domain-specific training data, facilitating the development of large language models capable of empathetic and personalized interactions. Our comprehensive evaluation framework, utilizing state-of-the-art models and advanced metrics, demonstrates the superior performance of LLMs fine-tuned on this dataset in delivering nuanced and compassionate mental health assistance. This work not only highlights the transformative potential of AI in augmenting mental health services but also establishes a new standard for ethical and effective AI development in this sensitive and vital domain.

## 7 Limitations

Despite the promising advancements facilitated by MentalChat16K, several limitations need to be acknowledged. First, the reliance on synthetic data generated by GPT-3.5 Turbo may introduce biases or lack the depth of real human interactions. Table 3 showed that combining synthetic and interview data during fine-tuning did not consistently improve model performance; in some cases, it led to performance degradation. Moreover, the interview data focuses on specific caregivers and patients, which differs significantly from the broad profile of the synthetic data. This suggests that users should handle the synthetic and interview datasets separately and exercise caution when combining them.

Second, the anonymization process, while crucial for privacy, may inadvertently strip conversations of contextual nuances essential for effective mental health support. Breaking down interviews into individual question-answer pairs may also result in the loss of broader conversational context, potentially affecting the quality of generated responses. Additionally, the paraphrasing process may introduce minor deviations or potential hallucinations. To mitigate this effect, paraphrasing was designed to preserve the essence of each interaction while making the QA pairs as self-contained as possible. Additionally, the original order of the QA pairs is maintained, allowing downstream processes to reconstruct context if needed.

Additionally, the dataset primarily focuses on English, limiting its applicability to non-English-speaking regions where cultural and linguistic differences play a significant role in mental health counseling. Furthermore, the interview data was collected from a specific group of caregivers offering care to patients in palliative or hospice care, which may limit the generalizability of the findings to other populations.

Finally, the evaluation framework, though robust, relies heavily on automated assessments and selected human evaluations, which might not fully capture the complexities of real-world counseling efficacy. The use of LLMs as evaluators introduces potential biases, and inconsistencies were observed between different evaluators. Advanced evaluation frameworks like G-Eval (Liu et al., 2023b) and Prometheus ((Kim et al., 2023)) were not utilized and could provide more nuanced assessments in future work.

It is also important to recognize that, while human evaluation is crucial for assessing the quality of LLM output, it inevitably introduces subjectivity and bias from personal preferences. Although we calculated inter-rater agreement to assess consistency among human evaluators, the moderate agreement level indicates room for improvement in evaluation methodologies. Addressing these limitations in future work will be crucial for further enhancing the utility and impact of AI-driven mental health support systems.

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

# A APPENDIX

## A.1 DETAILS OF FINE-TUNING TECHNIQUE

To perform efficient fine-tuning by using only one GPU that is more affordable, we adopt Quantized Low-Rank Adaptation (QLoRA) (Dettmers et al., 2024). QLoRA is a technique designed to optimize the fine-tuning process of LLMs, making it more efficient in terms of computational resources and time. QLoRA is based on Low-Rank Adaptation (LoRA) (Hu et al., 2021) which is a technique that compresses the update weight matrix $\Delta W \in \mathbb{R}^{d \times k}$ (often termed adapters) of the pre-trained weight matrix $W \in \mathbb{R}^{d \times k}$ by decomposing $\Delta W$ into two low-rank matrices, represented by $\Delta W = AB$, where $A \in \mathbb{R}^{d \times r}$ follows the Gaussian distribution, $B \in \mathbb{R}^{r \times k}$ is initialized to zero, and $r \ll \min(d, k)$. Here $r, d, k$ refer to rank, input dimension and output dimension respectively. $A$ and $B$ containing the trainable parameters are updated through backpropagation during fine-tuning while $W$ remains frozen. The forward pass is then represented as:

$$Y = XW + X\Delta W = XW + XAB \tag{1}$$

LoRA reduces the number of trainable parameters and accelerates computation. QLoRA enhances the LoRA method via *4-bit NormalFloat (NF4) Quantization* and *Double Quantization*. Quantization involves converting the precision of model's weights from higher precision representation (e.g. 32-bit floating-point number) to lower precision format (e.g. 8-bit fixed-point number). In QLoRA, model's pre-trained weight matrix $W$ is quantized and preserved in NF4 datatype. The trainable weights in the **A** and **B** are stored as 16-bit BrainFloat (BF16) datatype to perform computational operations. Double quantization further reduces memory usage by further quantizing the quantization constants. QLoRA stores the quantization constants $c$ in 8-bit floating-point numbers. The forward pass in Eq. (1) is then transformed to Eq. (2) in QLoRA:

$$\begin{aligned} Y^{BF16} = X^{BF16}\text{DDeq}(c^{FP32}, c^{FP8}, W^{NF4}) \\ + X^{BF16}A^{BF16}B^{BF16} \end{aligned} \tag{2}$$

where $\text{DDeq}(\cdot)$ is the double dequantization that first dequantizes the quantization constants and then the pre-trained weight matrix into the computational datatype BF16. These techniques together reduce the memory footprint of LLMs, making it possible to fine-tune LLMs with billions of parameters on a single GPU.

## A.2 RELATED WORK

**Mental Health** The significance of mental health often receives less attention compared to physical health, despite its profound impact on individuals and societies globally. Mental health disorders, encompassing conditions such as depression and anxiety, lead to substantial challenges, affecting personal well-being and causing widespread socio-economic consequences. The global economy faces an estimated annual productivity loss of approximately \$1 trillion due to these disorders, highlighting the urgent need for effective solutions and interventions (National Alliance on Mental Illness, 2023). The prevalence of depression among individuals aged 65 and older varies significantly, ranging from 7.2% to 49%, depending on various factors including living conditions (Djernes, 2006). Surprisingly, depression has been identified as more prevalent than dementia within this demographic, underscoring the critical need for addressing mental health issues among the elderly (Allan et al., 2014). In this evolving landscape, the integration of AI in healthcare, particularly through the development of LLMs such as Alpaca, GPT, LLaMA, and BERT, offers promising prospects for groundbreaking research and the creation of innovative mental health solutions (Xu et al., 2023; Zhang et al., 2022; Greco et al., 2023).

**LLMs in Mental Health Care**    In 2021, WHO highlighted depression as one of the primary causes of disability across the globe (World Health Organization, 2021). Moreover, a range of mental health disorders, including those stemming from depression, anxiety, acute panic, obsessive tendencies, paranoia, and hoarding, has significantly added to the worldwide disease burden (Dubey et al., 2020). The introduction of LLMs, notably OpenAI's GPT3.5 and GPT4, as well as Meta's LLaMA1 and LLaMA2, has brought transformative changes to several sectors, including mental health care. These advanced algorithms, built upon cutting-edge deep learning frameworks like transformer and self-attention, are trained on extensive text datasets. This training empowers them to grasp the nuanced semantic context of natural language and produce human-like textual outputs based on the given context (Demszky et al., 2023). As the application of LLMs in healthcare systems continues to grow, researchers are actively integrating the open-source LLMs into independent mental health chatbot, including Psy-LLM (Lai et al., 2023), Mental-LLM (Xu et al., 2023), ChatPsychiatrist (Liu et al., 2023a), MentalBERT (Ji et al., 2021), etc.

Historically, AI applications, especially those involving NLP, have been around for several decades (Weizenbaum, 1966). Since then, AI has been employed in various mental health tasks, such as: detecting suicide risk (Bantilan et al., 2021), assigning homework during psychotherapy sessions (Peretz et al., 2023), and recognizing patient emotions during therapy (Zhang et al., 2023). The newer LLMs have demonstrated exceptional capabilities in diverse tasks, including reasoning, natural language comprehension and generation, and problem-solving (Li et al., 2023a). For instance, LLMs like GPT3.5 have been instrumental in aiding non-professional counselors in delivering responses to patients (Fu et al., 2023), and depression diagnosis and treatment (Wang et al., 2023).

LLMs have also been evaluated for various mental health prediction tasks via online text data, showing that instruction fine-tuning can significantly boost the performance of LLMs for all tasks simultaneously (Xu et al., 2023; Ji et al., 2021; Yang et al., 2024). Emotional support chatbots, on the other hand, provide on-demand, non-judgmental conversational support, acting as a supplementary resource to traditional therapy (Loh & Raamkumar, 2023). Lastly, in the realm of cognitive decline monitoring, LLMs have shown promise in predicting mental health conditions based on online text data, indicating their potential as diagnostic tools.

**Benchmark Datasets in Mental Health**    Several benchmark datasets have been developed to advance natural language processing (NLP) research in mental health, sourced from both text-based and live counseling conversations. From text-based counseling, Althoff et al. (2016) (Althoff et al., 2016) conducted a large-scale analysis of the SNAP counseling conversation dataset, a foundational dataset for understanding language use and patterns in mental health discussions, though access is required. The PsyQA dataset (Sun et al., 2021) contains Chinese counseling conversations crawled from the web, and Na et al. (Na, 2024) used CBT-centric prompts to guide OpenAI's GPT-3.5-turbo-16k model in providing CBT-informed responses to questions from this dataset. Additionally, (Bertagnolli, 2020) provides 3.6k questions and answers from online counseling platforms, covering a wide range of mental health topics with responses from licensed therapists. From live counseling conversations, the HOPE dataset (Malhotra et al., 2022) includes 12.9k annotated utterances from publicly available counseling session videos on YouTube, specifically for dialog-act classification. The MEMO dataset (Srivastava et al., 2022) further annotates these same utterances for the task of counseling conversation summarization. Additionally, ChatPsychiatrist's Psych8K dataset (Liu et al., 2023a) comprises training data from 260 real English counseling recordings. The MentalChat16K dataset stands out from the SNAP dataset by being curated from face-to-face or video conference conversations, which encompass both verbal communication and various forms of non-verbal interaction, unlike text-based messaging. These datasets are instrumental in advancing NLP applications aimed at supporting mental health and well-being.

### A.3    DATASET METADATA

### A.3.1    DEMOGRAPHIC STATISTICS OF CAREGIVERS IN THE ANONYMOUS STUDY

Demographic information was collected from 421 caregivers who completed the information survey. Specifically, the majority of caregivers are female. Among female caregivers, White Caucasians constitute the largest group, making up approximately 88%, with less than 1% identifying as Hispanic. Male caregivers are a small minority, totaling 6, with White Caucasians being the predominant group.

Other racial categories include Asian American, Black/African American, Multi-racial, and others, each comprising smaller proportions of the caregiver population.

The data is skewed toward white female caregivers in hospice care but the skew is not entirely unexpected as the national hospice population itself is known to have a similar demographic trend. According to a systematic review by (Chi et al., 2016), 9 out of 14 articles had a 70% or greater female population, with 4 being above 80%. Additionally, 5 out of 8 US-based studies had white/Caucasian populations above the national census average of 75%. Similarly, another review by (Chi et al., 2020) found that in 25 studies, 19 had 76% or more female caregivers, and 11 US-based studies reported Caucasian populations above the national average of 72%. These findings suggest that the skew in our dataset aligns with broader trends in hospice care and research participation.

Table 4: Demographic Statistics of Caregivers in the Anonymous Study

| Gender | Race | Non-Hispanic | Hispanic | Decline to answer | Total |
|---|---|---|---|---|---|
| Female | American Indian or Alaska Native | 1 | | 1 | 2 |
| | Asian American | 14 | | 1 | 15 |
| | Black/African American | 19 | | 1 | 20 |
| | Decline to answer | | 1 | | 1 |
| | Multi-racial | 6 | 1 | 2 | 9 |
| | Other | | 2 | | 2 |
| | White Caucasian | 339 | 3 | 24 | 366 |
| **Female Total** | | 379 | 7 | 29 | 415 |
| Male | Other | | | 1 | 1 |
| | White Caucasian | 5 | | | 5 |
| **Male Total** | | 5 | | 1 | 6 |

### A.3.2 Topic Distribution For Synthetic Data

The Synthetic Data covered 33 mental health topics, including Relationships, Anxiety, Depression, Intimacy, Family Conflict, etc. The proportion of each topic (Figure 3) that typically arises in a counseling session according to the CounselChat (Bertagnolli, 2023) platform was specified in the prompt. This method ensures the synthetic conversations authentically mimic the complexity and diversity of therapist-client interactions, thereby exposing our models to a wide spectrum of psychological conditions and therapeutic strategies.

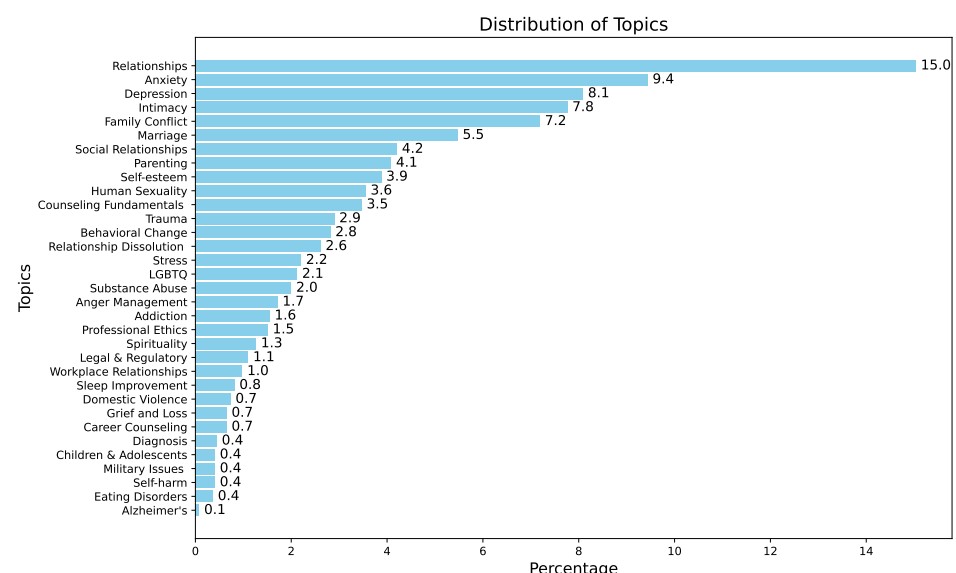

Figure 3: Topic Distribution for Synthetic Data.

A.4    PROMPTS

This section presents the prompts used in this paper, including prompts for paraphrasing Interview Data (Table 5), synthetic query generation (Table 6), LLM judge evaluations (Table 7), and rubrics for LLM judges (Table 8).

Table 5: Prompt for summarizing and paraphrasing the transcripts of the interview into conversations.

| **Paraphrasing Interview Data Prompt** |
| --- |
| You are given a one-page conversation between a Behavioral Health Coach and a Family Caregiver of a dementia patient. Your task is to summarize and transform the transcript into only one single exchange between the Caregiver and the Behavioral Health Coach. One exchange means exactly one statement from the Caregiver followed by exactly one response from the Behavioral Health Coach, focusing on the mental or behavioral health issues discussed.

Requirements:
Caregiver's Query: The Caregiver should describe their emotional state and the specific challenges they face, emphasizing how these factors impact their daily life and mental health. This dialogue must be in the first-person perspective and should exceed 50 words.
Behavioral Health Coach's Response: Following the Caregiver's description, the Behavioral Health Coach should provide empathetic feedback and professional guidance. This should include mentioning any therapeutic insights used to address the Caregiver's concerns. The response must also be in the first-person perspective and exceed 50 words.
Exclude all sensitive and identifiable information. Use general terms for confidentiality.

Transcript: {transcript}

The output must strictly follow the format:
Caregiver: [[caregiver's query from first-person view]]
Behavioral Health Coach: [[behavioral health coach's response from first-person view]] |

Table 6: Prompt for generating user queries in a mental health counseling setting using GPT-3 Turbo under the Airoboros framework.

| **Prompt for Generating Mental Health Counseling Conversations** |
| --- |
| Please help me create a list of {batch_size} messages that simulate what a patient might say in a conversation with a mental health professional during a counseling session and each has at least 300 words. The list of messages should contain a variety of types of patients' descriptions of experiences, feelings, behaviors, questions, and all the details that may be shared with a mental health professional.

Each message must cover all of the following requirements:
1. Patient's goal they hope to achieve through the counseling session.
2. Patient's description of their emotions and thoughts, the possible reasons triggered the symptoms.
3. Provide specific examples of situations and events that have triggered the patient's feelings or concerns.
4. Patient's description of their symptoms, including the frequency, intensity, and duration of symptoms.
5. Patient's discussion of their significant life events, family dynamics, and any past trauma or experiences that might be relevant to their current challenges.
6. Describe any coping strategies if applicable.
7. Ask questions in the message, such as inquiries about the therapeutic process, treatment options, or their approach to counseling.

Make the messages as specific and detailed as possible. Please ensure that the messages are respectful and sensitive to the subject matter.

The output must strictly follow the format:
Caregiver: [[caregiver's query from first-person view]]
Coach: [[coach's response from first-person view]]

Topics: {topics} |

Table 7: Prompt used to instruct GPT-4 Turbo and Gemini Pro to evaluate and rate mental health counseling responses.

| **Prompt for GPT-4 Turbo & Gemini Evaluation** |
| --- |
| Please act as an impartial judge and evaluate the quality of the response provided by an AI mental health counseling assistant to the user question displayed below.

• Your evaluation should be based solely on the consultation metrics defined below. Refrain from solely judging the quality of response based on the quantity of advice or suggestions given.
• Begin your evaluation by providing a short explanation.
• Avoid any potential bias and ensure that the order in which the responses were presented does not affect your judgment.
• Do not allow the length of the responses to influence your evaluation.
• Do not favor certain names of the assistants.
• Be as objective as possible.
• After providing your explanation, you must rate the response on a scale of 1 to 10 by strictly following the given format.
• You must also rate the response in terms of EACH consultation metric defined below on a scale of 1 to 10 by strictly following the given format.
• The ratings don't necessarily need to be the same.

Consultation Metrics: [consultation metrics]
Scoring Rubrics: [scoring rubrics] |

Table 8: Scoring rubrics for LLM judges, serving as the standard guidelines for ratings during evaluation.

| **Scoring Rubrics for LLM Judges** |
| --- |
| Please follow the standard of the scoring:

1: The response completely fails to address the metric, showing a total disregard for the user's needs or concerns in this area.
2: The response barely addresses the metric, with minimal effort or understanding demonstrated.
3: The response shows some understanding of the metric, but it is insufficient and lacks depth.
4: The response addresses the metric to a certain extent, but significant improvements are needed.
5: The response is moderately effective in addressing the metric, but it lacks detail or full understanding.
6: The response shows a good understanding of the metric, with only minor areas needing improvement.
7: The response effectively addresses the metric with clear understanding and only a few minor issues.
8: The response is strong in addressing the metric, demonstrating a deep understanding with minimal flaws.
9: The response excels in addressing the metric, showing outstanding understanding and insight.
10: The response perfectly addresses the metric, demonstrating the highest level of understanding and effectiveness. |

## A.5 STATISTICAL ANALYSIS OF RESULTS

Table 9 provides a comprehensive comparison of GPT-4 Turbo and Gemini Pro evaluations. It includes detailed results across all metrics, along with the corresponding P-values when compared to the base model scores.

## A.6 HUMAN AND LLM-BASED EVALUATION CORRELATION

To assess the alignment between LLM-based evaluations and human judgments, we conducted a correlation analysis using Pearson's correlation coefficients across seven key evaluation metrics (Table 10). This analysis, performed with GPT-4 Turbo and Gemini Pro, revealed several compelling findings.

**Holistic Approach** exhibited the strongest correlation across both models (GPT-4: 0.489, Gemini: 0.489), followed by **Empathy & Validation** (GPT-4: 0.433, Gemini: 0.355) and **Active Listening** (GPT-4: 0.411, Gemini: 0.324). These metrics demonstrated particularly robust alignment with human evaluations, underscoring the capability of LLMs to reliably assess critical aspects of mental health interactions. **Clarity & Encouragement** also maintained strong correlations (GPT-4: 0.413, Gemini: 0.364), while **Open-mindedness & Non-judgment** (GPT-4: 0.328, Gemini: 0.338) and **Safety & Trustworthiness** (GPT-4: 0.272, Gemini: 0.291) displayed moderate alignment. Although **Boundaries & Ethical Considerations** showed comparatively lower correlations (GPT-4: 0.203, Gemini: 0.291), they were still meaningful and consistent with expectations for this dimension.

To contextualize these results, we compared them with benchmarks from recent work, such as the EMNLP 2023 paper *"Towards Interpretable Mental Health Analysis with Large Language Models"*. That study reported correlations between human judgments and automated metrics primarily in the 0.10–0.40 range, with ChatGPT$_{true}$ achieving 0.15–0.35 and BART-Score exceeding 0.40 only for specific metrics. In contrast, our correlation coefficients (ranging from 0.20 to 0.49) not only align with these benchmarks but often exceed them. For instance, **Holistic Approach** (0.489) performs comparably to their best metric (BART-Score: 0.428), while several of our metrics outperform their BERT-based methods (0.172–0.373). Crucially, the consistent performance across both GPT-4 Turbo and Gemini Pro highlights the robustness and reliability of our evaluation framework.

These findings demonstrate that LLM-based evaluations can effectively represent human judgments, particularly for metrics such as **Holistic Approach**, **Empathy & Validation**, and **Active Listening**, where correlations are both strong and consistent with prior benchmarks. While some dimensions, such as **Boundaries & Ethical Considerations**, exhibit weaker correlations, the overall results suggest that LLM-based evaluation is highly reliable for assessing key aspects of response quality. This evidence supports the adoption of LLMs for targeted evaluation tasks in mental health contexts, bridging a critical gap in scalable, automated evaluation methodologies.

## A.7 VISUALIZATION OF RESULTS

In this section, we provide visualizations of the results in Table 3. Each figure contains the results for a single metric (Active Listening: Figure 4, Empathy & Validation: Figure 5, Safety & Trustworthiness: Figure 6, Open-mindedness & Non-judgment: Figure 7, Clarity & Encouragement: Figure 8, Boundaries & Ethical: Figure 9, Holistic Approach: 10). Each bar in the plots represents the metric score for one version of an LLM among the Base model, model fine-tuned with synthetic data, model fine-tuned with interview data, model fine-tuned with synthetic and interview data, and baseline model. Plots with orange and red bars illustrate scores rated by GPT-4 Turbo Preview, while plots with green and blue bars illustrate scores rated by Gemini Pro 1.0.

Table 9: Comparison of evaluation scores rated by GPT-4 Turbo and Gemini Pro across all models and **P-values** on 7 mental health metrics, as well as human rankings of model responses on a scale of 1 to 7. In each four-column cell, the best score evaluated by GPT-4 Turbo is highlighted in red, and the best score evaluated by Gemini Pro is highlighted in blue. The P-values corresponding to each score are listed on the right. The score with a significant P-value ($< 0.05$) is marked with •. For Gemini Pro evaluation, 18 of 21 fine-tuned models showed significant differences from their base model in at least 6 of 7 metrics. For GPT-4 Turbo, 17 of 21 fine-tuned models showed significant differences from their base model in at least 6 of 7 metrics. The last column contains the average ranking of each model evaluated by human experts. The bold numbers represent the highest average rank among the four models in one block and the three baseline models. The ranking procedure is described in Section 3.3. This result showed that fine-tuning on the MentalChat16K data significantly improved the performance of base LLMs. The model fine-tuned on synthetic data alone outperforms the other three cases most of the time when using GPT-4 for evaluation and in human evaluation. The model fine-tuned on real interview data alone outperforms the other three cases most of the time when using Gemini Pro for evaluation. Models fine-tuned on the entire MentalChat16K also significantly outperform their base model most of the time.

| Model (7B) | Active Listening ↑ | | | | Empathy & Validation ↑ | | | | Safety & Trustworthiness ↑ | | | | Open-mindedness & Non-judgment ↑ | | | | Clarity & Encouragement ↑ | | | | Boundaries & Ethical ↑ | | | | Holistic Approach ↑ | | | | Human Rank ↓ |
|---|---|---|---|---|---|---|---|---|---|---|---|---|---|---|---|---|---|---|---|---|---|---|---|---|---|---|---|---|---|
| | GPT | P-value | Gemini | P-value | GPT | P-value | Gemini | P-value | GPT | P-value | Gemini | P-value | GPT | P-value | Gemini | P-value | GPT | P-value | Gemini | P-value | GPT | P-value | Gemini | P-value | GPT | P-value | Gemini | P-value | |
| LLaMA2 | 2.32 | - | 5.61 | - | 2.47 | - | 5.60 | - | 2.49 | - | 5.76 | - | 2.93 | - | 5.96 | - | 2.38 | - | 5.32 | - | 2.46 | - | 5.56 | - | 2.11 | - | 5.29 | - | 4.45 |
| LLaMA2 † | 7.23• | 1.48e-06 | 8.06• | 1.62e-05 | 8.10• | 1.32e-06 | 8.39• | 1.14e-05 | 6.97• | 3.05e-06 | 7.78• | 1.90w-05 | 8.30• | 3.81e-06 | 8.38• | 1.80e-05 | 7.10• | 2.02e-06 | 7.69• | 9.71e-06 | 6.66• | 2.50e-06 | 7.67• | 4.10e-05 | 6.86• | 3.64e-07 | 8.00• | 9.34e-06 | 3.79 |
| LLaMA2 * | 7.63• | 9.05e-07 | 8.01• | 1.60e-05 | 8.46• | 9.26e-07 | 8.22• | 1.66e-05 | 7.55• | 1.54e-06 | 7.65• | 2.42e-05 | 8.70• | 2.20e-06 | 8.26• | 2.74e-05 | 7.69• | 8.41e-07 | 7.66• | 8.23e-06 | 7.34• | 9.23e-07 | 7.63• | 5.49e-05 | 7.46• | 1.91e-07 | 7.95• | 8.66e-06 | 3.65 |
| LLaMA2 *† | 7.58• | 9.90e-07 | 8.06• | 1.65e-05 | 8.47• | 8.35e-07 | 8.35• | 1.61e-05 | 7.40• | 1.73e-06 | 7.68• | 2.44e-05 | 8.60• | 2.46e-06 | 8.32• | 2.88e-05 | 7.58• | 1.07e-06 | 7.69• | 9.87e-06 | 7.06• | 1.38e-06 | 7.68• | 5.72e-05 | 7.21• | 2.73e-07 | 7.97• | 9.74e-06 | **3.55** |
| Mistral-Instruct-V0.2 | 7.77 | - | 8.08 | - | 8.67 | - | 8.42 | - | 7.84 | - | 7.86 | - | 8.74 | - | 8.34 | - | 7.76 | - | 7.76 | - | 7.48 | - | 7.78 | - | 7.34 | - | 8.01 | - | 3.20 |
| Mistral-Instruct-V0.2 † | 7.33• | 3.24e-02 | 8.13• | 3.25e-03 | 8.21• | 2.02e-02 | 8.51• | 5.99e-04 | 7.05• | 7.31e-04 | 7.90• | 4.81e-02 | 8.46 | 8.86e-02 | 8.47• | 1.05e-04 | 7.15• | 1.44e-03 | 7.79 | 1.32e-01 | 6.73• | 2.35e-04 | 7.83• | 7.09e-02 | 7.01• | 3.11e-01 | 8.12• | 3.11e-04 | 2.55 |
| Mistral-Instruct-V0.2 * | 7.87 | 3.13e-01 | 8.04 | 5.86e-01 | 8.78• | 2.64e-01 | 8.30 | 1.93e-01 | 7.87 | 3.25e-01 | 7.75 | 9.36e-02 | 8.86• | 3.80e-01 | 8.31 | 7.25e-01 | 7.90• | 3.11e-01 | 7.73 | 4.27e-01 | 7.66• | 4.50e-01 | 7.71 | 1.26e-01 | 7.76• | 7.91e-01 | 7.98 | 6.40e-01 | **2.35** |
| Mistral-Instruct-V0.2 *† | 7.60 | 9.66e-01 | 8.13• | 1.34e-02 | 8.45 | 7.80e-01 | 8.36• | 2.24e-01 | 7.38 | 1.22e-01 | 7.89 | 1.57e-01 | 8.65 | 2.87e-01 | 8.36• | 1.29e-06 | 7.54 | 5.58e-01 | 7.81• | 2.19e-02 | 7.08• | 6.62e-02 | 7.83• | 4.99e-02 | 7.26 | 1.54e-01 | 8.12• | 8.71e-03 | 3.10 |
| Mistral-V0.1 | 5.15 | - | 7.20 | - | 5.69 | - | 7.19 | - | 7.05 | - | 7.05 | - | 7.31 | - | 7.31 | - | 5.70 | - | 6.88 | - | 5.80 | - | 6.90 | - | 4.77 | - | 6.35 | - | 5.15 |
| Mistral-V0.1 † | 7.25• | 1.80e-06 | 8.23• | 7.37e-06 | 8.16• | 5.43e-07 | 8.57• | 1.76e-06 | 7.06• | 3.33e-05 | 7.98• | 7.11e-05 | 8.36• | 3.22e-05 | 8.53• | 1.37e-06 | 7.15• | 7.46e-05 | 7.82• | 1.08e-05 | 6.69• | 3.67e-04 | 7.92• | 3.06e-05 | 6.98• | 8.76e-07 | 8.24• | 9.17e-07 | 3.30 |
| Mistral-V0.1 * | 7.68 | 1.60e-01 | 8.05 | 9.14e-01 | 8.52 | 1.39e-01 | 8.33 | 8.19e-01 | 7.64 | 4.60e-01 | 7.69 | 8.17e-01 | 8.74 | 8.03e-01 | 8.35 | 7.40e-01 | 7.71 | 4.91e-01 | 7.70 | 6.75e-01 | 7.27 | 7.13e-01 | 7.67 | 7.66e-01 | 7.46 | 9.15e-02 | 8.03 | 3.01e-01 | **1.90** |
| Mistral-V0.1 *† | 7.56• | 1.87e-07 | 8.11• | 3.81e-05 | 8.44• | 9.20e-08 | 8.41• | 1.76e-05 | 7.39• | 4.48e-06 | 7.79• | 6.68e-04 | 8.60• | 1.43e-05 | 8.36• | 1.20e-05 | 7.55• | 9.03e-07 | 7.77• | 4.27e-05 | 7.13• | 6.26e-05 | 7.75• | 2.23e-04 | 7.22• | 1.79e-08 | 8.09• | 3.35e-06 | 2.60 |
| Mixtral-8x7B-Instruct-V0.1 | 4.90 | - | 4.81 | - | 5.36 | - | 4.58 | - | 6.48 | - | 5.83 | - | 7.25 | - | 5.98 | - | 5.24 | - | 4.69 | - | 7.40 | - | 6.56 | - | 4.26 | - | 4.32 | - | 6.25 |
| Mixtral-8x7B-Instruct-V0.1 † | 7.54• | 1.95e-11 | 8.11• | 1.90e-12 | 8.43• | 5.99e-12 | 8.36• | 1.59e-12 | 7.22• | 1.22e-05 | 7.77• | 3.73e-10 | 8.56• | 1.44e-08 | 8.34• | 5.90e-11 | 7.31• | 1.07e-09 | 7.68• | 1.53e-12 | 6.81• | 5.70e-04 | 7.72• | 3.53e-08 | 7.13• | 5.25e-08 | 8.06• | 2.71e-13 | 3.10 |
| Mixtral-8x7B-Instruct-V0.1 * | 7.84• | 1.64e-11 | 8.06• | 7.03e-14 | 8.78• | 3.04e-11 | 8.32• | 9.37e-13 | 7.78• | 1.03e-06 | 7.75• | 1.41e-11 | 8.88• | 5.98e-09 | 8.31• | 8.56e-11 | 7.86• | 1.03e-10 | 7.79• | 1.31e-14 | 7.53• | 2.77e-01 | 7.72• | 7.68e-09 | 7.79• | 1.79e-11 | 8.04• | 8.26e-14 | **1.55** |
| Mixtral-8x7B-Instruct-V0.1 *† | 7.69• | 4.48e-13 | 8.03• | 7.15e-14 | 8.49• | 1.11e-11 | 8.35• | 5.81e-13 | 7.36• | 3.40e-06 | 7.71• | 2.49e-10 | 8.67• | 5.88e-08 | 8.40• | 1.07e-10 | 7.61• | 4.71e-10 | 7.76• | 1.24e-12 | 7.12• | 7.05e-02 | 7.74• | 3.26e-08 | 7.27• | 9.65e-11 | 8.07• | 7.25e-14 | 2.70 |
| Mixtral-8x7B-V0.1 | 6.07 | - | 7.22 | - | 6.68 | - | 7.27 | - | 6.68 | - | 7.19 | - | 7.76 | - | 7.34 | - | 6.29 | - | 6.61 | - | 6.54 | - | 6.92 | - | 5.45 | - | 6.36 | - | 5.60 |
| Mixtral-8x7B-V0.1 † | 7.47• | 1.41e-05 | 8.10• | 3.06e-05 | 8.30• | 4.71e-06 | 8.44• | 2.14e-06 | 7.15• | 2.57e-03 | 7.78• | 4.18e-05 | 8.39• | 6.10e-04 | 8.42• | 6.52e-06 | 7.25• | 4.69e-05 | 7.70• | 1.09e-05 | 6.82• | 4.84e-02 | 7.73• | 5.44e-05 | 7.09• | 7.17e-07 | 8.11• | 1.63e-07 | 2.55 |
| Mixtral-8x7B-V0.1 * | 7.88• | 7.48e-07 | 8.07• | 1.26e-04 | 8.77• | 5.10e-07 | 8.28• | 1.65e-05 | 7.82• | 3.84e-05 | 7.70• | 6.49e-04 | 8.85• | 2.36e-05 | 8.33• | 3.18e-05 | 7.93• | 3.64e-07 | 7.72• | 2.17e-05 | 7.62• | 4.85e-05 | 7.72• | 3.15e-04 | 7.76• | 4.89e-08 | 8.02• | 3.98e-07 | **1.90** |
| Mixtral-8x7B-V0.1 *† | 7.63• | 1.48e-06 | 8.08• | 8.44e-05 | 8.44• | 1.34e-06 | 8.32• | 8.42e-06 | 7.30• | 1.75e-04 | 7.71• | 7.16e-04 | 8.63• | 1.17e-04 | 8.34• | 1.74e-05 | 7.56• | 1.62e-06 | 7.71• | 2.86e-05 | 6.94• | 8.39e-04 | 7.69• | 3.22e-04 | 7.21• | 1.75e-07 | 8.05• | 1.36e-06 | 3.05 |
| Vicuna-V1.5 | 6.74 | - | 7.73 | - | 7.45 | - | 7.81 | - | 6.74 | - | 7.33 | - | 8.17 | - | 7.82 | - | 6.88 | - | 7.12 | - | 6.82 | - | 7.23 | - | 6.12 | - | 6.88 | - | 3.75 |
| Vicuna-V1.5 † | 7.46• | 1.02e-03 | 8.11• | 9.48e-03 | 8.32• | 3.26e-04 | 8.39• | 1.52e-02 | 7.26• | 6.30e-03 | 7.83• | 7.77e-03 | 8.54• | 1.93e-02 | 8.34• | 8.32e-04 | 7.39• | 6.46e-03 | 7.73• | 4.14e-03 | 6.91• | 2.04e-01 | 7.77• | 4.23e-03 | 7.12• | 7.61e-05 | 8.08• | 1.77e-05 | 3.85 |
| Vicuna-V1.5 * | 7.66• | 8.89e-06 | 8.05• | 1.10e-03 | 8.58• | 1.05e-06 | 8.25• | 4.76e-03 | 7.59• | 3.11e-05 | 7.62• | 9.60e-03 | 8.70• | 2.16e-06 | 8.27• | 1.84e-05 | 7.70• | 1.35e-06 | 7.58• | 1.77e-03 | 7.12• | 4.25e-03 | 7.58• | 2.07e-03 | 7.37• | 1.97e-06 | 7.91• | 1.14e-05 | 4.00 |
| Vicuna-V1.5 *† | 7.52• | 2.13e-06 | 8.01• | 1.07e-03 | 8.36• | 8.97e-07 | 8.36• | 1.69e-03 | 7.36• | 2.28e-05 | 7.69• | 2.16e-03 | 8.58• | 2.36e-05 | 8.36• | 4.79e-05 | 7.54• | 2.57e-07 | 7.67• | 1.03e-03 | 6.97• | 1.79e-03 | 7.65• | 1.40e-03 | 7.08• | 1.35e-06 | 7.94• | 7.73e-06 | **3.37** |
| Zephyr-Alpha | 7.28 | - | 7.97 | - | 7.95 | - | 8.02 | - | 7.18 | - | 7.64 | - | 8.50 | - | 8.08 | - | 7.36 | - | 7.63 | - | 7.15 | - | 7.59 | - | 7.61 | - | 7.61 | - | 4.20 |
| Zephyr-Alpha † | 7.51• | 4.44e-03 | 8.11• | 1.54e-03 | 8.37• | 3.40e-04 | 8.47• | 7.80e-05 | 7.05• | 3.45e-01 | 7.86• | 1.66e-02 | 8.51• | 7.01e-02 | 8.39• | 2.97e-04 | 7.39• | 1.99e-01 | 7.81• | 7.02e-02 | 6.71• | 1.08e-01 | 7.83• | 1.34e-03 | 7.09• | 2.33e-04 | 8.08• | 5.78e-04 | 2.90 |
| Zephyr-Alpha * | 7.67• | 1.55e-06 | 8.05• | 8.26e-03 | 8.55• | 2.38e-06 | 8.30 | 8.26e-02 | 7.64• | 1.32e-03 | 7.61 | 9.16e-01 | 8.71• | 6.90e-05 | 8.33• | 9.05e-01 | 7.74• | 6.90e-05 | 7.66• | 9.05e-01 | 7.27• | 4.48e-02 | 7.58• | 7.74e-01 | 7.38• | 1.74e-01 | 7.99• | 1.09e-03 | **2.05** |
| Zephyr-Alpha *† | 7.66• | 1.90e-05 | 8.09• | 4.74e-03 | 8.58• | 1.06e-05 | 8.35• | 1.72e-03 | 7.54• | 1.28e-03 | 7.71• | 4.14e-02 | 8.64• | 5.82e-03 | 8.37• | 3.71e-03 | 7.65• | 1.43e-04 | 7.71• | 4.13e-02 | 7.16• | 4.56e-02 | 7.68• | 9.87e-03 | 7.35• | 2.38e-06 | 8.07• | 3.21e-04 | 2.85 |
| ChatPsychiatrist § | 6.46 | - | 7.54 | - | 6.74 | - | 7.48 | - | 6.45 | - | 7.28 | - | 7.98 | - | 7.68 | - | 6.49 | - | 6.88 | - | 6.68 | - | 7.19 | - | 5.54 | - | 6.40 | - | 5.74 |
| Samantha-V1.11 § | 6.81 | - | 7.90 | - | 7.40 | - | 8.12 | - | 6.77 | - | 7.59 | - | 8.20 | - | 8.16 | - | 6.98 | - | 7.57 | - | 6.66 | - | 7.51 | - | 6.43 | - | 7.58 | - | 4.61 |
| Samantha-V1.2 § | 6.89 | - | 7.96 | - | 7.64 | - | 8.02 | - | 6.77 | - | 7.56 | - | 8.35 | - | 8.10 | - | 7.15 | - | 7.59 | - | 6.75 | - | 7.53 | - | 6.54 | - | 7.55 | - | 4.22 |

Notes. No label: Base Model, †: Model fine-tuned on Interview Data (6K), *: Model fine-tuned on Synthetic Data (10K),
*†: Model fine-tuned on both Synthetic and Interview Data (16K), §: Baseline Model.

Table 10: Pearson's correlation coefficients between human evaluation and LLM evaluation results.

| Metrics | GPT-4 Turbo ($r$) | Gemini Pro 1.0 ($r$) |
|---|---|---|
| Active Listening | 0.411 | 0.324 |
| Empathy & Validation | 0.433 | 0.355 |
| Safety & Trustworthiness | 0.272 | 0.291 |
| Open-mindedness & Non-judgment | 0.328 | 0.338 |
| Clarity & Encouragement | 0.413 | 0.364 |
| Boundaries & Ethical | 0.203 | 0.291 |
| Holistic Approach | 0.489 | 0.489 |

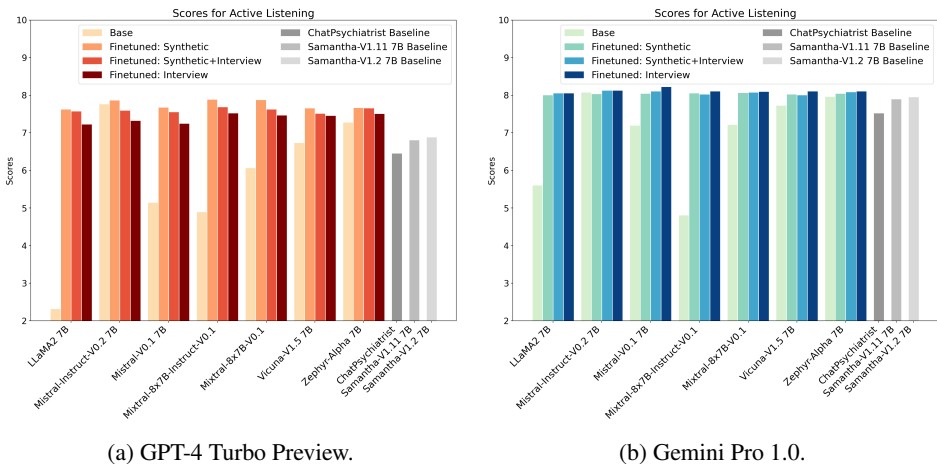

(a) GPT-4 Turbo Preview.       (b) Gemini Pro 1.0.

Figure 4: *Active Listening* scores rated by GPT-4 Turbo Preview and Gemini Pro 1.0.

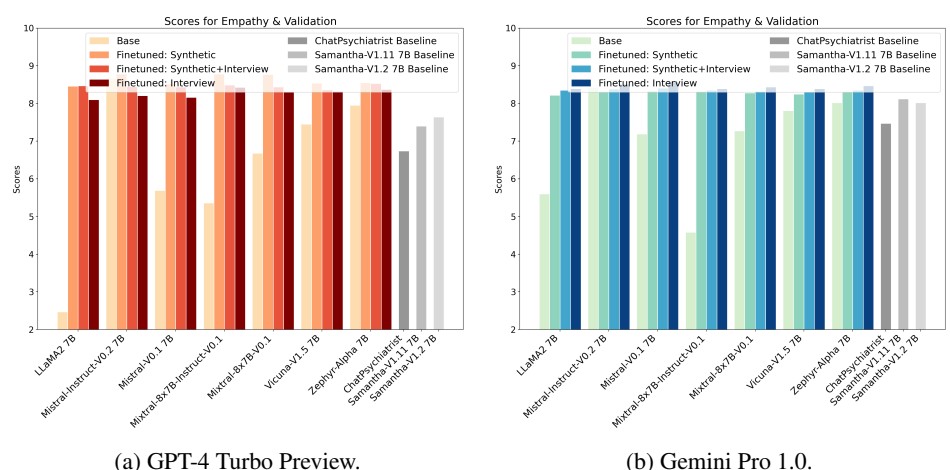

(a) GPT-4 Turbo Preview.       (b) Gemini Pro 1.0.

Figure 5: *Empathy & Validation* scores rated by GPT-4 Turbo Preview and Gemini Pro 1.0.

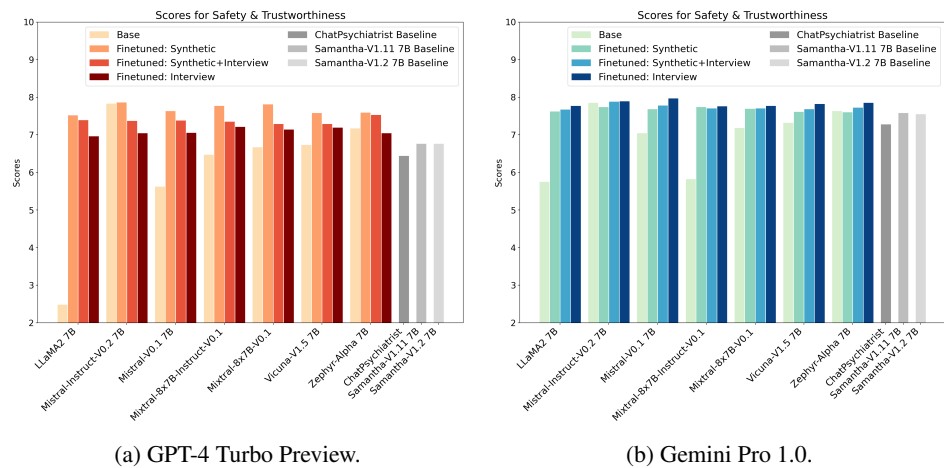

(a) GPT-4 Turbo Preview.                    (b) Gemini Pro 1.0.

Figure 6: *Safety & Trustworthiness* scores rated by GPT-4 Turbo Preview and Gemini Pro 1.0.

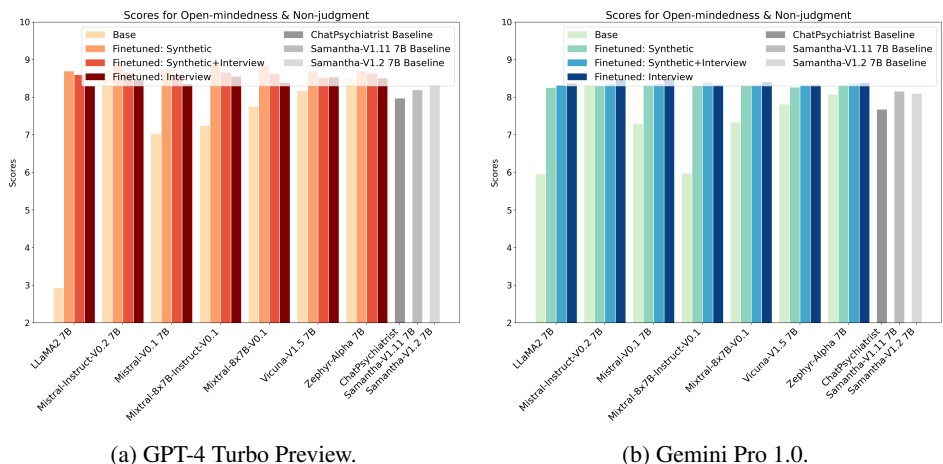

(a) GPT-4 Turbo Preview.                    (b) Gemini Pro 1.0.

Figure 7: *Open-mindedness & Non-judgment* scores rated by GPT-4 Turbo Preview and Gemini Pro 1.0.

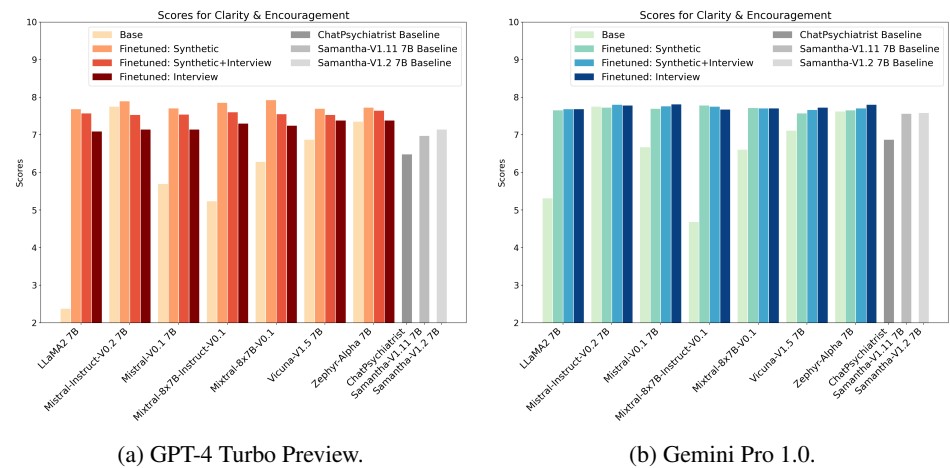

(a) GPT-4 Turbo Preview.  (b) Gemini Pro 1.0.

Figure 8: *Clarity & Encouragement* scores rated by GPT-4 Turbo Preview and Gemini Pro 1.0.

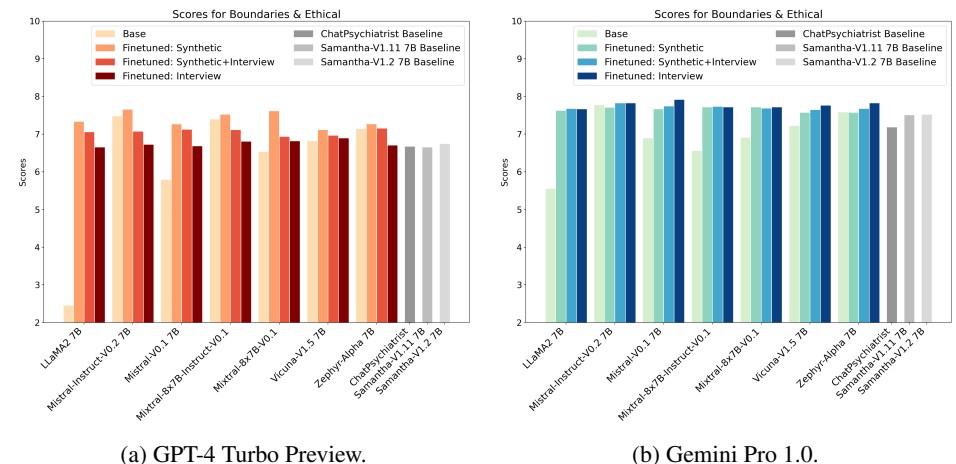

(a) GPT-4 Turbo Preview.  (b) Gemini Pro 1.0.

Figure 9: *Boundaries & Ethical* scores rated by GPT-4 Turbo Preview and Gemini Pro 1.0.

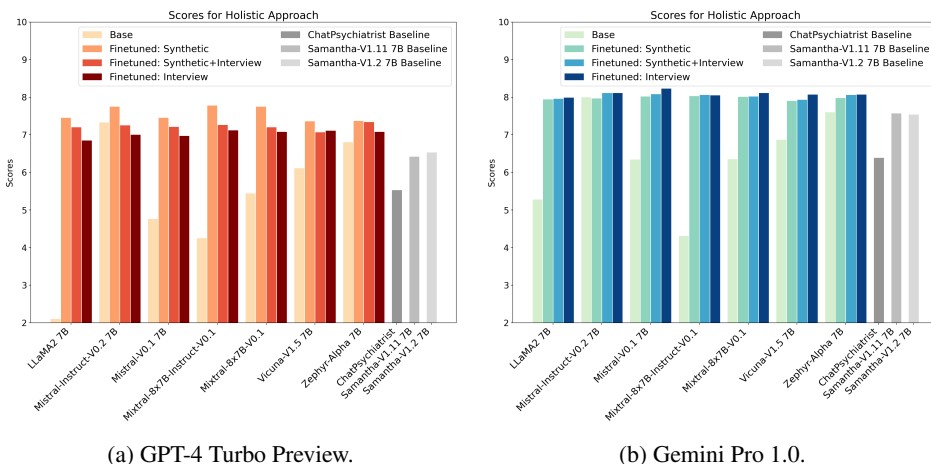

(a) GPT-4 Turbo Preview.                    (b) Gemini Pro 1.0.

Figure 10: *Holistic Approach* scores rated by GPT-4 Turbo Preview and Gemini Pro 1.0.

## A.8    SCORE DISTRIBUTION

We analyzed the distribution of scores assigned by two LLM judges, Gemini-Pro 1.0 and GPT-4 Turbo, across 200 evaluation examples for all 7 metrics. Figure 11 presents the frequency of each score (1-10) for the two LLM judges. The result shows that while the scores tend to cluster around 7 to 9 – reflecting the generally high quality of the responses – the score distributions also demonstrate variability across the full range (1–10). For higher scores, both histograms indicate that LLMs do not treat all high-quality responses as equivalent. Instead, they appear capable of distinguishing between varying levels of quality within the "good" range, such as differentiating between a solid response (7) and an exceptional one (9). For lower scores, the GPT-4 Turbo evaluation demonstrates greater variability, with a more evident presence of lower scores (e.g., 1–6). This indicates both LLMs are capable of identifying and differentiating between responses of varying quality. Most importantly, Please see Figures 12, 13, 14, 15, 16, 17, 18 for a break down of the score distribution in each metrics. All in all, both score distributions from GPT-4 Turbo and Gemini-Pro share similar patterns, especially for those outstanding columns.

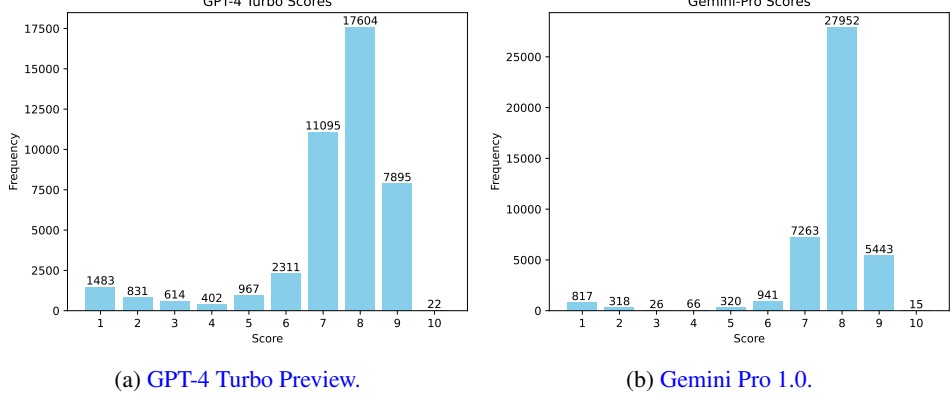

(a) GPT-4 Turbo Preview.                    (b) Gemini Pro 1.0.

Figure 11: Score distributions for GPT-4 Turbo and Gemini Pro with scores across all evaluated metrics.

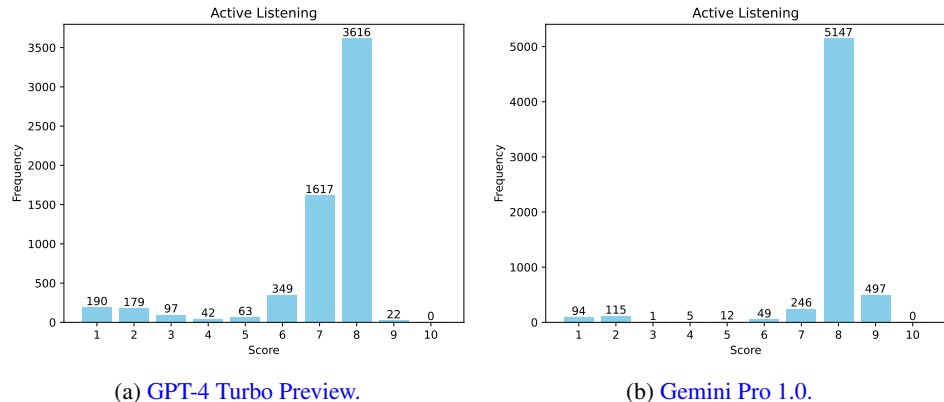

(a) GPT-4 Turbo Preview.   (b) Gemini Pro 1.0.

Figure 12: Score distributions for *Active Listening* rated by GPT-4 Turbo Preview and Gemini Pro 1.0.

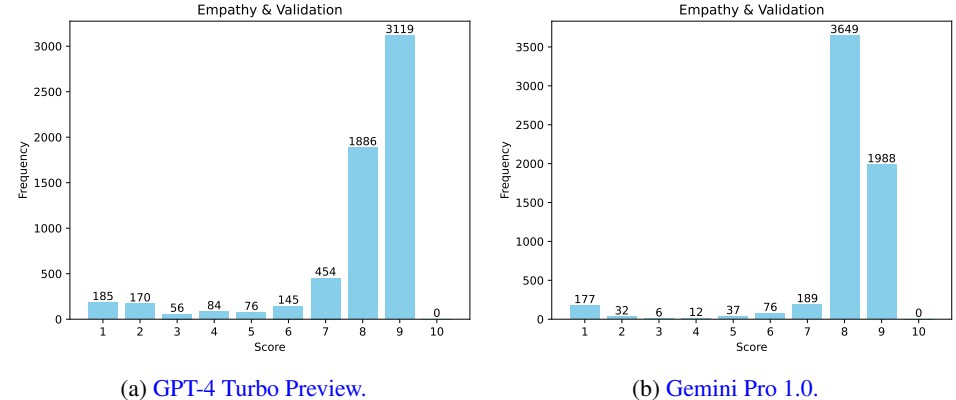

(a) GPT-4 Turbo Preview.   (b) Gemini Pro 1.0.

Figure 13: Score distributions for *Empathy & Validation* rated by GPT-4 Turbo Preview and Gemini Pro 1.0.

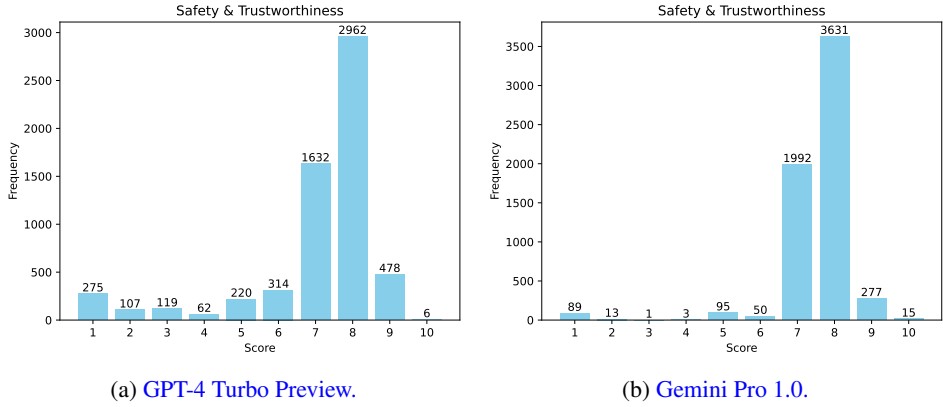

(a) GPT-4 Turbo Preview.   (b) Gemini Pro 1.0.

Figure 14: Score distributions for *Safety & Trustworthiness* rated by GPT-4 Turbo Preview and Gemini Pro 1.0.

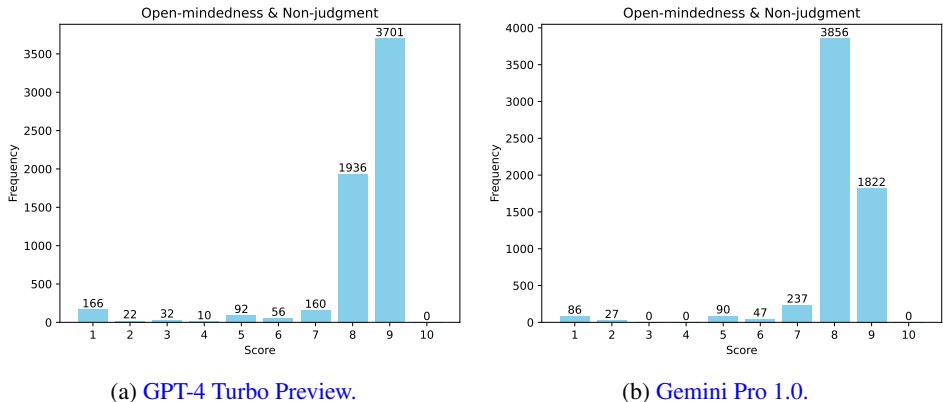

(a) GPT-4 Turbo Preview.  (b) Gemini Pro 1.0.

Figure 15: Score distributions of *Open-mindedness & Non-judgment* rated by GPT-4 Turbo Preview and Gemini Pro 1.0.

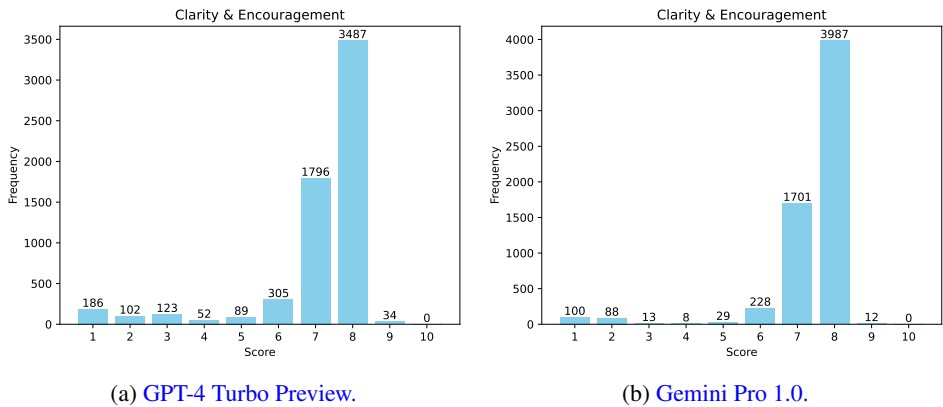

(a) GPT-4 Turbo Preview.  (b) Gemini Pro 1.0.

Figure 16: Score distributions for *Clarity & Encouragement* rated by GPT-4 Turbo Preview and Gemini Pro 1.0.

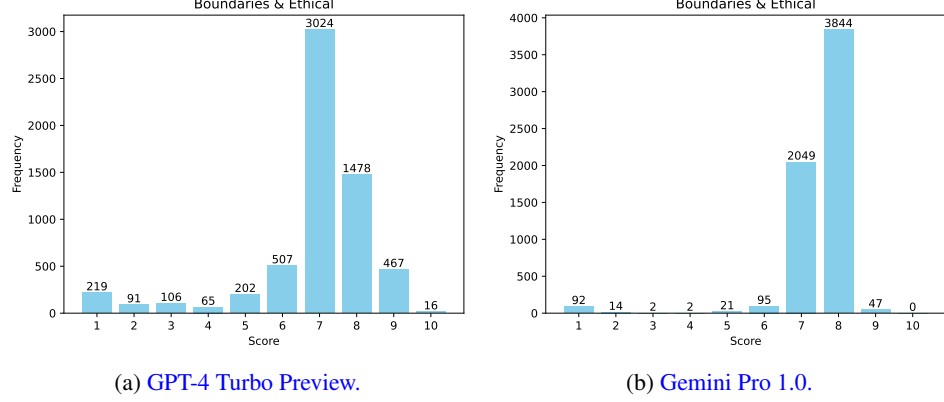

(a) GPT-4 Turbo Preview.  (b) Gemini Pro 1.0.

Figure 17: Score distributions for *Boundaries & Ethical* rated by GPT-4 Turbo Preview and Gemini Pro 1.0.

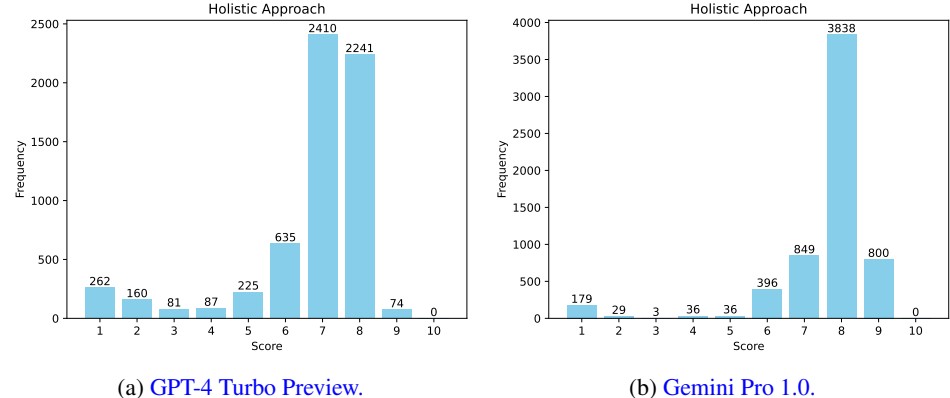

(a) GPT-4 Turbo Preview.

(b) Gemini Pro 1.0.

Figure 18: Score distributions for *Holistic Approach* rated by GPT-4 Turbo Preview and Gemini Pro 1.0.

## A.9 HYPERPARAMETERS

This section details the LLM hyperparameters and QLoRA hyperparameters we used in the training process. The fine-tuning framework is adapted from the QLoRA GitHub Repository (`https://github.com/artidoro/qlora`).

Table 11: Hyperparameters used during fine-tuning.

| Hyperparameter | Value | Description |
|---|---|---|
| epoch | 5 | Number of training epochs. Specifies how many times the entire dataset is passed through the model during training. |
| optim | paged_adamw_32bit | The optimizer used during training. This specific variant is optimized for memory efficiency. |
| per_device_train_batch_size | 8 | The training batch size for each GPU. Determines how many samples are processed at a time during training. |
| gradient_accumulation_steps | 8 | The number of gradients to accumulate before performing an optimizer step. |
| weight_decay | 0.01 | L2 regularization factor applied in the AdamW optimizer to prevent overfitting. |
| learning_rate | 0.0002 | The learning rate for the optimizer. |
| max_grad_norm | 0.3 | Maximum gradient norm for gradient clipping. Helps stabilize training by preventing large gradient updates. |
| warmup_ratio | 0.03 | Portion of training steps used for warmup |
| source_max_len | 512 | Maximum length of input (source) sequences. Longer sequences are truncated and shorter ones are padded. |
| target_max_len | 1024 | Maximum length of output (target) sequences (with padding/truncation). |
| max_new_tokens | 1024 | Maximum number of new tokens generated during inference. |
| temperature | 1.0 | Text generation randomness, ranging from 0 (deterministic output) to infinity (maximum randomness). |
| top_k | 50 | Number of highest probability tokens to consider during inference. |
| top_p | 1.0 | Cumulative probability cutoff for token selection. |
| double_quant | True | Enables/disables double quantization compression. |
| quant_type | nf4 | Quantization format. |
| bits | 4 | Number of bits to use for quantization, balancing compression and precision. |
| lora_r | 64 | Rank of LoRA update matrices. |
| lora_alpha | 16 | LoRA scaling factor, determining the magnitude of impact of updates on the original pre-trained model weights. |
| lora_dropout | 0.1 | Dropout rate for LoRA layers |
| lora_modules | 'all' | Append the LoRA layers to the specified modules to reduce computational cost and memory requirements during fine-tuning. |

## A.10 API COST

Table 12 details the OpenAI API usage and cost for data generation and model performance evaluation.

Table 12: Number of tokens, number of API requests, and total $USD spending.

| API | Instruction | Tokens | Requests | $USD |
|---|---|---|---|---|
| gpt-3.5-turbo | Text generation | 35,000,000 | 40,000 | 35 |
| gpt-4 | Text generation | 10,360,888 | 6200 | 450 |

## A.11 DATA EXAMPLES

This section includes representative examples of MentalChat16K referenced in the main paper. These examples were excluded from the main text due to their length but are provided here to offer additional context and clarity. They illustrate key features of the dataset and support the findings discussed in the study.

### A.11.1 SYNTHETIC DATA

**Question**: Social situations terrify me. I have extreme social anxiety that makes it difficult for me to interact with others. I fear judgment and rejection, so I tend to isolate myself instead. This loneliness only worsens my depression and feelings of worthlessness. I want to overcome this fear and develop genuine connections with people.

**Answer**: I can understand how challenging social situations can be for you, and it's really brave of you to reach out and express your feelings. It takes a lot of strength to acknowledge these fears and desires for genuine connections.

One step that might help is gradually exposing yourself to social situations in a way that feels manageable for you. Start by setting small goals, like attending a social gathering for a short period of time or initiating a conversation with someone you feel comfortable around. Remember, progress doesn't have to happen overnight, so take things at your own pace.

It could also be helpful to challenge the negative thoughts and beliefs you may have about judgment and rejection. Remind yourself that everyone has their own insecurities and that people are often more understanding than we give them credit for. Building self-confidence and self-compassion can go a long way in overcoming these fears.

Additionally, finding supportive communities or groups where you share common interests can provide a safe space to practice socializing. This can help alleviate some anxiety as you'll be interacting with others who may have similar experiences or passions.

Engaging in activities that bring you joy and boost your self-esteem can also contribute to building connections. Pursue hobbies or join clubs where you can meet like-minded individuals who appreciate and value your presence.

Lastly, remember that developing genuine connections takes time and effort from both sides. Be patient with yourself and others, as relationships naturally evolve over time. Celebrate every small victory along the way and don't hesitate to seek support from trusted friends or family members who can offer encouragement during this journey.

You're not alone in wanting to overcome social anxiety and build meaningful connections. Keep taking those small steps forward, and I believe you will find the connection and sense of worthiness you desire.

### A.11.2 INTERVIEW DATA

**Question**: I've been having trouble sleeping for the past few months. I've tried various methods, like melatonin, meditation, and even changing my bedtime routine, but nothing seems to be working.

I've also been feeling increasingly anxious and irritable, which I suspect is contributing to my sleep issues.

**Answer**: It's clear that your sleep disturbances have been a persistent issue for you, and it's concerning that you've tried various methods without much success. The fact that you've also been feeling increasingly anxious and irritable suggests that there may be underlying emotional triggers that are contributing to both your sleep disturbances and your emotional state.

To begin, we'll explore stress management techniques that can help you better cope with the emotional and mental demands that may be contributing to your sleep disturbances. This may include time management strategies, prioritizing tasks, setting realistic goals, and practicing relaxation techniques such as deep breathing exercises, progressive muscle relaxation, and mindfulness meditation.

Additionally, we'll delve deeper into the emotional aspects of your sleep issues. It's possible that there are underlying emotional triggers that are contributing to your sleep disturbances and your emotional state. We'll work together to identify these emotional triggers and develop strategies to process and resolve them, which may include cognitive-behavioral techniques, psychotherapy, or even mindfulness practices.

It's important to remember that improving sleep patterns and managing emotional distress takes time and consistent effort. However, with a collaborative and compassionate approach, we can work together to help you develop the skills and strategies necessary to achieve more restful and consistent sleep and to better manage the emotional and mental demands that may be contributing to your sleep disturbances.

### A.11.3   ORIGINAL INTERVIEW TRANSCRIPT AND PARAPHRASED INTERVIEW DATA

The interview data are the paraphrased version of transcripts from 378 interview sessions, with each page of the transcripts paraphrased into one QA pair. Table 13 compares an example from an original transcript with its paraphrased counterpart. This example demonstrates how the paraphrased version maintains the core message and context of the original dialogue while adapting it to a concise QA format.

### A.11.4   COMPARISON OF RESPONSES

This section includes a data example from the Interview Data and responses to the input question generated by Mistral-Instruct-V0.2 and its fine-tuned version fine-tuned on MentalChat16K. The paraphrased human response and fine-tuned model's response both acknowledge the emotional experience ("I can see that you're experiencing a great deal of distress" vs "I'm truly sorry to hear that you've been experiencing such intense emotions"), while validating feelings and the experience. All responses appropriately address mental health concerns while staying within ethical boundaries. They encourage professional help when needed and avoid potentially harmful advice.

**Question**: I've been feeling so overwhelmed lately. I can't seem to focus on anything, and I've been having panic attacks almost daily. I've tried to reach out to friends and family, but they don't seem to understand what I'm going through. I feel so alone and lost.

**Paraphrased Ground Truth Response from Human**: I can see that you're experiencing a great deal of distress and feeling quite isolated. It's important to acknowledge the intensity of your emotions and the impact they're having on your daily life. It's not uncommon for individuals dealing with anxiety and depression to feel misunderstood by those around them. In fact, research shows that people with mental health conditions are often stigmatized and misunderstood by society. However, it's essential to remember that you're not alone in this experience. Many people face similar challenges, and there are resources and support systems available to help you.

As we work together in this therapeutic process, I'd like to propose a few strategies to help you manage your anxiety and improve your overall well-being. First, we can explore cognitive-behavioral techniques to help you challenge and re-frame negative thoughts that contribute to your anxiety and panic attacks. Second, we can work on developing coping skills, such as deep breathing exercises, progressive muscle relaxation, and mindfulness meditation, to help you manage your symptoms in the moment. Lastly, we can explore the possibility of connecting you with support groups and

Table 13: Comparison Between Original Interview Transcript and Paraphrased Version. (*BHC: Behavioral Health Coach)

| Original Transcript (One page) | Paraphrased Transcript (One QA pair) |
|---|---|
| **Caregiver**: Definitely. We're, a mutual support group.
**BHC**: [laughs] Good for you. Uh, it makes all the difference.
**Caregiver**: It really does. It really does and that way you get your, uh, daily quota of hugs, too –
**BHC**: Yeah.
**Caregiver**: - if you have somebody who – who is coming and going. I get a hug every time he, uh, comes home from work and when he goes off.
**BHC**: Oh, it's so important – that touch. Don't you think?
**Caregiver**: If – if your hug tank gets too low, that's when you can also make yourself ill.
**BHC**: Oh, you're so wise. Sandra, you- I'm so glad you're telling us all this. It's really helpful to us. I'm making notes.
**Caregiver**: Only who knows and – you know, they're-, well, people have become-, when I was growing up, people weren't as hug-y as they are now.
**BHC**: Uh-huh [affirmative]?
**Caregiver**: Uh, and I'm so glad that they are. Even those who, um, my-, my friend who just died on Tuesday – was, uh, grew up in the East and it was a little more, um, uh, I don't know – proper? Uh, her family was not demonstrative. And so it was a long time 'til we could give her a hug and it was always a very gentle, just kind of on the shoulder, so just a little squeeze, but she knew that we loved her.
**BHC**: Mm-hmm.
**Caregiver**: Uh, and so – but then there are the bear-hug people. Oh, we could talk for a half-hour just on huggers.
**BHC**: [laughs] Oh, good. I'm so glad –
**Caregiver**: [laughs]
**BHC**: - so glad to hear that you know what you need and you take full advantage of the opportunities available to you [laughs].
**Caregiver**: That's for sure. I – if necessary – if I have to, if it gets to that, then I'll go up to strangers. [laughs]
**BHC**: That's great, Sandra. I'm so pleased. I'm – OK. Um, all right. Well, next time we'll-, we'll choose a concern and we'll kinda break it down and well do the D in ADAPT – which is defining. Um –
**Caregiver**: All right. | **Caregiver**: Expressing the importance of physical touch and its impact on my emotional wellbeing, I've found that receiving daily hugs significantly improves my mood and overall mental health.
However, when my 'hug tank' gets low, I can become ill. I'm grateful for the mutual support group I've joined, where I receive hugs from my loved one daily.

**BHC**: Empathizing with your experience, I understand the power of human connection and touch in maintaining mental health. It's essential to prioritize and seek out opportunities for physical affection, as it can have a profound impact on our emotional wellbeing. Additionally, when you're feeling low on hugs, consider reaching out to friends, family, or even strangers for a hug to help replenish your 'hug tank'. Remember, it's important to take care of yourself and prioritize your emotional needs. |

other resources in your community to help you build a network of understanding and compassionate individuals who can provide you with the emotional support you need.

It's important to remember that healing is a process, and it takes time and effort to make meaningful progress. But with a compassionate and supportive therapeutic environment, I believe you can overcome these challenges and find the peace and understanding you're seeking.

**Mistral-Instruct-V0.2**: I'm really sorry to hear that you've been feeling overwhelmed and experiencing panic attacks frequently. It's completely normal to feel lost and alone when dealing with mental health issues, especially when it feels like those closest to you don't fully understand what you're going through.

There are a few things you might find helpful in managing your symptoms and improving your overall well-being:

1. Reach out to mental health professionals: Consider speaking with a mental health professional, such as a therapist or counselor, who can provide you with evidence-based strategies and support. They can help you develop coping skills and provide a safe and non-judgmental space for you to express your feelings.

2. Practice self-care: Engage in activities that bring you joy and help you relax, such as taking a warm bath, reading a book, or going for a walk in nature. Make sure to get enough sleep, eat well, and exercise regularly.

3. Connect with online communities: There are many online communities and forums where people share their experiences and offer support to one another. These communities can be a valuable source of understanding, validation, and connection.

4. Consider joining a support group: Joining a support group, either in-person or online, can provide you with a sense of community and connection with others who are going through similar experiences.

5. Practice mindfulness and relaxation techniques: Mindfulness practices, such as deep breathing, meditation, or progressive muscle relaxation, can help you manage stress and anxiety in the moment.

Remember, it's important to be patient with yourself and give yourself the time and space you need to heal. Mental health struggles can be long-term, and it's okay to take things one step at a time.

I hope this information is helpful, and please don't hesitate to reach out if you have any further questions or need additional support. You're not alone in this, and there are resources and people who care and want to help.

**Fine-tuned Mistral-Instruct-V0.2**: I'm truly sorry to hear that you've been experiencing such intense emotions and a sense of isolation. It's important to acknowledge the validity of your feelings and the challenges you're facing. The symptoms you're describing, such as difficulty focusing and frequent panic attacks, are common indicators of anxiety and stress. It's essential to prioritize self-care and coping strategies to help manage these symptoms.

One effective coping strategy is practicing mindfulness meditation, which can help you stay present and focused on the current moment, rather than being overwhelmed by thoughts about the past or future. Deep breathing exercises are also a simple yet powerful tool for managing anxiety and panic attacks. By taking slow, deep breaths, you can help calm your body and mind, reducing the intensity of your symptoms.

Another important aspect of managing anxiety and stress is identifying and addressing the root causes. It's crucial to explore the underlying emotions and beliefs that contribute to your feelings of overwhelm and isolation. This may involve examining past experiences, relationships, or beliefs that have shaped your current emotional landscape. By gaining a deeper understanding of these factors, you can begin to challenge and re-frame negative thought patterns and beliefs, ultimately leading to greater emotional resilience and a sense of connection and belonging.

It's also essential to remember that seeking help and support is a sign of strength, not weakness. It's important to reach out to trusted friends, family members, or mental health professionals for additional support and guidance. By taking a proactive approach to managing your anxiety and stress, you'll be better equipped to navigate life's challenges and find a sense of peace and purpose.

