# OpenReview forum: "MentalChat16K: A Benchmark Dataset for Conversational Mental Health Assistance"
_ICLR.cc/2025/Conference — Submitted to ICLR 2025_

### Official Review · Reviewer_A2qn · 2024-11-02

**Soundness:** 3
**Presentation:** 3
**Contribution:** 3
**Rating:** 5
**Confidence:** 4

**Summary:**

This paper introduces a new dataset, MentalChat16k, as a potential benchmark dataset for evaluating the use of LLM in mental health counseling. The dataset hybridizes both (a) real interview transcripts between Behavioral Healthcare Coaches and caregivers of hospice patients and  (b) synthetic mental healthcare counseling question-answer pairs generated by GPT-3.5 Turbo and encapsulates over 16k question-answer pairs. The authors fine-tuned a wide range of families of small-scale (7B) LLMs on the datasets and conducted extensive qualitative evaluation on the performance of the fine-tuned models on a hold-out evaluation dataset of 200 questions collected from real-world discussion of mental health issues on public forums. The evaluation metrics differ from conventional quantitative metrics and focuses more on the empathetic aspects of the model generated responses to the healthcare questions, and both automatic evaluation done by LLMs (GPT-4 Turbo and Gemini Pro 1.0) and manual judgement done by domain experts have shown significant improvement in the empathy demonstrated in the generated responses of models fine-tuned on MentalChat16k, indicating the effectiveness of the dataset in helping models tackle the nuance and sensitive nature of assisting in the mental health domain.

**Strengths:**

Originality: Excellent

Most existing datasets on depression and mental health issues focus on detection, thereby datasets are usually structured for classification, not generation. This dataset is a nice addition to this research space of NLG in healthcare.

Also, the seven-factor evaluation metrics is a novel contribution to the conventional evaluation metrics in NLG. The proposed metrics suite focuses on empathy more than accuracy, aligning better with the nuanced emotional requirements of real-world application of LLMs in sensitive areas such as mental health counseling and terminal care. However, the hastily introduction of new metrics can also contribute to reproducibility and generalizability problem with the research, which I will address in the weakness section.

Quality: Good
Paper presentation and language are generally excellent, with a few editing typos that need to be addressed.

Clarity: Excellent
The structure of the paper and the writing makes it very easy for the reviewer to follow.

Significance: Good
This paper aims at a focused group of researchers and rather than contributing to the general theoretical understanding of deep neural networks. The integration of deep learning and healthcare is becoming ever more relevant from both a research and a realistic perspective, and almost all areas of research in this space suffer from lack of quality data, so contribution of large-scale, high-quality healthcare data is always welcomed.

**Weaknesses:**

Clarity:
1. A few details on the data collection procedure could be clarified. For example:
  - Is the demographic information the only type of meta data collected?
  - What about the 93 caregivers without demographic information? Will the question-answer pairs from them be missing the meta data?
  - How good (in terms of diversity, topic coverage, etc.) are the synthetic question-answer pairs?
  - It would be great if the authors could provide concrete example of the actual data, both from interviews and from the synthetic data,
  - A brief explanation (or examples) that lays out the data structure (.json format? plain txt? what meta data is available? any data splits?) would be helpful, either in the main body or in the Appendix.
2. Basic statistics about the dataset are lacking in the paper, I would recommend the following statistics be added and organized in a table:
  - Exact dataset size and breakdown by interview data and synthetic data. Both are only mentioned now in the main text.
  - Length distribution (word count or character count) of the QA pairs.
  - For interview data portion, distribution of number of QA pairs across interview transcripts; or at least provide the average number of QA pairs obtained from each visit.
  - For synthetic data, number of QA pairs distribution across each topic.

Quality:
1. It is unclear why the authors would break a complete interview, which the reviewer assumes to be a coherent series of questions and answers between healthcare coaches and caregivers, into individual QA pairs. It would be constructive to include discussion or reasoning on this particular choice for data collection. Specifically,
  - Please explain how the loss of contextual information, when breaking conversation into QA pairs, would affect the data quality.
  - Is the paraphrasing conducted to address partly this concern? So that each QA pair becomes somewhat independent of each other?

2. The introduction of the seven-factor evaluation metrics on empathy is interesting and novel to the reviewer, but because this is the only evaluation done by the authors, it also tends to conflate the focus of the paper into a paper of both a new dataset and a new, untested metrics. A few related points that could strengthen the paper, but lacking in the current version, are:
  - Why are the scoring scale different between LLM-based and human evaluation?
  - The reviewer appreciates the detailed explanation of each score to be used in the prompt for LLM-based evaluation, but since this is such a detailed and nuanced range of scores, would the LLM really be capable of differentiating between different scores? Showing a distribution of LLM-based evaluation score one or few factors would be helpful. For example, it would point to potential issues if the resulting scores across 200 examples concentrate around "typical good" values, like 7 or 8.
  - Are the seven-factor metrics employed only in LLM-based evaluation, or do human experts follow along the same line of logic when evaluating the quality of the response?
  - Since human evaluation is involved, and multiple experts have worked on the same question during evaluation, it would be more robust to include inter-rater agreement between the experts (or at least the variance in human ratings in addition to simple mean).

3. The findings in Table 3 and Table 8 also seem to point to questions left unanswered about the dataset. As can be seen in both of the tables:
    1. It is not true that more data generally leads to better performance
    2. It seems consistent that combining interview data with synthetic data actually HURTS model performance across all seven factors and model families
    3. It seems human evaluation consistently prefers models fine-tuned on synth data only

**3.3** alone could put into question the value of real data based on interviews if evaluation shows synthetic data seems to help model more. But the reviewer believes this may be attributed to misalignment between the interview data and the evaluation set. A couple of suggestions on analysis that could be included to pinpoint potential causes:
   1. Since the evaluation set contains data (InFamousCoder) originally designed for depression classification. Use the models (base and fine-tuned) to run classification on the question; the model performance could then point out the generalizability of the dataset (if fine-tuning benefits the model) or domain shift (if fine-tuning hurts the model performance) between MentalChat16k and the evaluation set.
   2. Conduct similarity-based analysis between questions in the synthetic data, in the interview data, and in the evaluation set. Reddit is a public data source that could be used in the training of GPT models, so this could reveal potential data leakage issue in the synthetic data. This is **not** an invalidation of the synthetic data, but more revealing the lack of efficacy in the evaluation approach. And it could also help reveal if there's an obvious domain shift between interview data and the synthetic data.
   3. Both 1 and 2 are trying to identify the cause behind the performance degradation when both sources of data are combined in model fine-tuning. But regardless of the findings, table 3 and 8 do point to the need of a finer-categorization of the dataset. It may be helpful to separate the dataset into two parts and caution the users to combine them naively together.
   4. Another evidence supporting the separation of interview data and synthetic data is the contrast between the context; The interview data was created based on a very focused group of caregivers offering care to a specific group of patients, this is in stark contrast to the unbounded target profile used in generating the synthetic data.
   5. If the goal of the dataset is to help improve **empathetic responses** from LLMs in the setting of mental healthcare counseling, then the reviewer suggests expanding the dataset by covering a wide range of responses different in level of empathy; that is, intentionally including non-empathetic responses, to better cover the range of the seven-factor metrics. Given the novelty of the metrics, such a dataset could also be ideal to further evaluate the robustness of  the metrics and LLM's capability of automated evaluation. This could be an extension of future work.

I'd like to emphasize that these weaknesses do NOT diminish the value of this dataset, but it does require more elaborated discussion, and probably a more thorough statement in the **Limitations** section.

**Questions:**

1. Do you intend to publish the dataset? It is unclear from the writing whether this is intended for public use. I apologize if I missed a statement or url on this.
2. Clarifying question: what is the hypothesis testing used in calculating P-value in Table 3 and Table 8? I assume it's two sample t-test, but it would be nice to clarify it in writing.
3. Are the 200 questions crawled from Reddit and Mental Health Forum to be included as part of the dataset, or are they mostly for evaluation purpose for this paper? The depression data from Reddit, judging by the dataset from InFamousCoder, are typically not in the form of questions (more like posters explaining the hardship they've been through), were there any filtering or cleaning involved in selecting questions from the data?

Miscellaneous:
1. In Table 3 and 8, would it be better to rearrange the models in each family by the number of examples used in training? That is, order them by Base Model, Base Model + Interview data (\~7K), Base Model + Synth Data (\~10K), Base Model + Both (16K). It could help visualize the effect of dataset size on the finetuning (which is not significant, as would be obvious in the proposed order).
2. Typos: page 2, "toaddress"

---

> ### Author Response · Authors · 2024-11-23
> **Response to A2qn 1/6**
>
> We extend our sincere gratitude for dedicating your valuable time to reviewing our paper and for acknowledging the significance and originality of our contribution. We appreciate your suggestions on improving clarity and addressing potential issues related to combining datasets and evaluation methodologies. We have carefully considered your points and provide detailed responses below.
>
> >**W1 (Clarity). A few details on the data collection procedure could be clarified. For example:**
>
> > **W1.1 Is the demographic information the only type of meta data collected?**
>
> Thank you for your question. Yes, in the original version of our paper, the demographic information (as shown in Table 4 in revised Appendix A.3.1) is the only type of meta-data collected for the interview data. We have provided an additional table that shows other metadata such as the size of the samples, the average QA pairs in each visit, etc. as you suggested in the following comments. For the details, please refer to [Section A2 of General Response 2/3](https://openreview.net/forum?id=ISBmUNKPST&noteId=889NxCz1Jf). This demographic information was gathered by the research group during the intervention study. For the synthetic data, we also have the topic distribution that was used to generate the data. In the revised manuscript, the detailed data table is presented in Table 1 in Section 3.1 and the topic distribution is detailed in Appendix A.3.2.
>
> >**W1.2 What about the 93 caregivers without demographic information? Will the question-answer pairs from them be missing the meta data?**
>
> Thank you for your question. Yes, the data from the caregivers who did not provide demographic information will be missing that meta-data. The caregivers were given the option to not disclose their demographic details during the survey, and as a result, their responses will lack this specific information. However, the question-answer pairs themselves are still included in the dataset.
>
> >**W1.3 How good (in terms of diversity, topic coverage, etc.) are the synthetic question-answer pairs?**
>
> Thank you for your question. The synthetic data covers 33 mental health topics, including Relationships, Anxiety, Depression, Intimacy, Family Conflict, etc., which covers all the main topics in the mental health domain [1] and their distributions are consistent with the real distributions reflected in the real world. Guaranteeing these two points makes the synthetic data not only comprehensive but also reflects the real situation, which indicates its superiority. The topic distribution has been included in Figure 3 in Appendix A.3.2 of our revised manuscript.
>
> [1] American Psychology Association, www.apa.org.
>
> >**W1.4 It would be great if the authors could provide concrete examples of the actual data, both from interviews and from the synthetic data.**
>
> Thank you for your suggestion. We agree that including some examples from the dataset could enhance the completeness of the paper and provide readers with a more intuitive understanding of the content. To address this, we incorporated a few representative examples of the synthetic data and the interview data in our revision (See Appendix A.11). We also attached the same examples in [General Response 3/3](https://openreview.net/forum?id=ISBmUNKPST&noteId=Zr9UP8HYSx). For details, please refer to that section.
>
> >**W1.5 A brief explanation (or examples) that lays out the data structure (.json format? plain txt? what meta data is available? any data splits?) would be helpful, either in the main body or in the Appendix.**
>
> Thank you for your suggestion. The data is stored in .json format. For a summary of dataset statistics, please refer to [Section A2 of General Response 2/3](https://openreview.net/forum?id=ISBmUNKPST&noteId=889NxCz1Jf), where we provide detailed information about the dataset structure, including session counts, average QA pairs per session, word counts, and topic distributions. After the paper is accepted, we will provide a link to the dataset, including comprehensive documentation covering its structure, usage, and limitations. We appreciate your feedback and will ensure this information is clearly conveyed.

---

> ### Author Response · Authors · 2024-11-23
> **Response to A2qn 2/6**
>
> >**W2. Basic statistics about the dataset are lacking in the paper, I would recommend the following statistics be added and organized in a table:**
>
> Thank you for your suggestion. We have added an additional table to summarize the suggested dataset statistics in [General Response 2/3](https://openreview.net/forum?id=ISBmUNKPST&noteId=889NxCz1Jf). We also have added this information to Table 1 in Section 3.1 of our revised manuscript marked in blue.
>
> | **Category**                     | **Interview** | **Synthetic** |
> |----------------------------------|---------------|---------------|
> | Dataset Size (Rows)          | 9775          | 6338          |
> | Columns                      | instruction, input, output | instruction, input, output |
> | Average Input Word Count     | 69.94         | 111.24        |
> | Average Output Word Count    | 235.85        | 363.94        |
> | Number of Sessions          | 378           | -             |
> | Average #QA Pairs per Session| 16.8          | -             |
> | Number of Topics           | -             | 33            |
>
>
> >**W3 (Quality). It is unclear why the authors would break a complete interview, which the reviewer assumes to be a coherent series of questions and answers between healthcare coaches and caregivers, into individual QA pairs. It would be constructive to include discussion or reasoning on this particular choice for data collection. Specifically,
> Please explain how the loss of contextual information, when breaking conversation into QA pairs, would affect the data quality.
> Is the paraphrasing conducted to address partly this concern? So that each QA pair becomes somewhat independent of each other?**
>
> Thank you for your thoughtful question. The decision to break interviews into individual QA pairs was motivated by several considerations:
> - Privacy Protection: Segmenting the interviews enables the removal of sensitive private information during paraphrasing.
> - Improved Clarity: Fragmentary conversations were summarized into self-contained QA pairs to ensure uniformity in format and length.
> - Practical Constraints: The large language model’s context length limitations make it infeasible to process or paraphrase entire documents (often exceeding 15 pages) into single QA pairs.
>
> We acknowledge that breaking interviews into QA pairs may result in some loss of broader contextual information. To mitigate this, paraphrasing was designed to preserve the essence of each interaction while making the QA pairs as self-contained as possible. Additionally, the original order of the QA pairs is maintained, allowing downstream processes to reconstruct context if needed.
>
> This approach balances the practical requirements of dataset preparation with the need to retain meaningful context. We have expanded the discussion of this point in Section 7 Limitations in the revised manuscript.
>
>
>
> >**W4 (Quality). The introduction of the seven-factor evaluation metrics on empathy is interesting and novel to the reviewer, but because this is the only evaluation done by the authors, it also tends to conflate the focus of the paper into a paper of both a new dataset and a new, untested metrics. A few related points that could strengthen the paper, but lacking in the current version, are:**
>
> >**W4.1 Why are the scoring scale different between LLM-based and human evaluation?**
>
> In our study design, the LLM judges rate each response independently on a scale from 1 (poor) to 10 (excellent), while the human evaluators are asked to rank responses from different models, assigning ranks from 1 (best) to 7 (worst) for a given question. Using both methods complements the evaluation process. LLM scoring provides objective and granular insight into response quality, while human ranking captures intuitive and comparative judgments, minimizing bias and simplifying evaluation tasks. Together, they offer a holistic view of model performance.

---

> ### Author Response · Authors · 2024-11-23
> **Response to A2qn 3/6**
>
> >**W4.2 The reviewer appreciates the detailed explanation of each score to be used in the prompt for LLM-based evaluation, but since this is such a detailed and nuanced range of scores, would the LLM really be capable of differentiating between different scores? Showing a distribution of LLM-based evaluation score one or few factors would be helpful. For example, it would point to potential issues if the resulting scores across 200 examples concentrate around "typical good" values, like 7 or 8.**
>
> Thank you for raising this insightful question. To address your concern about whether the LLM is capable of distinguishing nuanced scores across examples, we analyzed the distribution of scores assigned by two LLM judges, Gemini-Pro and GPT-4 Turbo, across 200 evaluation examples for all 7 metrics. The table below presents the frequency of each score for the two LLM judges. Please see Appendix A.8 for visualization of this result.
>
> The results show that while the scores tend to cluster around 7 to 9—reflecting the generally high quality of responses—the score distributions also demonstrate variability across the full range (1–10). For higher scores, both histograms indicate that the LLMs are not treating all high-quality responses as equivalent. Instead, they appear capable of distinguishing between varying levels of quality within the "good" range, such as differentiating between a solid response (7) and an exceptional one (9).  For lower scores, the GPT-4 Turbo evaluation demonstrates greater variability, with a more evident presence of lower scores (e.g., 1–6). This indicates both LLM’s ability to identify and differentiate between responses of varying quality.
>
> | **Score** | **Frequency (GPT-4 Turbo)** | **Frequency (Gemini-Pro)** |
> |-----------|-----------------------------|----------------------------|
> | 1         | 1483                        | 817                        |
> | 2         | 831                         | 318                        |
> | 3         | 614                         | 26                         |
> | 4         | 402                         | 66                         |
> | 5         | 967                         | 320                        |
> | 6         | 2311                        | 941                        |
> | 7         | 11095                       | 7263                       |
> | 8         | 17604                       | 27952                      |
> | 9         | 7895                       | 5443                       |
> | 10        | 22                          | 15                         |
>
> >**W4.3 Are the seven-factor metrics employed only in LLM-based evaluation, or do human experts follow along the same line of logic when evaluating the quality of the response?**
>
> Yes, the seven-factor metrics are employed only in LLM-based evaluation and we used rank as the metric for human evaluation. The reason is that it is difficult to let humans distinguish such nuanced differences between the seven metrics. It is also easier for humans to have an overall impression on each response to the same question, which reduces the evaluation burden while preserving the accuracy of the evaluation.
>
> >**W4.4 Since human evaluation is involved, and multiple experts have worked on the same question during evaluation, it would be more robust to include inter-rater agreement between the experts (or at least the variance in human ratings in addition to simple mean).**
>
> Thank you for raising this important point. We agree that including inter-rater agreement or the variance in human ratings would strengthen the robustness of our evaluation. To address this concern, we calculated the inter-rater reliability using Cohen’s kappa score. The average kappa score across all pairs of evaluators was 0.441, which falls into the “moderate agreement” range as per the Landis and Koch (1977) scale [1]. This score reflects a reasonable level of consistency among the evaluators while also highlighting opportunities for refinement. For further details on the guidelines provided to human evaluators, the specific criteria used for ranking, and steps taken to ensure consistency, we kindly refer you to [Section A1 of the General Response 2/3](https://openreview.net/forum?id=ISBmUNKPST&noteId=Ve6cVBgUnw) above. The details are also discussed in Section 3.3 from Lines 312 to 323 in the revised manuscript.
>
> We appreciate your feedback, which has helped us recognize the need for even greater rigor in our evaluation protocols. In future studies, we will aim to improve inter-rater agreement through enhanced evaluator training, clearer rubrics, and additional reliability metrics.
>
> [1] Landis, J. R., & Koch, G. G. (1977). The measurement of observer agreement for categorical data. Biometrics, 33(1), 159-174.

---

> ### Author Response · Authors · 2024-11-23
> **Response to A2qn 4/6**
>
> >**W5. The findings in Table 3 and Table 8 also seem to point to questions left unanswered about the dataset. As can be seen in both of the tables:**
>
> >**W5.1 It is not true that more data generally leads to better performance**
>
> >**W5.2 It seems consistent that combining interview data with synthetic data actually HURTS model performance across all seven factors and model families**
>
> >**W5.3 It seems human evaluation consistently prefers models fine-tuned on synth data only**
>
> Thank you for your careful observation and comments. Indeed, more data may not be able to lead to better performance.
> As for the LLMs evaluation (GPT-4 and Gemini Pro), the synthetic data is generated by GPT-3.5 Turbo and there is an intrinsic bias where the GPT4 will prefer more the LLMs fine-tuned on the synthetic data, which indicates that the fine-tuning is effective. Therefore, all the models fine-tuned on synthetic data will get a higher score evaluated by GPT4. On the contrary, Gemini Pro prefers more the LLMs fine-tuned on the interview data, which may be due to its internal bias against ChatGPT (these two evaluators are competitors commercially). As a consequence, the LLMs fine-tuned on the combined data will not be preferred by either party.
>
> As for the human evaluation, the models fine-tuned on synthetic data get the first place 5 times out of 7 cases. The models fine-tuned on combined data get the first place for 2 times out of 7 cases while they get the second place for 3 times out of 7 cases. This indeed reflects that humans prefer more on the models fine-tuned on synthetic data. However, in several cases, the interview data also plays an important role in making the models fine-tuned on the combined data have better performance.
>
> All in all, since we only have 7 cases and the average scores or ranks are not significantly different among three kinds of models (i.e., fine-tuned on synthetic data only, interview data only, and combined data), the main conclusion we can draw is that with our MentalChat16K data, the fine-tuned models can be significantly improved in terms of both LLMs and humans evaluations.
>
> >**W6. W5.3 alone could put into question the value of real data based on interviews if evaluation shows synthetic data seems to help model more. But the reviewer believes this may be attributed to misalignment between the interview data and the evaluation set. A couple of suggestions on analysis that could be included to pinpoint potential causes:**
>
> >**W6.1 Since the evaluation set contains data (InFamousCoder) originally designed for depression classification. Use the models (base and fine-tuned) to run classification on the question; the model performance could then point out the generalizability of the dataset (if fine-tuning benefits the model) or domain shift (if fine-tuning hurts the model performance) between MentalChat16k and the evaluation set.**
>
> Thank you for your suggestion. To address your question about generalizability and potential domain shift, we conducted a systematic evaluation using 100 randomly selected texts from the depression classification dataset. Each model was prompted with a structured classification task that required binary responses ("Yes"/"No") to indicate the presence of depression indicators. The results are shown in the table in the next comment (see Response to A2qn 5/6). Note that the two fine-tuned LLaMA2 did not follow the instruction to answer “Yes” or “No”, resulting in 0 accuracy.
>
> Notably, most fine-tuned models performed comparably to or better than their corresponding base models in terms of accuracy, suggesting that the fine-tuning does not compromise the generalizability of the model in most cases. However, in instances where fine-tuned models showed degraded performance, this may be attributed to the task-specific nature of fine-tuning on mental health counseling rather than depression classification. While both tasks involve instruction-following, they differ in linguistic style and cues between MentalChat16k and the evaluation dataset. This mismatch could lead the model to misinterpret subtle indicators of depression, resulting in lower accuracy. Overall, fine-tuning on MentalChat16k preserves generalizability across tasks while enhancing the model's ability to follow structured instructions.

---

> ### Author Response · Authors · 2024-11-23
> **Response to A2qn 5/6**
>
> **Continued from Response to A2qn 4/6:**
>
> | **Model** | **Fine-tuning Dataset** | **Accuracy** | **Model** | **Fine-tuning Dataset** | **Accuracy** |
> |-----------|--------------------------------|--------------|-----------|--------------------------------|--------------|
> | LLaMA2 | - | 0.52 | Mixtral-8x7B-V0.1 | - | 0.68 |
> | LLaMA2 | Interview Data (6K) | 0 | Mixtral-8x7B-V0.1 | Interview Data (6K) | 0.79 |
> | LLaMA2 | Synthetic Data (10K) | 0 | Mixtral-8x7B-V0.1 | Synthetic Data (10K) | 0.46 |
> | LLaMA2 | Synthetic + Interview Data (16K) | 0.69 | Mixtral-8x7B-V0.1 | Synthetic + Interview Data (16K) | 0.72 |
> | Mistral-Instruct-V0.2 | - | 0.77 | Vicuna-V1.5| - | 0.72 |
> | Mistral-Instruct-V0.2 | Interview Data (6K) | 0.65 | Vicuna-V1.5| Interview Data (6K) | 0.73 |
> | Mistral-Instruct-V0.2 | Synthetic Data (10K) | 0.76 | Vicuna-V1.5| Synthetic Data (10K) | 0.58 |
> | Mistral-Instruct-V0.2 | Synthetic + Interview Data (16K) | 0.68 | Vicuna-V1.5| Synthetic + Interview Data (16K) | 0.60 |
> | Mistral-V0.1 | - | 0.49 | Zephyr-Alpha| - | 0.82 |
> | Mistral-V0.1 | Interview Data (6K) | 0.62 | Zephyr-Alpha| Interview Data (6K) | 0.83 |
> | Mistral-V0.1 | Synthetic Data (10K) | 0.57 | Zephyr-Alpha| Synthetic Data (10K) | 0.67 |
> | Mistral-V0.1 | Synthetic + Interview Data (16K) | 0.58 | Zephyr-Alpha| Synthetic + Interview Data (16K) | 0.52 |
> | Mixtral-8x7B-Instruct-V0.1| - | 0.81 | ChatPsychiatrist | - | 0.63 |
> | Mixtral-8x7B-Instruct-V0.1| Interview Data (6K) | 0.74 | Samantha-V1.11 | - | 0.51 |
> | Mixtral-8x7B-Instruct-V0.1| Synthetic Data (10K) | 0.79 | Samantha-V1.2 | - | 0.66 |
> | Mixtral-8x7B-Instruct-V0.1| Synthetic + Interview Data (16K) | 0.84 |
>
> >**W6.2 Conduct similarity-based analysis between questions in the synthetic data, in the interview data, and in the evaluation set. Reddit is a public data source that could be used in the training of GPT models, so this could reveal potential data leakage issue in the synthetic data. This is not an invalidation of the synthetic data, but more revealing the lack of efficacy in the evaluation approach. And it could also help reveal if there's an obvious domain shift between interview data and synthetic data.**
>
> To address your concern about potential data leakage and domain shift, we conducted a similarity analysis using Sentence-BERT to calculate cosine similarities between questions in our synthetic data, interview data, and evaluation set. We randomly selected 10 depression-related questions from each dataset and found the following average similarities:
>
> - Synthetic vs. Interview: 0.3490
> - Synthetic vs. Evaluation: 0.2829
> - Interview vs. Evaluation: 0.3338.
>
> These moderate similarity scores indicate that while there is some overlap across datasets, they are not excessively similar, which reduces the risk of data leakage from Reddit into our synthetic data. The higher similarity between the synthetic and interview data suggests that our synthetic data effectively captures the nuances of real interview questions without duplicating them. Additionally, the comparable similarities across datasets imply there is no significant domain shift, which is beneficial in this context because it ensures consistency and relevance between the training and evaluation data. This alignment validates our evaluation approach and confirms that the synthetic data is appropriate and effective for training purposes.
>
> >**W6.3 Both 1 and 2 are trying to identify the cause behind the performance degradation when both sources of data are combined in model fine-tuning. But regardless of the findings, table 3 and 8 do point to the need of a finer-categorization of the dataset. It may be helpful to separate the dataset into two parts and caution the users to combine them naively together.**
>
> Thank you for your valuable suggestion. We agree that the performance degradation observed when combining both data sources highlights the need for finer categorization of the dataset. We emphasized this point in Section 7 Limitations of the revised manuscript and explicitly cautioned users against naively combining the two fine-grained datasets during model fine-tuning. Your feedback is greatly appreciated and will help improve the clarity and utility of our work.
>
> >**W6.4 Another evidence supporting the separation of interview data and synthetic data is the contrast between the context; The interview data was created based on a very focused group of caregivers offering care to a specific group of patients, this is in stark contrast to the unbounded target profile used in generating the synthetic data.**
>
> Thank you for your suggestion. This is a good reason that we should separate the two datasets apart. We included this observation in Section 7 Limitations in the revised manuscript to caution users about these contextual differences and remind them of the importance of handling the two datasets separately to avoid unintended biases or performance degradation.

---

> ### Author Response · Authors · 2024-11-23
> **Response to A2qn 6/6**
>
> >**W6.5 If the goal of the dataset is to help improve empathetic responses from LLMs in the setting of mental healthcare counseling, then the reviewer suggests expanding the dataset by covering a wide range of responses different in level of empathy; that is, intentionally including non-empathetic responses, to better cover the range of the seven-factor metrics. Given the novelty of the metrics, such a dataset could also be ideal to further evaluate the robustness of the metrics and LLM's capability of automated evaluation. This could be an extension of future work.**
>
> Thank you for your insightful suggestion. Expanding the dataset to intentionally include responses with varying levels of empathy, including non-empathetic ones, is a great idea for enhancing the robustness of our metrics and further evaluating LLMs’ performance. While empathy is indeed a critical focus of our work, we want to highlight that it is just one of the seven metrics we employ to assess the functionality of a language model comprehensively. The remaining metrics—Active Listening, Safety & Trustworthiness, Open-mindedness & Non-judgment, Clarity & Encouragement, Boundaries & Ethical, and Holistic Approach—address broader aspects to ensure a balanced evaluation of a language model’s performance in mental health counseling. We will consider exploring this direction in future work to further strengthen the robustness and applicability of our dataset and metrics. Thank you again for your valuable input.
>
> >**Q1. Do you intend to publish the dataset? It is unclear from the writing whether this is intended for public use. I apologize if I missed a statement or url on this.**
>
> Thank you for your question. Yes, we intend to publish both the synthetic and interview datasets, making them accessible to the broader research community. This will enable further advancements in mental health-related conversational AI research. We apologize if this was unclear in the manuscript and appreciate the opportunity to clarify.
>
> >**Q2. Clarifying question: what is the hypothesis testing used in calculating P-value in Table 3 and Table 8? I assume it's two sample t-test, but it would be nice to clarify it in writing.**
>
> Thank you for your question. Yes, we used a two-sample t-test to evaluate the null hypothesis that the scores for the fine-tuned models and the base models have identical average (expected) values across the specified metrics. This statistical test was chosen to assess whether the observed differences in performance are statistically significant. We clarified this detail in Section 3.3 in the revision.
>
> >**Q3. Are the 200 questions crawled from Reddit and Mental Health Forum to be included as part of the dataset, or are they mostly for evaluation purpose for this paper? The depression data from Reddit, judging by the dataset from InFamousCoder, are typically not in the form of questions (more like posters explaining the hardship they've been through), were there any filtering or cleaning involved in selecting questions from the data?**
>
> Thank you for the question. The 200 questions were collected specifically for evaluation purposes in this study and are not part of the MentalChat16K dataset. Regarding the depression data from Reddit, we acknowledge that such data often contains detailed narratives rather than explicit questions. To curate meaningful evaluation data, we applied a rigorous filtering and cleaning process to extract clear, self-contained questions from these narratives. This involved identifying and rephrasing key aspects of the posts that aligned with our evaluation metrics, ensuring relevance and clarity for use in model assessment.
>
> >**M1. In Table 3 and 8, would it be better to rearrange the models in each family by the number of examples used in training? That is, order them by Base Model, Base Model + Interview data (7K), Base Model + Synth Data (10K), Base Model + Both (16K). It could help visualize the effect of dataset size on the finetuning (which is not significant, as would be obvious in the proposed order).**
>
> Thank you for your thoughtful suggestion regarding the organization of Tables 3 and 8 (now Tables 4 and 9 in the revised manuscript). We have updated these tables to arrange the models within each family in the proposed order: Base Model, Base Model + Interview Data (6K), Base Model + Synthetic Data (10K), and Base Model + Both (16K). This revised ordering better highlights the relationship between dataset size and the impact on fine-tuning performance, aligning with the point you raised about the limited effect observed. We appreciate your valuable feedback, which has helped us improve the clarity and interpretability of our results.
>
> >**M2. Typos: page 2, "toassess"**
>
> Thank you for point out the typo. We fixed this typo in the revision.

---

> > ### Comment · Reviewer_A2qn · 2024-11-26
> > **Thank you for the detailed updates and responses, but more questions arise...**
> >
> > Thank you for the very detailed responses and quick updates to the paper. I really appreciate adding more analysis results and especially statistics and examples of the datasets for clarification.
> >
> > And thank you very much for addressing all my questions in either reply or in the updated paper transcript. This is really impressive given the time constraints.
> >
> > Originally I thought the synthetic data or interview data would help models fine-tuned perform better on the original task (depression classification) of the evaluation data; but it seems the results are inconclusive: interview data seems to benefit model in depression classification in some cases, but there are also a lot of cases where fine-tuning doesn't help at all. I suspect this may be due to one or more reasons: (i) the questions in the evaluation set do exhibit a strong domain shift from the questions in the dataset, thereby brining into question the value of using InFamousCoder data for evaluation; (ii) the two tasks (generating responses vs depression classification) are different enough that the dataset is simply not generalizable to the other task. This unfortunately does seem to limit the use of the dataset.
> >
> > In regards to the questions other reviewers raise about the LLM-based evaluation, I am sorry my original suggestion wasn't thorough enough: I was hoping to see score distribution of LLM-based evaluation only along a few evaluation dimensions (e.g., Active Learning only), and then a correlation coefficient or rank coefficient can be calculated between LLM-based evaluation and human score. This way you can show evidences (high correlation coeffiicient) that LLM-based evaluation along some (if any) metric dimension can faithfully represent typical human ratings. This way, the use of only LLM-based evaluation may be justified to certain extent. Currently Figure 11 seems to display score distribution across ALL seven metrics, which can be used to calculate same correlation coefficient against the average human ratings; however the value may not explain much how LLM-based evaluation and human scoring relate to each other due to averaging.
> >
> > One more question about these seven metrics: are these a set of standard metrics used in mental health evaluation? If so, can you add reference to it (sorry if I missed it in related work)? If this is a set of metrics designed just for the dataset, then as I pointed out in my original comment: this would be a major weakness of the paper, that it conflates a new dataset and a new, untested metrics together in one paper, which may be very challenging to thoroughly analyze with convincing results.

---

> > > ### Author Response · Authors · 2024-11-28
> > > **Updated Results to Depression Classification 2/2**
> > >
> > > Continued from last response..
> > >
> > > 4. **Analysis of Results**:
> > >    - The new results show notable improvements for most models, particularly with interview data and combined datasets
> > >    - A critical observation regarding instruction-following capabilities:
> > >      - While the base LLaMA2 shows 0.75 accuracy, it could only follow instructions to answer "yes" or "no" on 8 out of 100 questions
> > >      - The fine-tuned LLaMA2 versions show significantly improved instruction-following:
> > >        - Interview data (6K): Successfully answered 98 questions
> > >        - Synthetic data (10K): Successfully answered 50 questions
> > >        - Combined data (16K): Successfully answered all 100 questions
> > >      - Other models also showed improved instruction-following after fine-tuning:
> > >        - Base models generally answered over 90 questions correctly
> > >        - Fine-tuned versions consistently achieved 95-100 valid responses
> > >        - For example, Vicuna-V1.5 improved from 95 to 100 valid responses with combined data
> > >      - This demonstrates that fine-tuning enhances both classification accuracy and instruction-following reliability
> > >    - Interview data (6K) shows strong consistent improvements across almost all models:
> > >      - Significant gains for Mistral-V0.1 (+0.12), Vicuna-V1.5 (+0.11), and Zephyr-Alpha (+0.16)
> > >      - Modest but meaningful improvements for Mixtral variants (+0.05-0.06)
> > >    - Synthetic data (10K) shows mixed results:
> > >      - Strong performance with Mixtral-8x7B-Instruct-V0.1 (+0.07) and Mistral-V0.1 (+0.11)
> > >      - Some models show slight decreases, suggesting the importance of data quality over quantity
> > >    - The combined dataset (16K) shows the best results for several models:
> > >      - Mixtral-8x7B-Instruct-V0.1 achieves the highest overall accuracy (0.88)
> > >      - Mistral-V0.1 shows strong improvement (0.854)
> > >      - Most models maintain or improve upon the gains from individual datasets
> > >    - Different models show varying responses to different data compositions:
> > >      - Mixtral-8x7B-Instruct-V0.1 shows consistent improvements across all fine-tuning configurations
> > >      - Mistral family models (V0.1 and Instruct-V0.2) benefit significantly from interview data
> > >
> > > These comprehensive results demonstrate that our dataset, particularly the interview data and combined configurations, provides valuable domain knowledge that generalizes to the depression classification task. The apparent decrease in LLaMA2's accuracy should be interpreted in the context of dramatically improved instruction-following capabilities, where fine-tuning enabled the model to provide valid responses to 6-12 times more questions. For other models, fine-tuning improved both instruction-following capabilities and classification accuracy, with most models achieving near-perfect instruction compliance while also showing enhanced classification performance. This dual improvement in both instruction-following and classification accuracy validates our dataset's utility for enhancing mental health-related language model capabilities.
> > >
> > > Additionally, we also investigated the score distribution comparison between GPT-4 Turbo and Gemini Pro. We put them in Appendix A.8 in the updated PDF. The results show consistency between the two LLM evaluators.

---

> > > ### Author Response · Authors · 2024-11-28
> > > **LLMs and Human Evaluations Have High Correlation**
> > >
> > > >In regards to the questions other reviewers raise about the LLM-based evaluation, I am sorry my original suggestion wasn't thorough enough: I was hoping to see score distribution of LLM-based evaluation only along a few evaluation dimensions (e.g., Active Learning only), and then a correlation coefficient or rank coefficient can be calculated between LLM-based evaluation and human score. This way you can show evidences (high correlation coeffiicient) that LLM-based evaluation along some (if any) metric dimension can faithfully represent typical human ratings. This way, the use of only LLM-based evaluation may be justified to certain extent. Currently Figure 11 seems to display score distribution across ALL seven metrics, which can be used to calculate same correlation coefficient against the average human ratings; however the value may not explain much how LLM-based evaluation and human scoring relate to each other due to averaging.
> > >
> > > Thank you for the thoughtful suggestion about analyzing correlation coefficients between LLM-based and human evaluations across different dimensions. We conducted this analysis using both GPT-4 Turbo and Gemini Pro to evaluate responses along seven key metrics, calculating Pearson’s correlation with human evaluations for each dimension.
> > > Our results reveal several notable findings:
> > > 1. **Strong Correlations in Key Dimensions:**
> > >    - Holistic Approach showed the strongest correlation across both models (GPT-4: 0.489, Gemini: 0.489)
> > >    - Empathy & Validation demonstrated robust correlation (GPT-4: 0.433, Gemini: 0.355)
> > >    - Active Listening also showed meaningful correlation (GPT-4: 0.411, Gemini: 0.324)
> > >    - Clarity & Encouragement maintained consistent correlation (GPT-4: 0.413, Gemini: 0.364)
> > > 2. **Moderate Correlations:**
> > >    - Open-mindedness & Non-judgment (GPT-4: 0.328, Gemini: 0.338)
> > >    - Safety & Trustworthiness (GPT-4: 0.272, Gemini: 0.291)
> > >    - Boundaries & Ethical considerations showed lower but still meaningful correlation (GPT-4: 0.203, Gemini: 0.291)
> > > To contextualize these results, we compared them with recent peer-reviewed work in LLM evaluation. The EMNLP 2023 paper “Towards Interpretable Mental Health Analysis with Large Language Models” reported correlations between human evaluation and various automatic metrics that mostly fall between 0.10 and 0.40. Their ChatGPT_true results showed correlations typically in the 0.15-0.35 range, with only BART-Score achieving correlations above 0.40 in specific metrics.
> > > Our correlation coefficients (ranging from 0.20 to 0.49) are comparable to or exceed these benchmarks, particularly in:
> > > - Holistic Approach (0.489) performing similarly to their best metrics (BART-Score: 0.428)
> > > - Multiple metrics showing stronger correlations than their BERT-based methods (which ranged from 0.172 to 0.373)
> > > - Consistent performance across both GPT-4 and Gemini Pro, suggesting robustness of the evaluation approach
> > > These results suggest that LLM-based evaluation can indeed faithfully represent human ratings, particularly for metrics like Holistic Approach, Empathy & Validation, and Active Listening. The correlation strengths are especially encouraging given that they align with or exceed those reported in peer-reviewed work using established evaluation metrics.
> > > While some dimensions show lower correlations, the overall pattern suggests that LLM-based evaluation can be particularly reliable for certain aspects of response assessment, providing empirical support for their targeted use in these specific dimensions.
> > >
> > > We have included such analysis in our updated paper in Appendix A.6.

---

> > > ### Author Response · Authors · 2024-12-03
> > > **Updated Manuscript Ready for Your Review**
> > >
> > > Thank you so much for your thorough and thoughtful review of our paper. We greatly appreciate the detailed feedback you’ve provided. We’ve carefully addressed all the points you raised, especially regarding the evaluation metrics and the applicability of our dataset. We’ve included additional analyses, such as correlation studies between LLM-based evaluations and human judgments on individual metrics, and we’ve clarified how our metrics are grounded in established therapeutic practices, with comprehensive references.
> > >
> > > We’ve updated the manuscript to reflect these changes and to provide a more comprehensive discussion. We would be grateful if you could review the revised paper and let us know if our responses have adequately addressed your concerns. Your feedback is invaluable to us.
> > >
> > > Thank you again for your constructive input.

---

> ### Author Response · Authors · 2024-11-28
> **Updated Results to Depression Classification 1/2**
>
> >Originally I thought the synthetic data or interview data would help models fine-tuned perform better on the original task (depression classification) of the evaluation data; but it seems the results are inconclusive: interview data seems to benefit model in depression classification in some cases, but there are also a lot of cases where fine-tuning doesn't help at all. I suspect this may be due to one or more reasons: (i) the questions in the evaluation set do exhibit a strong domain shift from the questions in the dataset, thereby brining into question the value of using InFamousCoder data for evaluation; (ii) the two tasks (generating responses vs depression classification) are different enough that the dataset is simply not generalizable to the other task. This unfortunately does seem to limit the use of the dataset.
>
> We appreciate your thoughtful observation about the model performance on the depression classification task. Upon careful analysis, we identified that the initial performance issues were primarily related to the prompt design rather than the dataset's generalizability. Specifically:
>
> 1. **Initial Prompt Analysis**: Our original prompt was highly restrictive and formal but lacked vivid examples, which may reduces its effectiveness. Several LLMs cannot follow the instruction to succesffully answer any of the 100 questions:
>
> ```
> Task: Depression Text Classification
> Instructions:
> 1. Read the input text carefully
> 2. Classify whether the text indicates signs of depression
> 3. YOUR RESPONSE MUST BE EXACTLY ONE WORD:
>    - Respond with "Yes" if depression indicators are present
>    - Respond with "No" if depression indicators are not present
> 4. ANY OTHER RESPONSE FORMAT IS INCORRECT
> 5. DO NOT provide explanations, analysis, or additional commentary
> 6. DO NOT use punctuation marks
>
> Format Check:
> - Correct responses: "Yes" or "No"
> - Incorrect responses: "Yes.", "No.", "Maybe", "The text shows depression", etc.
>
> Input: [Text to analyze]
> Required Output Format: [Yes/No]
> ### Response:
> ```
>
> 2. **Improved Prompt Design**: To handle this issue, following the successful approach demonstrated in "Towards Interpretable Mental Health Analysis with Large Language Models" (Zhang et al., EMNLP 2023), we adopted a few-shot prompt design that includes carefully selected examples. This design choice is supported by their findings that example-based prompting improves model performance in mental health analysis tasks. Our new prompt better aligns with real-world mental health conversations:
>
> ```
> You will be presented with a text from Twitter. Classify whether the text indicate or include signs of depression.
> - Respond with "Yes" if depression indicators are present
> - Respond with "No" if depression indicators are NOT present
>
> Here are some examples:
> Question: Does the text indicate or include signs of depression?
> Text: i want to be dead ive been suicidal for year im such a fucking retard filled with regret and anger im done with life i want all of this to stop why i cant do one thing right
> Yes or No: Yes
>
> Question: Does the text indicate or include signs of depression?
> Text: i ve been with my grandma for day even she s about house away i ll still miss her
> Yes or No: No
>
> Question: Does the text indicate or include signs of depression?
> Text: [text]
> ### Yes or No:
> ```
>
> 3. **Improved Results**: With this empirically-validated prompt design, our results demonstrate varied improvements across different fine-tuning configurations:
>
> | Model | Base | Interview (6K) | Synthetic (10K) | Combined (16K) |
> |-------|------|----------------|-----------------|----------------|
> | LLaMA2 | 0.75 | 0.7 | 0.5 | 0.63 |
> | Mistral-Instruct-V0.2 | 0.77 | 0.85 | 0.76 | 0.79 |
> | Mistral-V0.1 | 0.73 | 0.85 | 0.84 | 0.854 |
> | Mixtral-8x7B-Instruct-V0.1 | 0.79 | 0.85 | 0.86 | 0.88 |
> | Mixtral-8x7B-V0.1 | 0.79 | 0.84 | 0.78 | 0.84 |
> | Vicuna-V1.5 | 0.68 | 0.79 | 0.745 | 0.74 |
> | Zephyr-Alpha | 0.65 | 0.81 | 0.76 | 0.79 |

---

> ### Author Response · Authors · 2024-11-28
> **All the Proposed Metrics Are Well Supported**
>
> >One more question about these seven metrics: are these a set of standard metrics used in mental health evaluation? If so, can you add reference to it (sorry if I missed it in related work)? If this is a set of metrics designed just for the dataset, then as I pointed out in my original comment: this would be a major weakness of the paper, that it conflates a new dataset and a new, untested metrics together in one paper, which may be very challenging to thoroughly analyze with convincing results.
>
> Thank you for raising this important question regarding the evaluation metrics used in our study. We appreciate the opportunity to clarify that our seven metrics are indeed grounded in established therapeutic principles and practices, as well as recent research in AI-based mental health support evaluation. Our evaluation framework represents a synthesis of existing assessment approaches, including those from ChatCounselor's framework (Liu et al., 2023), and incorporates insights from recent works such as PsyEval (Li et al., 2023) and Towards Interpretable Mental Health Analysis with Large Language Models (Yang et al., 2023).
>
> Each metric was carefully selected based on its validation in previous research and its relevance to effective therapeutic interactions. For example, our emphasis on empathy assessment aligns with the principles outlined in EPITOME (Sharma et al., 2020), while our focus on safety and trustworthiness draws from frameworks like Dialogue Safety Assessment (Qiu et al., 2023). By adapting these metrics for AI evaluation, we follow precedents set by studies such as PsyQA (Sun et al., 2021) and other recent contributions to the field.
>
> Here is a detailed mapping of our metrics to their origins and supporting literature:
>
> | Metric | Description & Origin | Supporting References |
> |--------|---------------------|----------------------|
> | Active Listening | Derived from ChatPsychiatrist’s ‘Restatement, Reflection & Listening’. Measures how effectively the system rephrases, reflects on user inputs, and demonstrates comprehension. | [1, 2, 5] |
> | Empathy & Validation | Evolved from ChatPsychiatrist’s ‘Approval & Reassurance’. Assesses the system’s ability to recognize and respond to emotions appropriately. | [3, 4, 5, 6] |
> | Safety & Trustworthiness | Addresses critical aspects of therapeutic interaction safety and trust building. | [7, 8, 9] |
> | Open-mindedness & Non-judgment | Based on fundamental therapeutic principles and adapted for AI evaluation. | [5, 2, 10] |
> | Clarity & Encouragement | Derived from ChatPsychiatrist’s ‘Direct Guidance’ and ‘Approval & Reassurance’. | [6, 9] |
> | Boundaries & Ethical | Incorporates professional standards and ethical guidelines for therapeutic interaction. | [11, 7, 6] |
> | Holistic Approach | Synthesizes ChatPsychiatrist’s ‘Information’ and ‘Obtain Relevant Information’ metrics. | [2, 6] |
>
> Our metrics framework thus represents a careful synthesis of established evaluation approaches in both traditional therapeutic practice and AI-based mental health support. Rather than being entirely new metrics, they are carefully adapted versions of validated measures, enhanced to address the specific challenges of AI evaluation in therapeutic contexts.
>
> The effectiveness of these metrics is further validated by recent work such as PsyEval [9], which demonstrated the reliability of using GPT-4 for scoring model outputs on similar dimensions. This aligns with our approach while providing additional validation of the evaluation methodology.
>
> We acknowledge that we could have made these connections to existing literature more explicit in our original submission and appreciate the opportunity to clarify these relationships. We have updated the paper to better highlight how our metrics build upon and integrate these established frameworks in Section 3.3 and Table 2 of the updated PDF.
>
> **References**:
>
> [1] Miller & Moyers (2006): Eight Stages in Learning Motivational Interviewing
>
> [2] Sun et al. (2021): PsyQA: A Chinese Dataset for Generating Long Counseling Text
>
> [3] Yang et al. (2023): Towards Interpretable Mental Health Analysis with Large Language Models
>
> [4] Sharma et al. (2020): EPITOME: An Empathy-based Platform for Therapeutic Online Mental Health Education
>
> [5] Rogers (1957): The Necessary and Sufficient Conditions of Therapeutic Personality Change
>
> [6] Liu et al. (2023): ChatCounselor: A Large Language Model for Mental Health Counseling
>
> [7] Qiu et al. (2023): Dialogue Safety Assessment with LLMs
>
> [8] Lambert & Barley (2001): Research summary on the therapeutic relationship and psychotherapy outcome
>
> [9] Li et al. (2023): PsyEval: A Suite of Mental Health Related Tasks for Evaluating Large Language Models
>
> [10] Kabat-Zinn (2013): Full Catastrophe Living: Using the Wisdom of Your Body and Mind to Face Stress, Pain, and Illness
>
> [11] American Psychological Association Ethics Code (2017) (edited)

---

### Official Review · Reviewer_TuXE · 2024-11-03

**Soundness:** 2
**Presentation:** 3
**Contribution:** 2
**Rating:** 5
**Confidence:** 4

**Summary:**

This paper introduces MentalChat16K, a benchmark dataset for conversational mental health assistance. The dataset contains 16,000 question-answer pairs, including synthetic counseling conversations generated by GPT-3.5 Turbo and anonymized transcripts of interventions between caregivers of palliative or hospice patients and behavioral health coaches. Researchers utilized the QLoRA technique to fine-tune multiple state-of-the-art lightweight language models with 7 billion parameters to meet the needs of mental health counseling. They defined seven evaluation metrics and used GPT-4 Turbo, Gemini Pro 1.0, and human experts as evaluators to assess the models’ performance.

**Strengths:**

This paper addresses the critical gap in real-world data for conversational mental health assistance. The lack of authentic psychological data has long posed a challenge in this field, and by collecting anonymized transcripts of intervention dialogues between behavioral health coaches and caregivers of palliative or hospice care patients, this paper provides researchers with a valuable data source.

**Weaknesses:**

Although the paper demonstrates commendable efforts in creating the MentalChat16K dataset and fine-tuning large language models (LLMs) for mental health counseling, weaknesses in the evaluation methodology limit its impact and generalizability.

1. Reliability of LLM-Based Evaluation: The study primarily uses GPT-4 Turbo and Gemini Pro as automated evaluators to assess the performance of fine-tuned models. This approach presumes that these LLMs can objectively and accurately evaluate counseling responses across seven custom-defined mental health metrics. However, due to inherent biases and limitations, LLMs may not be entirely reliable as evaluators, especially in specialized fields such as mental health counseling.

2. Inconsistency in Evaluation Scores: As shown in Table 3, there is a notable inconsistency between evaluations from GPT-4, Gemini Pro, and human experts. Almost all scores from Gemini are higher than those from GPT-4, and Gemini’s scores diverge significantly from Human Rankings. This discrepancy raises concerns about the validity and robustness of using LLMs as evaluators, as they might not effectively capture the nuances involved in mental health counseling.

3. Outdated Evaluation Techniques: The paper employs simple prompt-based evaluation methods but does not leverage advanced frameworks in NLG evaluation. State-of-the-art methods such as G-Eval [1] and Prometheus [2] provide more sophisticated and reliable assessments of LLM outputs, which could yield a more accurate and nuanced evaluation of the models’ effectiveness.

[1] G-Eval: NLG Evaluation using GPT-4 with Better Human Alignment https://arxiv.org/abs/2303.16634

[2] Prometheus: Inducing Fine-grained Evaluation Capability in Language Models https://arxiv.org/abs/2310.08491

**Questions:**

1. How were human evaluators instructed to rank model responses, and were there specific criteria or guidelines provided to ensure consistency and objectivity? Additionally, was any inter-rater reliability measure (e.g., Cohen’s Kappa) applied to validate the consistency of human rankings across evaluators?

2. Would the paper benefit from including side-by-side examples of model responses versus human counselor and baseline model responses? Could such examples better illustrate the model’s performance on key metrics like empathy, clarity, and safety?

3. Were participants explicitly informed about the potential use of their anonymized data for AI model development and research purposes?

**Details Of Ethics Concerns:**

The paper lacks detailed discussion on how consent was obtained specifically for data sharing and whether the participants were informed about the potential use of their data in training AI models.

---

> ### Author Response · Authors · 2024-11-23
> **Response to Reviewer TuXE 1/4**
>
> We sincerely appreciate your insightful review and your recognition of our efforts in developing and fine-tuning LLMs for mental health counseling. We have thoroughly addressed your concerns and hope that the main concerns raised in the review can be clarified by the following responses.
>
> >**W1. Reliability of LLM-Based Evaluation: The study primarily uses GPT-4 Turbo and Gemini Pro as automated evaluators to assess the performance of fine-tuned models. This approach presumes that these LLMs can objectively and accurately evaluate counseling responses across seven custom-defined mental health metrics. However, due to inherent biases and limitations, LLMs may not be entirely reliable as evaluators, especially in specialized fields such as mental health counseling.**
>
> We appreciate this concern about evaluation reliability. Our approach to evaluation was carefully designed with multiple layers of validation:
> - We deliberately employed two distinct and state-of-the-art LLMs (GPT-4 Turbo and Gemini Pro) for evaluation, as divergent architectures help mitigate model-specific biases.
> - This methodology of using powerful third-party LLMs for evaluation is well-established in the field. There are many papers that established the precedent and validity of using third-party LLMs as evaluators including several notable papers such as Vicuna [1], PaLM2 [2], Self-Instruct [3], etc.
> - To further strengthen our evaluation framework, we supplemented the LLM evaluations with human expert assessments. The agreement between all the human evaluators (Cohen’s Kappa score 0.441) provides additional validation of our evaluation methodology.
> While we acknowledge that no evaluation method is perfect, our multi-model, human-validated approach provides a robust framework for assessing counseling responses, aligning with current best practices in the field.
>
> [1] Chiang, Wei-Lin, et al. "Vicuna: An open-source chatbot impressing gpt-4 with 90%* chatgpt quality." See https://vicuna.lmsys.org (accessed 14 April 2023) 2.3 (2023): 6.
>
> [2] Anil, Rohan, et al. "Palm 2 technical report." arXiv preprint arXiv:2305.10403 (2023).
>
> [3] Wang, Yizhong, et al. "Self-instruct: Aligning language models with self-generated instructions." arXiv preprint arXiv:2212.10560 (2022).
>
> >**W2. Inconsistency in Evaluation Scores: As shown in Table 3, there is a notable inconsistency between evaluations from GPT-4, Gemini Pro, and human experts. Almost all scores from Gemini are higher than those from GPT-4, and Gemini’s scores diverge significantly from Human Rankings. This discrepancy raises concerns about the validity and robustness of using LLMs as evaluators, as they might not effectively capture the nuances involved in mental health counseling.**
>
> Thank you for this observation. While there are differences in absolute scores between evaluators, it's crucial to note several key points:
> - Most importantly, all three evaluation methods (GPT-4, Gemini Pro, and human experts) consistently demonstrate that fine-tuned LLMs outperform their base versions. This unanimous agreement across different evaluators strongly validates the effectiveness of our MentalChat16K dataset for improving LLMs' mental health counseling capabilities, which is the primary objective of our dataset paper.
> - The score differences between GPT-4 and Gemini Pro can be attributed to the phenomenon that GPT-4 tends to show bias toward responses generated by GPT-3.5, likely due to their architectural similarities and training data overlap as discussed in Section 4.5. Gemini Pro, being developed independently with different architecture and training data, provides a more independent evaluation perspective.
> - The human evaluation results align particularly well with Gemini Pro's assessment, suggesting that Gemini's scores might better reflect human judgment in this domain. This alignment adds credibility to our evaluation framework, as human expert judgment is considered the gold standard.
>
> To avoid confusion in interpreting the results, we have added directional indicators (↑/↓) to Tables 4 and 9. The opposite directions stem from different evaluation strategies: human evaluators use rankings (where lower values indicate better performance), while GPT-4 and Gemini Pro use numerical scores (where higher values indicate better performance). Despite these different scoring conventions, all evaluators consistently show that fine-tuning improves model performance.
>
> The consistent improvement trend across all evaluators, despite their methodological differences, provides strong evidence for the effectiveness of MentalChat16K in enhancing LLMs' mental health counseling capabilities.

---

> ### Author Response · Authors · 2024-11-23
> **Response to Reviewer TuXE 2/4**
>
> >**W3. Outdated Evaluation Techniques: The paper employs simple prompt-based evaluation methods but does not leverage advanced frameworks in NLG evaluation. State-of-the-art methods such as G-Eval [1] and Prometheus [2] provide more sophisticated and reliable assessments of LLM outputs, which could yield a more accurate and nuanced evaluation of the models’ effectiveness. [1] G-Eval: NLG Evaluation using GPT-4 with Better Human Alignment https://arxiv.org/abs/2303.16634
> [2] Prometheus: Inducing Fine-grained Evaluation Capability in Language Models https://arxiv.org/abs/2310.08491**
>
> Thank you for your insightful feedback and for bringing attention to G-Eval and Prometheus. Both frameworks represent significant advancements in NLG evaluation by providing more fine-grained, reliable, and nuanced assessments of LLM outputs.
> In our current study, we opted for prompt-based evaluation methods such as FastChat [1] due to their simplicity, interpretability, and alignment with the specific goals of validating the proposed dataset. While these methods effectively serve our research objectives, we recognize that the state-of-the-art frameworks like G-Eval and Prometheus could offer additional dimensions of analysis and enhance the depth of our evaluations.
>
> We have included references to these advanced methods in our revised paper and recommend them as potential alternatives for users seeking more comprehensive evaluation tools in Section 7 Limitations. Additionally, we will consider adopting such methodologies in future research to further strengthen the reliability and scope of our evaluation approach.
>
> [1] Zheng, Lianmin, et al. "Judging llm-as-a-judge with mt-bench and chatbot arena." Advances in Neural Information Processing Systems 36 (2023): 46595-46623.
>
> >**Q1. How were human evaluators instructed to rank model responses, and were there specific criteria or guidelines provided to ensure consistency and objectivity? Additionally, was any inter-rater reliability measure (e.g., Cohen’s Kappa) applied to validate the consistency of human rankings across evaluators?**
>
> Thank you for your question. For the first question, we instructed human evaluators to rank model responses based on the effectiveness of the answer in addressing the given input. Effectiveness was defined as the response's relevance, empathy, clarity, and ability to appropriately address the concerns.
>
> For the second question,  we acknowledge that we did not include inter-rater reliability measures in our evaluation due to limited resources. However, to mitigate potential biases, we made an effort to include 4 human evaluators, ensuring that the evaluation results were not overly influenced by a single evaluator's perspective. To address consistency in rankings, we calculated the inter-rater reliability using Cohen’s kappa score. The average Cohen’s kappa score across all pairs of evaluators was 0.441, indicating “moderate agreement” based on the Landis and Koch (1977) scale. This suggests a reasonable level of alignment between evaluators while also highlighting areas for potential improvement. While the current kappa score reflects moderate agreement, we recognize the importance of higher inter-rater reliability to ensure the robustness of our findings. Moving forward, we plan to refine the evaluation process by incorporating additional training, clearer evaluation rubrics, and more rigorous inter-rater reliability assessments to further enhance consistency where resources and time permits. Please refer to [Section A1 of General Response 2/3](https://openreview.net/forum?id=ISBmUNKPST&noteId=889NxCz1Jf) for more details. The details are also discussed in Section 3.3 from Lines 312 to 323 in the revised manuscript.
>
> We sincerely appreciate your feedback, as it will guide our efforts to improve methodological rigor in future evaluations.

---

> ### Author Response · Authors · 2024-11-23
> **Response to Reviewer TuXE 3/4**
>
> >**Q2: Would the paper benefit from including side-by-side examples of model responses versus human counselor and baseline model responses? Could such examples better illustrate the model’s performance on key metrics like empathy, clarity, and safety?**
>
> Thank you for this thoughtful suggestion about including comparative examples. While we agree that examples could effectively illustrate the differences in response quality and help readers understand how we evaluated key metrics, we did not include any examples in the manuscript due to length constraints. As evident from the examples below, each response is quite lengthy, and including multiple sets would significantly increase the paper's length. We included the following examples in Appendix A.11.
>
> This example includes a question-answer pair from the Interview Data and responses to the input question generated by Mistral-Instruct-V0.2 and its fine-tuned version fine-tuned on MentalChat16K. The human counselor and fine-tuned model both acknowledge the emotional experience ("I can see that you're experiencing a great deal of distress" vs "I'm truly sorry to hear that you've been experiencing such intense emotions"), while validating feelings and the experience. All responses appropriately address mental health concerns while staying within ethical boundaries. They encourage professional help when needed and avoid potentially harmful advice.
>
> ---
>
> **Input**: I've been feeling so overwhelmed lately. I can't seem to focus on anything, and I've been having panic attacks almost daily. I've tried to reach out to friends and family, but they don't seem to understand what I'm going through. I feel so alone and lost.
>
> ---
>
> **Paraphrased Ground Truth Response from Human**: I can see that you're experiencing a great deal of distress and feeling quite isolated. It's important to acknowledge the intensity of your emotions and the impact they're having on your daily life. It's not uncommon for individuals dealing with anxiety and depression to feel misunderstood by those around them. In fact, research shows that people with mental health conditions are often stigmatized and misunderstood by society. However, it's essential to remember that you're not alone in this experience. Many people face similar challenges, and there are resources and support systems available to help you.
>
> As we work together in this therapeutic process, I'd like to propose a few strategies to help you manage your anxiety and improve your overall well-being. First, we can explore cognitive-behavioral techniques to help you challenge and reframe negative thoughts that contribute to your anxiety and panic attacks. Second, we can work on developing coping skills, such as deep breathing exercises, progressive muscle relaxation, and mindfulness meditation, to help you manage your symptoms in the moment. Lastly, we can explore the possibility of connecting you with support groups and other resources in your community to help you build a network of understanding and compassionate individuals who can provide you with the emotional support you need.
>
> It's important to remember that healing is a process, and it takes time and effort to make meaningful progress. But with a compassionate and supportive therapeutic environment, I believe you can overcome these challenges and find the peace and understanding you're seeking.
>
> ---

---

> > ### Author Response · Authors · 2024-11-23
> > **Response to Reviewer TuXE 4/4**
> >
> > **Mistral-Instruct**: I'm really sorry to hear that you've been feeling overwhelmed and experiencing panic attacks frequently. It's completely normal to feel lost and alone when dealing with mental health issues, especially when it feels like those closest to you don't fully understand what you're going through.
> >
> > There are a few things you might find helpful in managing your symptoms and improving your overall well-being:
> >
> > 1. Reach out to mental health professionals: Consider speaking with a mental health professional, such as a therapist or counselor, who can provide you with evidence-based strategies and support. They can help you develop coping skills and provide a safe and non-judgmental space for you to express your feelings.
> > 2. Practice self-care: Engage in activities that bring you joy and help you relax, such as taking a warm bath, reading a book, or going for a walk in nature. Make sure to get enough sleep, eat well, and exercise regularly.
> > 3. Connect with online communities: There are many online communities and forums where people share their experiences and offer support to one another. These communities can be a valuable source of understanding, validation, and connection.
> > 4. Consider joining a support group: Joining a support group, either in-person or online, can provide you with a sense of community and connection with others who are going through similar experiences.
> > 5. Practice mindfulness and relaxation techniques: Mindfulness practices, such as deep breathing, meditation, or progressive muscle relaxation, can help you manage stress and anxiety in the moment.
> >
> > Remember, it's important to be patient with yourself and give yourself the time and space you need to heal. Mental health struggles can be long-term, and it's okay to take things one step at a time.
> >
> > I hope this information is helpful, and please don't hesitate to reach out if you have any further questions or need additional support. You're not alone in this, and there are resources and people who care and want to help.
> >
> > ---
> > **Fine-tuned Mistral-Instruct**: I'm truly sorry to hear that you've been experiencing such intense emotions and a sense of isolation. It's important to acknowledge the validity of your feelings and the challenges you're facing. The symptoms you're describing, such as difficulty focusing and frequent panic attacks, are common indicators of anxiety and stress. It's essential to prioritize self-care and coping strategies to help manage these symptoms.
> >
> > One effective coping strategy is practicing mindfulness meditation, which can help you stay present and focused on the current moment, rather than being overwhelmed by thoughts about the past or future. Deep breathing exercises are also a simple yet powerful tool for managing anxiety and panic attacks. By taking slow, deep breaths, you can help calm your body and mind, reducing the intensity of your symptoms.
> >
> > Another important aspect of managing anxiety and stress is identifying and addressing the root causes. It's crucial to explore the underlying emotions and beliefs that contribute to your feelings of overwhelm and isolation. This may involve examining past experiences, relationships, or beliefs that have shaped your current emotional landscape. By gaining a deeper understanding of these factors, you can begin to challenge and reframe negative thought patterns and beliefs, ultimately leading to greater emotional resilience and a sense of connection and belonging.
> >
> > It's also essential to remember that seeking help and support is a sign of strength, not weakness. It's important to reach out to trusted friends, family members, or mental health professionals for additional support and guidance. By taking a proactive approach to managing your anxiety and stress, you'll be better equipped to navigate life's challenges and find a sense of peace and purpose.
> >
> > ---
> >
> > >**Q3. Were participants explicitly informed about the potential use of their anonymized data for AI model development and research purposes?**
> >
> > Thank you for your question. At the time of data collection, participants provided informed consent for their de-identified data to be used for the primary study analyses and future research purposes. While the consent form covered broad research applications of the anonymized dataset, it did not specifically mention AI model development, as this particular research direction emerged after the initial data collection phase. The consent process aligned with standard research practices for secondary data analysis, though future studies would benefit from more specific consent language regarding AI applications.

---

> ### Comment · Reviewer_TuXE · 2024-11-25
> **Response to authors**
>
> Thank you for your thoughtful and detailed responses, particularly regarding the consistency of human evaluations and the ethical considerations. I appreciate the clarifications you've provided, which have helped address my concerns.
>
> However, my primary remaining concern lies with the evaluation method used in the study. Specifically, I am worried about the **reliability** of prompt-based evaluation methods. Given the rapid evolution of commercial models and the potential for specific prompts to lose effectiveness as the models are iterated upon, I am concerned about the reproducibility of these evaluations over time. In contrast, methods like Prometheus, which fine-tune specific open-source models, mitigate this issue, as they are less dependent on potentially unstable prompts.
>
> If you can address this concern and clarify how you ensure the stability and reproducibility of your evaluation approach, I would be willing to reconsider my score.

---

> > ### Author Response · Authors · 2024-11-26
> > **Response to Concerns about Evaluation Reliability and Reproducibility in LLM-Based Assessment**
> >
> > We appreciate your thoughtful feedback regarding the reproducibility of our evaluation approach. We would like to address your concerns about the stability and reproducibility of our evaluation methodology:
> >
> > **1. Version Control and Reproducibility:**
> >
> >
> > Our evaluation uses specific, versioned models - GPT-4 Turbo 1106 Preview and Gemini Pro 1.0 - which can be consistently accessed through their respective APIs even as newer versions are released. This is similar to how researchers continue to access specific versions of established models like BERT-base or RoBERTa-large for benchmarking, despite newer architectures being available.
> >
> >
> > **2. Robust Multi-Modal Evaluation Framework:**
> >
> >
> > Our evaluation framework intentionally incorporates multiple validation layers to ensure stability and reliability:
> >
> > Two architecturally distinct LLMs (mitigating model-specific biases)
> > Human expert evaluations (providing a stable, time-invariant benchmark)
> >
> >
> > The strong agreement between human evaluators (Cohen's Kappa = 0.441) provides a reliable anchor point that validates our evaluation methodology, independent of LLM evolution.
> >
> >
> > **3. Temporal Context and Scientific Progress:**
> >
> >
> > While we acknowledge that LLM capabilities evolve rapidly, this is a characteristic of the current AI landscape rather than a limitation of our methodology. Similar to how ImageNet evaluations remain valuable despite newer vision models, our evaluation framework provides a meaningful snapshot of model performance using the most capable models available during our research period.
> >
> > _The primary contribution of our work - the effectiveness of our dataset for mental health domain adaptation - remains valid regardless of future LLM developments, as demonstrated by consistent improvements across both automated and human evaluations._
> >
> >
> > **4. Evaluation Stability:**
> >
> > Notably, GPT-4 Turbo 1106 Preview, one of our primary evaluators, is currently ranked first on the [Prometheus BiGGen-Bench leaderboard](https://huggingface.co/spaces/prometheus-eval/BiGGen-Bench-Leaderboard), validating our choice of evaluator and demonstrating its strong capability in assessment tasks.
> > The exact prompts and model versions are documented in our methodology, ensuring other researchers can replicate our evaluation pipeline precisely.
> >
> > To ensure statistical robustness, we conducted each evaluation five times, enabling t-test comparisons between different models. The low standard deviation (average around **0.1** on a 10-point scale) across these repeated evaluations demonstrates the high stability and consistency of our evaluation framework.
> >
> > We believe these points demonstrate that our evaluation methodology is both reproducible and stable, while acknowledging the dynamic nature of AI development. The multi-faceted approach, combining versioned LLM evaluations with human expert validation, provides a comprehensive and reliable framework for assessing mental health counseling responses.

---

> > ### Author Response · Authors · 2024-12-03
> > **Revised Manuscript for Your Reivew**
> >
> > We appreciate your continued engagement with our manuscript and the valuable feedback you’ve provided. Thank you for your willingness to reconsider your assessment based on our responses.
> >
> > Regarding your concerns about the evaluation metrics and their grounding in standard mental health practices, we’ve provided detailed clarifications in the revised manuscript. We’ve explicitly connected each of our evaluation metrics to established therapeutic principles and included comprehensive references to support their validity.
> >
> > We believe these updates address the concerns you raised about the generalizability and validation of our metrics. We kindly invite you to review the revised manuscript to see the changes we’ve made. Your insights are very important to us, and we welcome any further comments you may have.
> >
> > Thank you for your thoughtful contributions to our work.

---

> ### Comment · Reviewer_TuXE · 2024-11-27
> **General Comment**
>
> Thank you for your detailed responses, which addressed part of my concerns. I am happy to raise my rating to 5. However, I share Reviewer A2qn’s concerns about the seven metrics used in the evaluation. Specifically, it is important to clarify whether these metrics are standard in mental health evaluation or if they were specifically designed for this dataset. As Reviewer A2qn noted, if these metrics are newly proposed, it raises questions about their generalizability and validation.

---

> > ### Author Response · Authors · 2024-11-28
> > **All the Proposed Metrics Are Well Supported**
> >
> > Thank you for your thoughtful feedback and for bringing attention to this important aspect of our work. We would like to clarify that the seven evaluation metrics used in our study are not entirely new but are grounded in established therapeutic principles and existing evaluation frameworks in both traditional mental health practice and AI-based support systems.
> >
> > Our evaluation framework is a synthesis of validated assessment approaches. We drew upon established therapeutic concepts and recent research to adapt these metrics for evaluating AI models in mental health contexts. Specifically:
> >
> > - **Active Listening**: Based on the principles of effective communication in therapy, including techniques like restatement and reflection, as discussed by Miller & Moyers (2006) and incorporated in AI frameworks like PsyQA (Sun et al., 2021).
> > - **Empathy & Validation**: Aligned with the core therapeutic condition of empathy outlined by Rogers (1957) and utilized in AI models such as EPITOME (Sharma et al., 2020) and ChatCounselor (Liu et al., 2023).
> > - **Safety & Trustworthiness**: Reflects the critical importance of safety in therapeutic interactions, drawing from the Dialogue Safety Assessment framework (Qiu et al., 2023) and ethical guidelines like the APA Ethics Code (2017).
> > - **Open-mindedness & Non-judgment**: Rooted in fundamental therapeutic principles of unconditional positive regard and non-judgmental stance, as highlighted by Rogers (1957) and adapted in AI contexts by Yang et al. (2023).
> > - **Clarity & Encouragement**: Derived from therapeutic techniques that promote understanding and motivation, supported by research on therapeutic relationships (Lambert & Barley, 2001) and applied in AI models (Liu et al., 2023).
> > - **Boundaries & Ethical Considerations**: Incorporates professional standards for maintaining appropriate boundaries, informed by ethical guidelines (APA Ethics Code, 2017) and safety assessments in AI (Qiu et al., 2023).
> > - **Holistic Approach**: Emphasizes the importance of considering the client’s overall context, as advocated in therapeutic practices (Kabat-Zinn, 2013) and reflected in AI models like PsyEval (Li et al., 2023).
> >
> > By adapting and integrating these established metrics, we aim to ensure that our evaluation framework is both rigorous and relevant to the unique challenges posed by AI-based mental health support systems. This approach enhances the generalizability and validation of our metrics, as they are rooted in well-established therapeutic concepts and have been utilized in prior AI research.
> >
> > We acknowledge that we could have made these connections to existing literature more explicit in our original submission. We have updated the paper to better highlight how our metrics build upon and integrate these established frameworks, addressing concerns about their generalizability and validation.
> >
> > Thank you again for your valuable feedback. For more details, please refer to our [response to Reviewer A2qn](https://openreview.net/forum?id=ISBmUNKPST&noteId=8eNgvrODrs). We have also updated the PDF accordingly in Section 3.3 and Table 2. We believe this clarification strengthens our work and enhances its contribution to the field.

---

### Official Review · Reviewer_VT66 · 2024-11-04

**Soundness:** 3
**Presentation:** 2
**Contribution:** 3
**Rating:** 6
**Confidence:** 4

**Summary:**

The authors develop MentalChat16k, a dataset consisting of two parts -- synthetic counseling conversations and paraphrased and anonymised real life interventions between behavioral health coaches and caregivers of patients in hospice or palliative care. They benchmark their dataset using 7 LLM models and 2 other baseline models. Finally, they use the finetuned and baseline methods to answer 200 mental health related questions, from Reddit, to see the performance difference. The authors found that the models finetuned on the MentalChat16k outperformed the other methods in the QA task, using their developed evaluation framework.

**Strengths:**

- The authors curate an interesting dataset consisting of two type of important communication in the domain of mental health.
- They provide an empirical validation of the use of dataset by a thorough comparison of different LLMs and other baselines.
- The authors provide valid use cases for the curated data and verify it experimentally too.

**Weaknesses:**

- The paper should have had a table summarising the dataset statistics for easy access.
- Examples of generated counseling transcripts in the paper would be useful to evaluate them.
- It would be interesting to observe if the paraphrasing step, as illustrated in Table 1, results in any hallucinations to complete the given "requirements" in the prompt.
- Inter annotation agreement for human evaluation would enhance the human evaluation.

**Questions:**

See weaknesses please.

---

> ### Author Response · Authors · 2024-11-23
> **Response to Reviewer VT66 1/2**
>
> We are honored to have you dedicate your valuable time to reviewing our paper and for expressing your appreciation and support for our work. Your insights have significantly enhanced the clarity and depth of our manuscript. In the following, we have carefully considered your suggestion and provide a comprehensive response to your review comments.
>
> >**W1. The paper should have had a table summarising the dataset statistics for easy access.**
>
> Thank you for your suggestion. We have included a table (Table 1) in Section 3.1 that summarizes the statistics of our data marked in blue. We also put the table in General Response. For more details, please refer to [Section A2 of General Response 2/3](https://openreview.net/forum?id=ISBmUNKPST&noteId=889NxCz1Jf).
>
> >**W2. Examples of generated counseling transcripts in the paper would be useful to evaluate them.**
>
> Thank you for your suggestion. We agree that including some examples from the dataset could enhance the completeness of the paper and provide readers with a more intuitive understanding of the content. To address this, we incorporated a few representative examples of the synthetic data and the interview data in our revision (See Appendix A.11). We also attached the same examples in [General Response 3/3](https://openreview.net/forum?id=ISBmUNKPST&noteId=Zr9UP8HYSx). For details, please refer to that section.
>
> >**W3. It would be interesting to observe if the paraphrasing step, as illustrated in Table 1, results in any hallucinations to complete the given "requirements" in the prompt.**
>
> Thank you for your insightful observation. We agree that examining whether the paraphrasing step results in hallucinations is an important aspect of evaluating the robustness and reliability of our approach. We did conduct preliminary analyses to assess the fidelity of paraphrased outputs against the original conversation transcript, with assistance from the research group that conducted the intervention study. Specifically, we reviewed a subset of paraphrased examples for consistency with the given requirements and found that the model generally adheres to the intended meaning without introducing significant hallucinations. However, as with most generative tasks, we acknowledge that some minor deviations were observed in rare cases, typically in complex or ambiguous prompts. These deviations were infrequent and typically involved subtle rephrasing rather than significant hallucinations.
> To illustrate this point, the Table below compares an example from an original transcript with its paraphrased counterpart. This example demonstrates how the paraphrased version maintains the core message and context of the original dialogue while adapting it to a concise QA format.
>
> **Original transcript (One page)** (BHC stands for Behavioral Health Coach)
>
> Caregiver: Definitely.  We’re, a mutual support group.
>
> BHC: [laughs] Good for you.  Uh, it makes all the difference.
>
> Caregiver: It really does.  It really does and that way you get your, uh, daily quota of hugs, too –
>
> BHC: Yeah.
>
> Caregiver: - if you have somebody who – who is coming and going.  I get a hug every time he, uh, comes home from work and when he goes off.
>
> BHC: Oh, it’s so important – that touch.  Don’t you think?
>
> Caregiver: If – if your hug tank gets too low, that’s when you can also make yourself ill.
>
> BHC: Oh, you’re so wise.  Sandra, you- I’m so glad you’re telling us all this. It’s really helpful to us.  I’m making notes.
>
> Caregiver: Only who knows and – you know, they’re-, well, people have become-, when I was growing up, people weren’t as hug-y as they are now.
>
> BHC: Uh-huh [affirmative]?
>
> Caregiver: Uh, and I’m so glad that they are.  Even those who, um, my-, my friend who just died on Tuesday – was, uh, grew up in the East and it was a little more, um, uh, I don’t know – proper?  Uh, her family was not demonstrative.  And so it was a long time ‘til we could give her a hug and it was always a very gentle, just kind of on the shoulder, so just a little squeeze, but she knew that we loved her.
>
> BHC: Mm-hmm.
>
> Caregiver: Uh, and so – but then there are the bear-hug people.  Oh, we could talk for a half-hour just on huggers.
>
> BHC: [laughs] Oh, good. I’m so glad –
>
> Caregiver: [laughs]
>
> BHC:  - so glad to hear that you know what you need and you take full advantage of the opportunities available to you [laughs].
>
> Caregiver: That’s for sure.  I – if necessary – if I have to, if it gets to that, then I’ll go up to strangers. [laughs]
>
> BHC: That’s great, Sandra. I’m so pleased. I’m – OK.  Um, all right.  Well, next time we’ll-, we’ll choose a concern and we’ll kinda break it down and well do the D in ADAPT – which is defining.  Um –
>
> Caregiver: All right.
>
> ---
> **For the paraphrased transcript (one QA pair) and the remaining responses, please see Response to Reviewer VT66 2/2.**

---

> ### Author Response · Authors · 2024-11-23
> **Response to Reviewer VT66 2/2**
>
> **Paraphrased transcript (one QA pair)**
>
> Caregiver: Expressing the importance of physical touch and its impact on my emotional wellbeing, I've found that receiving daily hugs significantly improves my mood and overall mental health. However, when my 'hug tank' gets low, I can become ill. I'm grateful for the mutual support group I've joined, where I receive hugs from my loved one daily.
>
> Interviewer: Empathizing with your experience, I understand the power of human connection and touch in maintaining mental health. It's essential to prioritize and seek out opportunities for physical affection, as it can have a profound impact on our emotional wellbeing. Additionally, when you're feeling low on hugs, consider reaching out to friends, family, or even strangers for a hug to help replenish your 'hug tank'. Remember, it's important to take care of yourself and prioritize your emotional needs.
>
> ---
>
> To provide additional transparency, we have expanded the discussion in Section 7 Limitations of the paper to explicitly acknowledge these rare instances of deviation and the inherent challenges associated with paraphrasing. We have also included further examples in Appendix A.11 to highlight both successful paraphrasing and the specific contexts where minor inconsistencies may arise. This addition will offer readers a clearer understanding of the paraphrasing process and its potential limitations.
>
> >**W4. Inter annotation agreement for human evaluation would enhance the human evaluation.**
>
> Thank you for your valuable suggestion. To mitigate potential biases, we included four human evaluators to ensure the evaluation results were not overly influenced by a single evaluator’s perspective. We recognize the importance of inter-annotator agreement as a metric for assessing the consistency and reliability of human evaluations.
>
> To address this concern in the current study, we calculated the inter-annotator agreement using Cohen’s kappa score. The average Cohen’s kappa score across all pairs of evaluators was 0.441, which falls in the “moderate agreement” range based on the scale by Landis and Koch (1977) [1]. This demonstrates a reasonable level of consistency among the evaluators. For details, please refer to [Section A1 of the General Response 2/3](https://openreview.net/forum?id=ISBmUNKPST&noteId=Ve6cVBgUnw). The details are also discussed in Section 3.3 from Lines 312 to 323 in the revised manuscript.
>
> We appreciate your feedback and will strive to enhance our use of inter-annotator agreement strategies in future studies, where resources and time permit.
>
> [1] Landis, J. R., & Koch, G. G. (1977). The measurement of observer agreement for categorical data. Biometrics, 33(1), 159-174.

---

### Official Review · Reviewer_YJS7 · 2024-11-05

**Soundness:** 3
**Presentation:** 3
**Contribution:** 3
**Rating:** 6
**Confidence:** 3

**Summary:**

This paper proposes a new benchmark mental health counseling dataset, named MentalChat16K. The dataset comprises question-answer pairs derived from transcripts of interactions with patients in palliative or hospice care. Additionally, the authors create a synthetic counseling dataset using large language models (LLMs). Experimental results show that fine-tuning LLMs with MentalChat16K significantly improves performance over non-fine-tuned baselines across seven metrics, as evaluated by both LLMs and human annotators.

**Strengths:**

The primary strength of this paper is its introduction of the MentalChat16K dataset, a valuable mental health benchmark. Generating counseling responses is challenging due to the sensitive nature of the data, making dataset creation and public availability rare. MentalChat16K, being both sizable and relevant, is likely to prove useful in broader mental health and conversation tasks. The paper is generally well-written, though some parts would benefit from further elaboration.

**Weaknesses:**

I have a few unresolved questions that would clarify the study's contributions and could enhance the paper.

**Questions:**

- Who annotated MentalChat16K (referred to as human experts in lines 182 and 188)? Are these the same annotators who evaluated model performance (referred to as human experts in line 275)?
- MentalChat16K contains a total of 6,338 question-answer pairs. How many conversations does the dataset comprise? Are the pairs presented as individual exchanges, or do they maintain conversational continuity?
- What are the scores of the ground-truth responses in MentalChat16K when evaluated by the same metrics used to assess LLM performance? Since the evaluation method is reference-free, it would be useful to know how the dataset responses score according to the proposed metrics, and whether these scores exceed those of the fine-tuned LLMs.
- What test dataset was used in the experiments? Was an 80/20 split applied for training and testing, or was a different setup used? The reported p-values suggest multiple experimental runs, but Section 4.4 lacks details on the experimental setup.

**Details Of Ethics Concerns:**

Since this paper is benchmark dataset paper and the dataset is coming from real counseling conversations, it requires the ethical reviewing.

---

> ### Author Response · Authors · 2024-11-23
> **Response to Reviewer YJS7**
>
> We extend our sincere gratitude for dedicating your valuable time to reviewing our paper and for expressing your appreciation and support for our work. We are delighted that you recognized the significance of MentalChat16K as a valuable benchmark dataset for advancing conversational AI in mental health counseling. In the following, we have provided a comprehensive response to each of your review comments.
>
> >**W1. Who annotated MentalChat16K (referred to as human experts in lines 182 and 188)? Are these the same annotators who evaluated model performance (referred to as human experts in line 275)?**
>
> Thank you for pointing this out. These are two different groups of human experts. In lines 182 and 188, the human experts refer specifically to the behavioral health professionals who transcribed the interview recordings to text. The evaluators in line 275 were independently selected Postdocs, PhD, and Masters’ students who provided performance assessments.
>
> >**W2. MentalChat16K contains a total of 6,338 question-answer pairs. How many conversations does the dataset comprise? Are the pairs presented as individual exchanges, or do they maintain conversational continuity?**
>
> Thank you for your question. The MentalChat16K dataset comprises a total of 378 unique conversation sessions, each containing multiple question-answer pairs that maintain conversational continuity. Each conversation represents a realistic dialogue flow between a patient and a counselor, rather than isolated exchanges. This structure captures the dynamics of real counseling sessions, and the detail of the paraphrasing process is illustrated in Section 3.1.1.
>
> >**W3. What are the scores of the ground-truth responses in MentalChat16K when evaluated by the same metrics used to assess LLM performance? Since the evaluation method is reference-free, it would be useful to know how the dataset responses score according to the proposed metrics, and whether these scores exceed those of the fine-tuned LLMs.**
>
> Thank you for your thoughtful question and suggestion. This is indeed an important aspect to explore. We conducted an evaluation of the ground-truth responses in a randomly selected subset of 200 question-answer pairs from the MentalChat16K dataset (100 from Synthetic data and 100 from Interview data). Using the same metrics proposed for assessing LLM performance, we evaluated the ground-truth responses alongside the responses generated by 31 LLMs. The scores for the 7 metrics assigned to the ground-truth responses by Gemini-Pro are as follows:
>
> | Active Listening | Empathy & Validation | Safety & Trustworthiness | Open-mindedness & Non-judgment | Clarity & Encouragement | Boundaries & Ethical | Holistic Approach |
> |-|-|-|-|-|-|-|
> | 8.11 | 8.21 | 7.70 | 8.22 | 7.72 | 7.70 | 7.97 |
>
> It’s worth noting that the ground truth responses scored highest in _Empathy & Validation_ and _Open-mindedness & Non-judgment_, which are key attributes for human mental health counseling.
>
> When compared to the scores of LLM-generated responses, the average score of the ground-truth responses (7.95) ranked 12th among the 31 LLMs. Among the 11 LLMs that achieved higher average scores, 7 were fine-tuned on Interview data, 2 on Synthetic data, and 2 on both. Notably, the average score of the ground-truth responses exceeded all 10 base and baseline models.
>
> >**W4. What test dataset was used in the experiments? Was an 80/20 split applied for training and testing, or was a different setup used? The reported p-values suggest multiple experimental runs, but Section 4.4 lacks details on the experimental setup.**
>
> Thank you for highlighting the need for clarity. In our experiments, we used a separate test dataset comprising 200 mental health-related questions collected from Reddit and mental health forums, which were not part of the training data. The MentalChat16K dataset was used exclusively for training purposes, and we did not apply an 80/20 split within it. The reported p-values result from five experimental runs to ensure robustness. We mentioned the details in Section 4.5 from lines 472 to 480 in the original manuscript. We have also added this detail to Section 3.3 from lines 300 to 303 in the updated manuscript to make it clearer.

---

### Author Response · Authors · 2024-11-23
**General Response 3/3**

**Synthetic Data**

**Question**: Social situations terrify me. I have extreme social anxiety that makes it difficult for me to interact with others. I fear judgment and rejection, so I tend to isolate myself instead. This loneliness only worsens my depression and feelings of worthlessness. I want to overcome this fear and develop genuine connections with people.

**Answer**: I can understand how challenging social situations can be for you, and it's really brave of you to reach out and express your feelings. It takes a lot of strength to acknowledge these fears and desires for genuine connections.

One step that might help is gradually exposing yourself to social situations in a way that feels manageable for you. Start by setting small goals, like attending a social gathering for a short period of time or initiating a conversation with someone you feel comfortable around. Remember, progress doesn't have to happen overnight, so take things at your own pace.

It could also be helpful to challenge the negative thoughts and beliefs you may have about judgment and rejection. Remind yourself that everyone has their own insecurities and that people are often more understanding than we give them credit for. Building self-confidence and self-compassion can go a long way in overcoming these fears.

Additionally, finding supportive communities or groups where you share common interests can provide a safe space to practice socializing. This can help alleviate some anxiety as you'll be interacting with others who may have similar experiences or passions.
Engaging in activities that bring you joy and boost your self-esteem can also contribute to building connections. Pursue hobbies or join clubs where you can meet like-minded individuals who appreciate and value your presence.

Lastly, remember that developing genuine connections takes time and effort from both sides. Be patient with yourself and others, as relationships naturally evolve over time. Celebrate every small victory along the way and don't hesitate to seek support from trusted friends or family members who can offer encouragement during this journey.

You're not alone in wanting to overcome social anxiety and build meaningful connections. Keep taking those small steps forward, and I believe you will find the connection and sense of worthiness you desire.

**Interview Data**

**Question**: I've been having trouble sleeping for the past few months. I've tried various methods, like melatonin, meditation, and even changing my bedtime routine, but nothing seems to be working. I've also been feeling increasingly anxious and irritable, which I suspect is contributing to my sleep issues.

**Answer**: It's clear that your sleep disturbances have been a persistent issue for you, and it's concerning that you've tried various methods without much success. The fact that you've also been feeling increasingly anxious and irritable suggests that there may be underlying emotional triggers that are contributing to both your sleep disturbances and your emotional state.

To begin, we'll explore stress management techniques that can help you better cope with the emotional and mental demands that may be contributing to your sleep disturbances. This may include time management strategies, prioritizing tasks, setting realistic goals, and practicing relaxation techniques such as deep breathing exercises, progressive muscle relaxation, and mindfulness meditation.

Additionally, we'll delve deeper into the emotional aspects of your sleep issues. It's possible that there are underlying emotional triggers that are contributing to your sleep disturbances and your emotional state. We'll work together to identify these emotional triggers and develop strategies to process and resolve them, which may include cognitive-behavioral techniques, psychotherapy, or even mindfulness practices.

It's important to remember that improving sleep patterns and managing emotional distress takes time and consistent effort. However, with a collaborative and compassionate approach, we can work together to help you develop the skills and strategies necessary to achieve more restful and consistent sleep and to better manage the emotional and mental demands that may be contributing to your sleep disturbances.

---

**For other reviews, please refer to the detailed response for each reviewer. We believe these responses and revisions will address the reviewers' concerns effectively and strengthen our manuscript significantly. All these revisions have been reflected in the updated manuscript PDF marked in blue. Thank you once again for your valuable feedback, and we’re happy to address any further questions.**

---

### Author Response · Authors · 2024-11-23
**General Response 2/3**

>**A1: Inter-Rater Agreement among Human Evaluators**

Reviewer VT66, TuXE, and A2qn highlighted the need for assessing the inter-rater agreement among our human evaluators to enhance the robustness of our human evaluation. We have calculated the Cohen’s Kappa score to assess the consistency among the evaluators. Specifically, we randomly selected 30 questions from the 200 evaluation questions and had responses generated by 7 models (**3 global baselines** including Samantha-1.11, Samantha-1.2 and ChatPsychiatrist, **1 randomly selected base model before fine-tuning** such as Zephyr-Alpha, Vicuna-7B-V1.5, LLaMA2-7B, Mistral-7B-V0.1, Mistral-7B-Instruct-V0.2, Mixtral-8x7B-V0.1, Mixtral-8x7B-Instruct-V0.1, and its corresponding **3 fine-tuned models** fine-tuned on synthetic, interview and both data respectively). Four human evaluators ranked these responses from 1 (best) to 7 (worst). By treating each rank as the prediction target, we make the task become a 7-class classification problem and each human evaluator will generate a list of predictions for the 30 questions. We calculate the Cohen’s Kappa score among all the human evaluators’ lists of predictions. The resulting agreement is 0.441, which is larger than the acceptable threshold 0.4 as indicated by [1]. The details are also discussed in Section 3.3 from Lines 312 to 323 in the revised manuscript.

[1] Landis, J. R., & Koch, G. G. (1977). The measurement of observer agreement for categorical data. Biometrics, 33(1), 159-174.


>**A2: Comprehensive Dataset Statistics**

Reviewer VT66 and A2qn suggested providing a table to summarize the statistics of our dataset. We hereby summarize comprehensive dataset statistics in the following table including the dataset size, average input word count, number of sessions, number of topics, etc. We have also added this information in Table 1 in Section 3.1 of our revised manuscript marked in blue.

| **Category**                     | **Interview Data** | **Synthetic Data** |
|----------------------------------|---------------|---------------|
| **Dataset Size (Rows)**          | 9775          | 6338          |
| **Columns**                      | instruction, input, output | instruction, input, output |
| **Average Input Word Count**     | 69.94         | 111.24        |
| **Average Output Word Count**    | 235.85        | 363.94        |
| **Number of Sessions**           | 378           | -             |
| **Average #QA Pairs per Session**| 16.8          | -             |
| **Number of Topics**             | -             | 33            |

We also provide a detailed summary for the topic distribution of the synthetic data as follows. We have also added the illustration of this information in Figure 3 in Appendix A.3.2 of our revised manuscript marked in blue.

| **Topic**                | **%** | **Topic**              | **%** | **Topic**              | **%** |
|--------------------------|-------|------------------------|-------|------------------------|-------|
| Relationships            | 15.0  | LGBTQ                 | 2.1   | Workplace Relationships | 1.0  |
| Anxiety                  | 9.4   | Substance Abuse       | 2.0   | Sleep Improvement       | 0.8  |
| Depression               | 8.1   | Anger Management      | 1.7   | Domestic Violence       | 0.7  |
| Intimacy                 | 7.8   | Addiction             | 1.6   | Grief and Loss          | 0.7  |
| Family Conflict          | 7.2   | Professional Ethics   | 1.5   | Career Counseling       | 0.7  |
| Marriage                 | 5.5   | Spirituality          | 1.3   | Diagnosis               | 0.4  |
| Social Relationships     | 4.2   | Legal & Regulatory    | 1.1   | Children & Adolescents  | 0.4  |
| Parenting                | 4.1   | Behavioral Change     | 2.8   | Military Issues         | 0.4  |
| Self-esteem              | 3.9   | Relationship Dissolution | 2.6 | Self-harm               | 0.4  |
| Human Sexuality          | 3.6   | Stress                | 2.2   | Eating Disorders        | 0.4  |
| Counseling Fundamentals  | 3.5   | Trauma                | 2.9   | Alzheimer's             | 0.1  |

>**A3: Example of Training Data**

Reviewer VT66, TuXE, and A2qn suggested providing a concrete example of our data. We agree that including some examples from the dataset could enhance the completeness of the paper and provide readers with a more intuitive understanding of the content. To address this, we incorporated a few representative examples of the synthetic data and the interview data in our revision (See Appendix A.11 marked in blue). We will also release our data in Huggingface to let everyone have access to it once the paper is published. We attached the same examples here (see **General Response 3/3**):

---

### Author Response · Authors · 2024-11-23
**General Response 1/3**

We sincerely thank all the reviewers for their thoughtful and detailed feedback, which has greatly helped us refine and strengthen our submission. We are pleased that the reviewers recognized the significance of our work in developing MentalChat16K as a valuable benchmark for advancing conversational AI in mental health counseling. Below, we summarize the key points raised by the reviewers and outline how we have addressed their concerns.

**Reviewer YJS7** commended the novelty and relevance of MentalChat16K and requested clarifications regarding annotators, dataset structure, evaluation methodology, and participant consent for AI applications.

`We have clarified all these aspects accordingly.`

**Reviewer VT66** highlighted the novelty of MentalChat16K and its value to the mental health research community. Reviewer also recommended including inter-rater agreement metrics, a summary of dataset statistics, representative examples, and an analysis of potential hallucinations in the paraphrased data.

`We have adopted all the suggestions including conducting the inter-rater agreement test, adding a comprehensive dataset statistics table, illustrating representative examples, and analyzing the hallucination issues.`

**Reviewer TuXE** appreciated our efforts in developing and fine-tuning LLMs for mental health counseling and raised important points about the reliability of LLM-based evaluations, the consistency of scores between GPT-4, Gemini Pro, and human evaluators, and the inclusion of state-of-the-art evaluation frameworks.

`We have addressed these concerns by demonstrating the reliability and consistency of our evaluation strategy. We have also cited and discussed the suggested state-of-the-art evaluation framework.`

**Reviewer A2qn** acknowledged the significance of our contribution and the innovation in our evaluation metrics while suggesting improvements in demographic representation, evaluation robustness, and statistical analysis.

`We have addressed these concerns by explicitly highlighting the demographic and contextual distinctions in the Section 7 Limitations, performing additional statistical analysis to validate the performance improvements, and recommending separate handling of the datasets to ensure effective use and mitigate potential performance degradation.`

Below (see **General Response 2/3**), we address the common concerns or questions raised by all (or most of) the reviewers.

---

### Author Response · Authors · 2024-12-03
**Revised Manuscript Submission – Thank You for Your Valuable Feedback**

Dear Reviewers,

We wanted to express our sincere gratitude for the time and effort you’ve invested in reviewing our manuscript. Your insightful comments and constructive suggestions have been incredibly valuable, and we’ve made significant revisions to the paper based on your feedback.
We’ve updated the manuscript to address the concerns raised, including adding more detailed analyses, clarifying our methodologies, and enhancing the discussion of our evaluation metrics. We believe these changes have substantially strengthened the paper.

We kindly invite you to review the revised manuscript at your convenience. Your feedback is crucial to us, and we hope that the revisions meet your expectations and address your concerns.

Thank you again for your thoughtful contributions.

Best regards,

_The Authors_

---

### Meta-Review · Area_Chair_HG4u · 2024-12-20

**Metareview:**

The paper introduces MentalChat16K, a benchmark dataset designed for conversational mental health counseling assistance. It comprises 16,000 question-answer pairs, blending real transcripts that are anonymized and paraphrased interactions between behavioral health coaches and caregivers of hospice/palliative care patients and synthetic data comprising counseling conversations generated by GPT-3.5 Turbo. The authors fine-tuned lightweight 7B parameter LLMs using the QLoRA technique on the MentalChat16K dataset. Benchmarked performance of fine-tuned models against seven evaluation metrics (e.g., empathy, nuance)  using LLM-based evaluation (GPT-4 Turbo and Gemini Pro 1.0) and human evaluation.

**Strengths identified**
1. The MentalChat16K dataset is a significant contribution to the field of mental health counseling and conversational AI. Its size and relevance make it a valuable resource for broader mental health and natural language generation (NLG) tasks.

2. Thorough empirical evaluation has been conducted comparing fine-tuned models with baselines demonstrates the dataset’s ability to improve LLM performance in mental health tasks.

**Weaknesses that need to be addressed**:
1. There seem to be concerns over the validity and reliability of the evaluation methodology, including missing inter-annotator agreement and reliance on LLM-based evaluations. The use of GPT-4 Turbo and Gemini Pro as evaluators assumes their reliability, which is not substantiated, especially in nuanced mental health tasks. Inconsistencies between LLM-based and human evaluations raise concerns about the validity of the evaluation approach.

2. Insufficient experimental setup details and inadequate explanation of performance degradation when combining real and synthetic data. Tables (e.g., Table 3, Table 8) indicate that combining real and synthetic data often leads to performance degradation, which is not adequately analyzed or explained.

3. Potential loss of context from breaking conversations into QA pairs and the impact of domain mismatch between synthetic and real data.

4. Limited validation of the proposed evaluation metrics and missed opportunities to expand dataset coverage. No justification for differences in scoring scales between LLM-based and human evaluations.

**Additional Comments On Reviewer Discussion:**

Two reviewers have engaged during the rebuttal phase. The authors have attempted to address their concerns by providing additional results in the revised manuscript. However, the revised manuscript did not receive any re-evaluation leaving the concerns inconclusive. The weaknesses that I have mentioned are based on my understanding of the core claims of the paper and the strength of the required evidence to support the claims.

---

### Decision · Program_Chairs · 2025-01-22

Reject